# Global Impact of COVID-19 Restrictions on the Surface Concentrations of Nitrogen Dioxide and Ozone

Christoph A. Keller[1,2], Mat. J. Evans[3,4], K. Emma Knowland[1,2], Christa A. Hasenkopf[5], Sruti Modekurty[5], Robert A. Lucchesi[1,6], Tomohiro Oda[1,2], Bruno B. Franca[7], Felipe C. Mandarino[7], M. Valeria Díaz Suárez[8], Robert G. Ryan[9], Luke H. Fakes[3,4], Steven Pawson[1]

[1]NASA Global Modeling and Assimilation Office, Goddard Space Flight Center, Greenbelt, MD, USA
[2]Universities Space Research Association, Columbia, MD, USA
[3]Wolfson Atmospheric Chemistry Laboratories, Department of Chemistry, University of York, York, YO10 5DD, UK
[4]National Centre for Atmospheric Science, University of York, York, YO10 5DD, UK
[5]OpenAQ, Washington, DC, USA
[6]Science Systems and Applications, Inc., Lanham, MD, USA
[7]Municipal Government of Rio de Janeiro, Rio de Janeiro, Brazil
[8]Secretaria de Ambiente, Quito, Ecuador
[9]School of Earth Sciences, The University of Melbourne, Australia

*Correspondence to*: Christoph A. Keller (christoph.a.keller@nasa.gov)

**Abstract.** Social-distancing to combat the COVID-19 pandemic has led to widespread reductions in air pollutant emissions. Quantifying these changes requires a business-as-usual counterfactual that accounts for the synoptic and seasonal variability of air pollutants. We use a machine learning algorithm driven by information from the NASA GEOS-CF model to assess changes in nitrogen dioxide ($NO_2$) and ozone ($O_3$) at 5,756 observation sites in 46 countries from January through June 2020. Reductions in $NO_2$ coincide with timing and intensity of COVID-19 restrictions, ranging from 60% in severely affected cities (e.g., Wuhan, Milan) to little change (e.g., Rio de Janeiro, Taipei). On average, $NO_2$ concentrations were 18 (13-23)% lower than business as usual from February 2020 onward. China experienced the earliest and steepest decline, but concentrations since April have mostly recovered and remained within 5% to the business-as-usual estimate. $NO_2$ reductions in Europe and the US have been more gradual with a halting recovery starting in late March. We estimate that the global $NO_x$ ($NO+NO_2$) emission reduction during the first 6 months of 2020 amounted to 3.1 (2.6-3.6) TgN, equivalent to 5.5 (4.7-6.4)% of the annual anthropogenic total. The response of surface $O_3$ is complicated by competing influences of non-linear atmospheric chemistry. While surface $O_3$ increased by up to 50% in some locations, we find the overall net impact on daily average $O_3$ between February - June 2020 to be small. However, our analysis indicates a flattening of the $O_3$ diurnal cycle with an increase in night time ozone due to reduced titration and a decrease in daytime ozone, reflecting a reduction in photochemical production.

The $O_3$ response is dependent on season, time scale, and environment, with declines in surface $O_3$ forecasted if $NO_x$ emission reductions continue.

# 1 Introduction

The stay-at-home orders imposed in many countries during the Northern Hemisphere spring of 2020 to slow the spread of the severe acute respiratory syndrome coronavirus 2 (SARS-CoV-2, hereafter COVID-19), led to a sharp decline in human activities across the globe (Le Quéré et al., 2020). The associated decrease in industrial production, energy consumption, and transportation resulted in a reduction in the emissions of air pollutants, notably nitrogen oxides ($NO_x=NO+NO_2$) (Liu et al., 2020a; Dantas et al., 2020; Petetin et al., 2020; Tobias et al., 2020; Le et al., 2020). $NO_x$ has a short atmospheric lifetime and are predominantly emitted during the combustion of fossil fuel for industry, transport and domestic activities (Streets et al., 2013, Duncan et al., 2016). Atmospheric concentrations of nitrogen dioxide ($NO_2$) thus readily respond to local changes in $NO_x$ emissions (Lamsal et al., 2011). While this may provide both air quality and climate benefits, a quantitative assessment of the magnitude of these impacts is complicated by the natural variability of air pollution due to variations in synoptic conditions (weather), seasonal effects, and long-term emission trends as well as the non-linear responses between emissions and concentrations. Thus, simply comparing the concentration of pollutants during the COVID-19 period to those immediately before or to the same period in previous years is not sufficient to indicate causality. An emerging approach to address this problem is to develop machine-learning based 'weather-normalization' algorithms to establish the relationship between local meteorology and air pollutant surface concentrations (Grange et al., 2018; Grange and Carslaw, 2019; Petetin et al., 2020). By removing the meteorological influence, these studies have tried to better quantify emission changes as a result of a perturbation.

Here we adapt this weather-normalization approach to not only include meteorological information but also compositional information in the form of the concentrations and emissions of chemical constituents. Using a collection of surface observations of $NO_2$ and ozone ($O_3$) from across the world from 2018 to present (Section 2.1), we develop a 'bias-correction' methodology for the NASA global atmospheric composition model GEOS-CF (Section 2.2) which corrects the model output at each observational site based on the observations for 2018 and 2019 (Section 2.3). These biases reflect errors in emission estimates, sub-gridscale local influences (representational error), or meteorology and chemistry. Since the GEOS-CF model makes no adjustments to the anthropogenic emissions in 2020, and no 2020 observations are included in the training of the bias corrector, the bias-corrected model (hereafter BCM) predictions for 2020 represent a business-as-usual scenario at each observation site that can be compared against the actual observations. This allows the impact of COVID-19 containment measures on air quality to be explored, taking into account meteorology and the long-range transport of pollutants. We first apply this to the concentration of $NO_2$ (Section 3.1), and then $O_3$ (Section 3.2) and explore the differences between the counterfactual prediction and the observed concentrations. In Section 3.3 we explore how the observed changes in the $NO_2$ concentrations relate to emission of $NO_x$,

and in Section 3.4 we speculate what the COVID-19 restrictions might mean for the second half of
2020.

## 2 Methods

### 2.1 Observations

Our analysis builds on the recent development of unprecedented public access to air pollution model output and air quality observations in near real-time. We compile an air quality dataset of hourly surface observations for a total of 5,756 sites (4,778 for $NO_2$ and 4,463 for $O_3$) in 46 countries for the time period January 1, 2018 to July 1, 2020, as summarized in Fig. 1 and Table 1. More detailed maps of the spatial distribution of observation sites over China, Europe, and North America are given in Fig A1-3. The vast majority of the observations were obtained from the OpenAQ platform and the air quality data portal of the European Environment Agency (EEA). Both platforms provide harmonized air quality observations in near real-time, greatly facilitating the analysis of otherwise disparate data sources. For the EEA observations, we use the validated data (E1a) for years 2018-2019 and revert to the real-time data (E2a) for 2020. For Japan, we obtained hourly surface observations for a total of 225 sites in Hokkaido, Osaka, and Tokyo from the Atmospheric Environmental Regional Observation System (AEROS) (MOE, 2020). To improve data coverage in under-sampled regions, we further included observations from the cities of Rio de Janeiro (Brazil), Quito (Ecuador), and Melbourne (Australia). All cities offer continuous, hourly observations of $NO_2$ and $O_3$ over the full analysis period, thus offering an excellent snapshot of air quality at these locations. We include all sites with at least 365 days of observations between Jan 1, 2018 and December 31, 2019, and an overall data coverage of 75% or more since the first day of availability. Only days with at least 12 hours of valid data are included in the analysis. The final $NO_2$ and $O_3$ dataset comprise $8.9 \times 10^7$ and $8.2 \times 10^7$ hourly observations, respectively.

## Observation sites

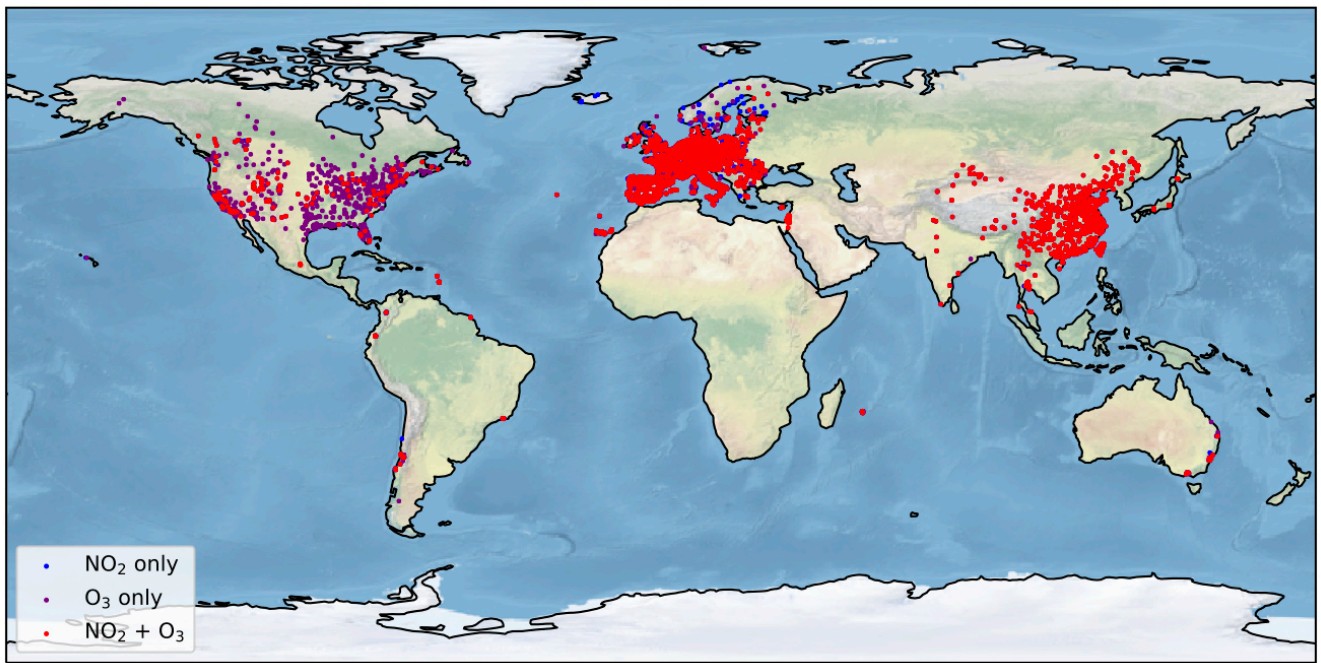

**Figure 1:** Location of the 5,756 observation sites included in the analysis. Red points indicate sites with both $NO_2$ and $O_3$ observations (3,485 in total), purple points show locations with $O_3$ observations only (978 sites) and blue points show locations with $NO_2$ observations only (1,293 sites). See Appendix for detailed maps for North America, Europe, and China.

100

**Table 1:** Observational data sources used in the analysis. Time period covers Jan 1, 2018 - July 1, 2020.

| Name | Countries | Sites | Source |
|---|---|---|---|
| OpenAQ | Australia, China, India, Hong Kong, Taiwan, Thailand, Canada, Chile, Colombia, United States | 2410 | https://openaq.org/ |
| EEA | Austria, Belgium, Bosnia and Herzegovina, Bulgaria, Croatia, Cyprus, Czech Republic, Denmark, Estonia, Finland, France, Germany, Greece, Hungary, Iceland, Ireland, Italy, Latvia, Lithuania, Luxembourg, Macedonia, Malta, Netherlands, Norway, Poland, Portugal, Romania, Serbia, Slovakia, Slovenia, Spain, Sweden, Switzerland, United Kingdom, | 3101 | https://discomap.eea.europa.eu/map/fme/AirQualityExport.htm |
| AEROS | Japan | 225 | http://soramame.taiki.go.jp/Index.php |
| EPA Victoria | Australia (Melbourne) | 4 | http://sciwebsvc.epa.vic.gov.au/aqapi/Help |
| Secretaria de Ambiente, Quito | Ecuador (Quito) | 8 | http://www.quitoambiente.gob.ec/ambiente/index.php/datos-horarios-historicos |

| Municipal Government of Rio de Janeiro | Brazil (Rio de Janeiro) | 8 | http://www.data.rio/datasets/dados-hor%C3%A1rios-do-monitoramento-da-qualidade-do-ar-monitorar |
|---|---|---|---|

## 2.2 Model

Meteorological and atmospheric chemistry information at each of the air quality observation sites is obtained from the NASA Goddard Earth Observing System Composition Forecast (GEOS-CF) model (Keller et al., 2020). GEOS-CF integrates the GEOS-Chem atmospheric chemistry model (v12-01) into the GEOS Earth System Model (Long et al., 2015; Hu et al., 2018) and provides global hourly analyses of atmospheric composition at 25x25 km$^2$ spatial resolution, available in near real-time at https://gmao.gsfc.nasa.gov/weather_prediction/GEOS-CF/data_access/ (Knowland et al., 2020). Anthropogenic emissions are prescribed using monthly Hemispheric Transport of Air Pollution (HTAP) bottom-up emissions (Janssens-Maenhout et al., 2015), with imposed weekly and diurnal scale factors as described in Keller et al. (2020). The same anthropogenic base emissions are used for years 2018-2020. Therefore, GEOS-CF does not account for any anthropogenic emission changes since 2018, notably any anthropogenic emission reductions related to COVID-19 restrictions. However, it does capture the variability in natural emissions such as wildfires (based on the Quick Fire Emissions Dataset, QFED) (Darmenov and Da Silva, 2015), or lightning and biogenic emissions (Keller et al., 2014). While the meteorology and stratospheric ozone in GEOS-CF are fully constrained by pre-computed analysis fields produced by other GEOS systems (Lucchesi, 2015; Wargan et al., 2015), no trace-gas observations are directly assimilated into the current version of GEOS-CF. It thus provides a "business as usual" estimate of $NO_2$ and $O_3$ that can be used as a baseline for input into the meteorological normalization process.

## 2.3 Machine learning bias correction

### 2.3.1 Overall strategy

We use the XGBoost machine learning algorithm (https://xgboost.readthedocs.io/en/latest/#) (Chen and Guestrin, 2016; Frery et al., 2017) to develop a machine learning model to predict the time-varying bias at each observation site at an hourly scale. XGBoost uses the Gradient Boosting framework to build an ensemble of decision trees, trained iteratively on the residual errors to stage-wise improve the model predictions (Friedman, 2001). Based on the 2018-2019 observation-model differences, the machine learning model is trained to predict the systematic (recurring) model bias between hourly observations and the co-located model predictions. These biases can be due to errors in the model, such as emission estimates, sub-gridscale local influences (representational error), or meteorology and chemistry. Since model biases are often site-specific, we train a separate machine learning model for each site. The design of the XGBoost framework is determined by a set of hyperparameters, such as the learning rate, maximum tree depth, or minimum loss reduction. While a full hyperparameter optimization across all sites - e.g., by using a grid search approach – would be computationally prohibitive, we conducted hyperparameter sensitivity tests at few selected sites and found that the XGBoost performance only

improved marginally at these sites when using other hyperparameter than the model defaults (less than 5% improvement). In addition, we found that the sites respond differently to the same change in hyperparameter setup, suggesting that there is no uniform hyperparameter design that is optimal across all sites. Based on this, we chose to use the default XGBoost model parameters at all locations, with a learning rate of 0.3, minimum loss reduction of 0, maximum tree depth of 6, and L1 and L2

regularization terms of 0 and 1, respectively.

For each location, we split the 2-year training dataset into 8 quarterly segments (Jan-Mar, Apr-Jun, etc.) and train the model 8 times, each time omitting one of the segments (8-fold cross validation). The omitted segment is used as test data to validate the general performance of the machine learning model and to provide an uncertainty estimate, as further discussed below. This approach aims to reduce the

auto-correlation signal that can lead to overly optimistic machine-learning results (Kleinert et al., 2020) while still including data from all four seasons in the testing. Once trained, the final model prediction at each location consists of the average prediction of the eight models.

The observations used in this analysis are not always quality-controlled, which can cause issues if erroneous observations are included in the training, such as unrealistically high $O_3$ concentrations of

several thousand ppbv. As an ad-hoc solution to this problem, we remove all observations below or above 2 standard deviations from the annual mean from the analysis. Sensitivity tests using more stringent thresholds of 3 or even 4 standard deviations resulted in no significant change in our results.

### 2.3.2 Evaluation of model predictors

The input variables fed into the XGBoost algorithm are provided in Table A1. The input features encompass 9 meteorological parameters (as simulated by the GEOS-CF model: surface north- and eastward wind components, surface temperature and skin temperature, surface relative humidity, total cloud coverage, total precipitation, surface pressure, and planetary boundary layer height), modelled surface concentrations of 51 chemical species ($O_3$, $NO_x$, carbon monoxide, VOCs, and aerosols), and 21

modelled emissions at the given location. In addition, we provide as input features the hour-of-day, day of week, and month of the year; these allow the machine learning model to identify systematic observation-model mismatches related to the diurnal, weekly and seasonal cycle of the pollutants. In addition, for sites with observations available for the full two years, we provide the calendar days since Jan 1, 2018 as an additional input feature to also correct for inter-annual trends in air pollution, e.g., due

to a steady decrease in emissions not captured by the model. This follows a similar technique to Ivatt and Evans (2020) and Petetin et al. (2020).

Gradient boosted tree models consist of a tree-like decision structure, which can be analysed to understand how the model uses the input features to make a prediction. Particularly useful in this context is the SHapely Additive exPlanations (SHAP) approach, which is based on game-theoretic

Shapely values and represents a measure of each feature's responsibility for a change in the model prediction (Lundberg et al., 2018). SHAP values are computed separately for each individual model prediction, offering detailed insight into the importance of each input feature to this prediction while also considering the role of feature interactions (Lundberg et al., 2020). In addition, combining the local SHAP values offers a representation of the global structure of the machine learning model.

Figure A4 shows the distribution of the SHAP values for all $NO_2$ predictors separated by polluted sites (left panel) and non-polluted sites (right panel), with polluted sites defined as locations with an annual average $NO_2$ concentration of more than 15 ppbv. Generally, the model-predicted (unbiased) $NO_2$ concentration is the most important predictor for the model bias, followed by the hour of the day, the day since Jan $1^{st}$ 2018 ('Trendday'), and a suite of meteorological variables including wind speed

(u10m, v10m), planetary boundary hight (zpbl), and specific humidity (q10m). All of these factors are expected to highly impact $NO_2$ concentrations and it is thus not surprising that the model biases are most sensitive to them. While there is considerable spread in the feature importance across the individual sites, there is little overall difference in the feature ranking between polluted vs. non-polluted sites.

Figure A5 shows the SHAP value distribution for all $O_3$ predictors, again separated into polluted and non-polluted sites (using the same definition as for the $NO_2$ sites). Unlike for $NO_2$, the bias-correction models for polluted sites exhibit different feature sensitivities than the non-polluted sites. At polluted locations, the availability of reactive nitrogen ($NO_2$, $NO_y$, PAN) is the dominant factor for explaining the model $O_3$ bias, reflecting the tight chemical coupling between $NO_x$ and $O_3$ (Seinfeld and Pandis,

2016). This is followed by the month of the year, total precipitation (tprec) and $O_3$ concentration, again variables expected to be correlated to $O_3$. At non-polluted sites, the uncorrected $O_3$ concentration is on average the most relevant input feature for the bias correctors, followed by the month of the year and the odd oxygen concentration (ox). The non-polluted sites are generally more sensitive to wind speed, reflecting the fact that $O_3$ production and loss at these locations is less dominated by local processes

compared to the polluted sites.

### 2.3.3 Machine learning model skill scores

    Figures 2 and 3 summarize the machine learning model statistics for $NO_2$ and $O_3$, respectively. The normalized mean bias (NMB), normalized root mean square error (NRMSE), and Pearson correlation

coefficient (R) at each site are shown for both the training (blue) and the test (red) dataset. We define NMB as mean bias normalized by average concentration at the given site, and the NRMSE as the root mean square error normalized by the range of the 95-percentile concentration and 5-percentile concentration. Rather than using the mean as the denominator for the NRMSE, we choose the percentile window as a better reference point for the concentration variability at a given site. Using the mean as

the denominator for the NRMSE would lead to very similar qualitative results.
    For both $NO_2$ and $O_3$, the bias-corrected model predictions show no bias when evaluated against the training data, NRMSE's of less than 0.3, and correlation coefficients between 0.6-1.0 ($NO_2$) and 0.75-1.0 ($O_3$). Compared to the training data, the skill scores on the test data show a higher variability, with an average NMB of -0.047 for $NO_2$ and -0.034 for $O_3$, a NRMSE of 0.25 ($NO_2$) and 0.18 ($O_3$), and a

correlation of 0.64 ($NO_2$) and 0.84 ($O_3$). We find no significant difference in skill scores between background vs. polluted sites or different countries.
    A number of factors likely contribute to the poorer statistical results at some of the sites. Importantly, some sites might be prone to overfitting if the training data includes events that are not easily generalizable, such as unusual emission activity (e.g., biomass burning, fireworks, closure of nearby

point source, etc.) or weather patterns not frequently observed. Also, the availability of test data at some locations is weak (less than 50%), which can contribute to a poorer skill score.

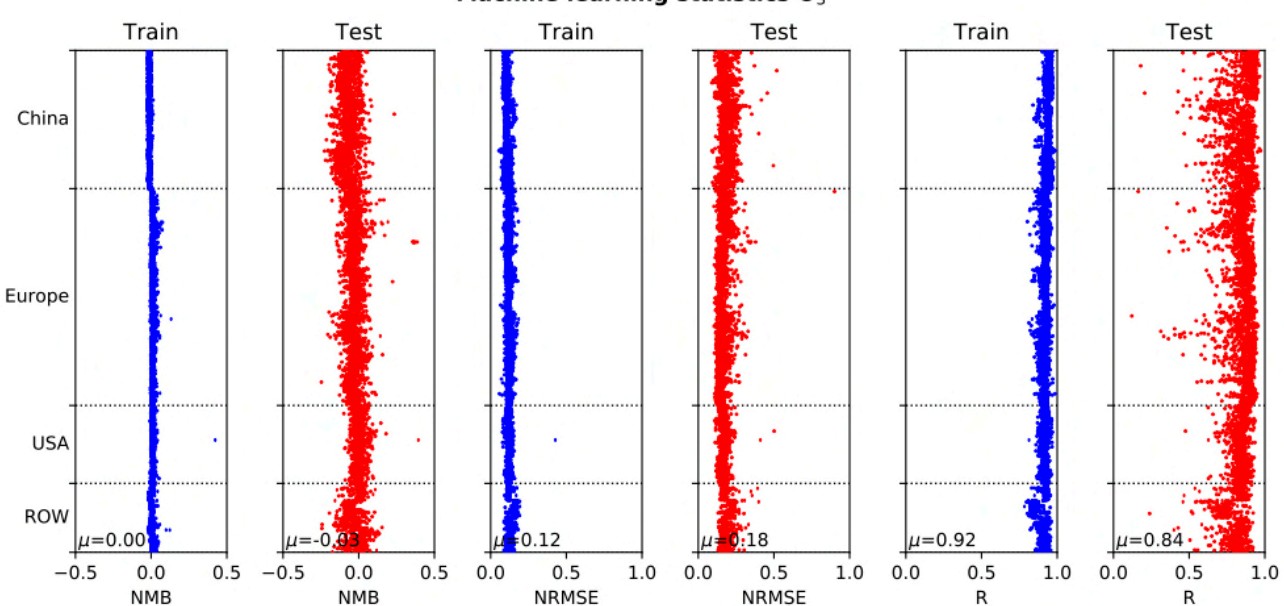

**Figure 2:** Machine learning statistics between hourly observations and the corresponding bias-corrected model predictions for each observation location. Shown are the normalized mean bias (NMB), normalized root mean square error (NRMSE) and Pearson correlation
coefficient (R) for the training data (blue) and the test data (red). Data sorted by region: China, Europe, United States (USA), and rest of the world (ROW). The mean values across all locations are shown in the figure inset.

**Figure 3:** As Figure 2 but for $O_3$.

### 2.3.4 Uncertainty estimation

To quantify the uncertainty of an individual model predictions at any given site, we use the standard deviation of the model-observation differences on the test data. For sites with 100% test data coverage, this represents the standard deviation from a sample of 17,520 hourly model-observation pairs. The thus obtained individual $NO_2$ prediction uncertainties range between 3.9 – 28 ppbv (mean = 8.5 ppbv) at polluted sites and 0.1 – 18 ppbv at clean sites (average of 4.9 ppbv). On a relative basis, this corresponds to an average uncertainty of 45% at polluted sites and 65% at clean sites. For $O_3$, we obtain an average individual prediction uncertainty of 14 ppbv (4.6 – 33 ppbv) at polluted sites and 9.0 ppbv (2.8 – 45 ppbv) at clean sites, corresponding to an average relative uncertainty of an individual prediction of 29% and 33% at polluted and clean sites, respectively.

The results presented in this paper are averages aggregated over multiple hours and locations, and the reported uncertainties are adjusted accordingly by calculating the mean uncertainty $\bar{\sigma}$ from the above-described hourly uncertainties $\sigma_i$:

$$\bar{\sigma}^2 = \sum_{i=1}^{N} \left(\frac{\sigma_i}{N}\right)^2$$

This assumes that the errors across individual sites are uncorrelated. The error covariance across sites is complex: two urban sites close to each other might show a low degree of error correlation due to local (street canyon etc) scale differences, whereas two background sites further apart might show significantly more correlation due to regional (synoptic) scale processes. In addition, our uncertainty calculation also implies that the aggregated mean error approaches zero. Given that the average mean biases of the machine learning models are clustered around zero (Fig. 2 and Fig. 3), this is a valid general assumption - especially when aggregating across multiple sites. For simplicity we keep the current analysis, but acknowledge that it might lead to overly optimistic uncertainty estimates for sites with a relatively large mean bias.

### 2.4 Lockdown dates

To support interpretation and guide visualizations, we include approximate national lockdown dates in all figures. The start and end dates for these are from https://en.wikipedia.org/wiki/COVID-19_pandemic_lockdowns (as of July 1, 2020) or based on local knowledge, with the full list of start and end dates given in Table A2. It should be noted that in many countries, lockdown policy varied regionally and many locations enacted 'soft' stay-at-home orders before the official lockdowns. Human behavior is therefore expected to have changed considerably in many locations before the official lockdowns went into force.

## 3 Results

### 3.1 Nitrogen dioxide

Figure 4 shows the weekly mean observations of $NO_2$ concentration, the GEOS-CF estimate and the BCM prediction based on the machine-learning predictor trained on 2018-2019 for the five cities of Wuhan (China), Taipei (Taiwan), Milan (Italy), New York (USA) and Rio de Janeiro (Brazil) from January 2018 through June 2020. We choose these five cities for illustration as they represent a diverse

level of socio-economic development and due to the cities' variable responses to the COVID-19 pandemic. These five cities are also illustrative of the varying quality of the uncorrected GEOS-CF predictions compared to the observations. For example, as shown by the dashed grey lines vs. the solid black lines in Fig. 4, the uncorrected model predictions are in good agreement with observations in Rio de Janeiro but underestimate the observed $NO_2$ concentrations in Taipei and Milan while overestimating

concentrations over New York. These differences are a combination of the observation-model scale mismatch (25x25 $km^2$ vs. point observation) and model errors, such as the simulated spatiotemporal distribution of $NO_x$ emissions or the modelling of the local boundary layer. The model-observation mismatch is particularly pronounced for Wuhan, where the model does not capture the observed seasonal cycle, pointing to errors in the imposed seasonal cycle of $NO_x$ emissions in the model.


In contrast to the uncorrected model predictions, the BCM closely follows the observations for years 2018 and 2019 (dashed black lines in Fig. 4). The grey region in Fig. 4 shows the start and end of the implementation of COVID-19 containment measures. Once containment is implemented, observed concentrations start to diverge from the BCM prediction for Wuhan, Milan and New York (Fig. 4). For

Wuhan, we find a reduction in $NO_2$ of 54 (48-59)% relative to the expected BCM value for February and March 2020, and average decreases of 30-40% are found over Milan (24-43%) and New York (20-34%) starting in mid-March and lasting through April (Fig. 4; Tables A3-A5). For cities where restrictions have been mainly removed (Wuhan, Milan) concentrations rise back towards the BCM value, although in neither city are the concentrations fully restored to what might be expected based on

the business-as-usual GEOS-CF simulation.

Looking more broadly at cities around the globe, 53 of the 64 specifically analysed cities feature $NO_2$ reductions of between 20-50% (Fig. A6-A8 and Tables A3-A5). Most locations issued social distancing recommendations prior to the legal lockdowns and observed $NO_2$ declines often precede the official

lockdown date by 7-14 days (e.g., Brussels, London, Boston, Phoenix, and Washington, DC).

For Taipei and Rio de Janeiro, the observations and the BCM show little difference (Fig. 4), consistent with the less stringent quarantine measures in these places. Other cities with only short-term $NO_2$ reductions of less than 25% include Atlanta (USA), Prague (Czech Republic), and Melbourne

(Australia), again fitting with the comparatively relaxed containment measures in these places (Fig. A6-A8). In contrast, Tokyo (Japan) and Stockholm (Sweden), which also implemented a less aggressive COVID-19 response, exhibit $NO_2$ reductions comparable to those of cities with official lockdowns (>20%), suggesting that economic and human activities were similarly subdued in those cities.

Substantial differences exist between cities in South America, with Rio de Janeiro and Santiago de Chile showing little change thus far in 2020, whereas Quito (Ecuador) and Medellin (Colombia) experienced a greater than 50% reduction in $NO_2$ after the initiation of strict restrictions measures in mid-March (Fig. A8 and Table A5). Concentrations in Medellin rebounded sharply in April and May, while concentrations in Quito remained 55 (52-58)% below business as usual throughout May and only
started to return back to normal in June.

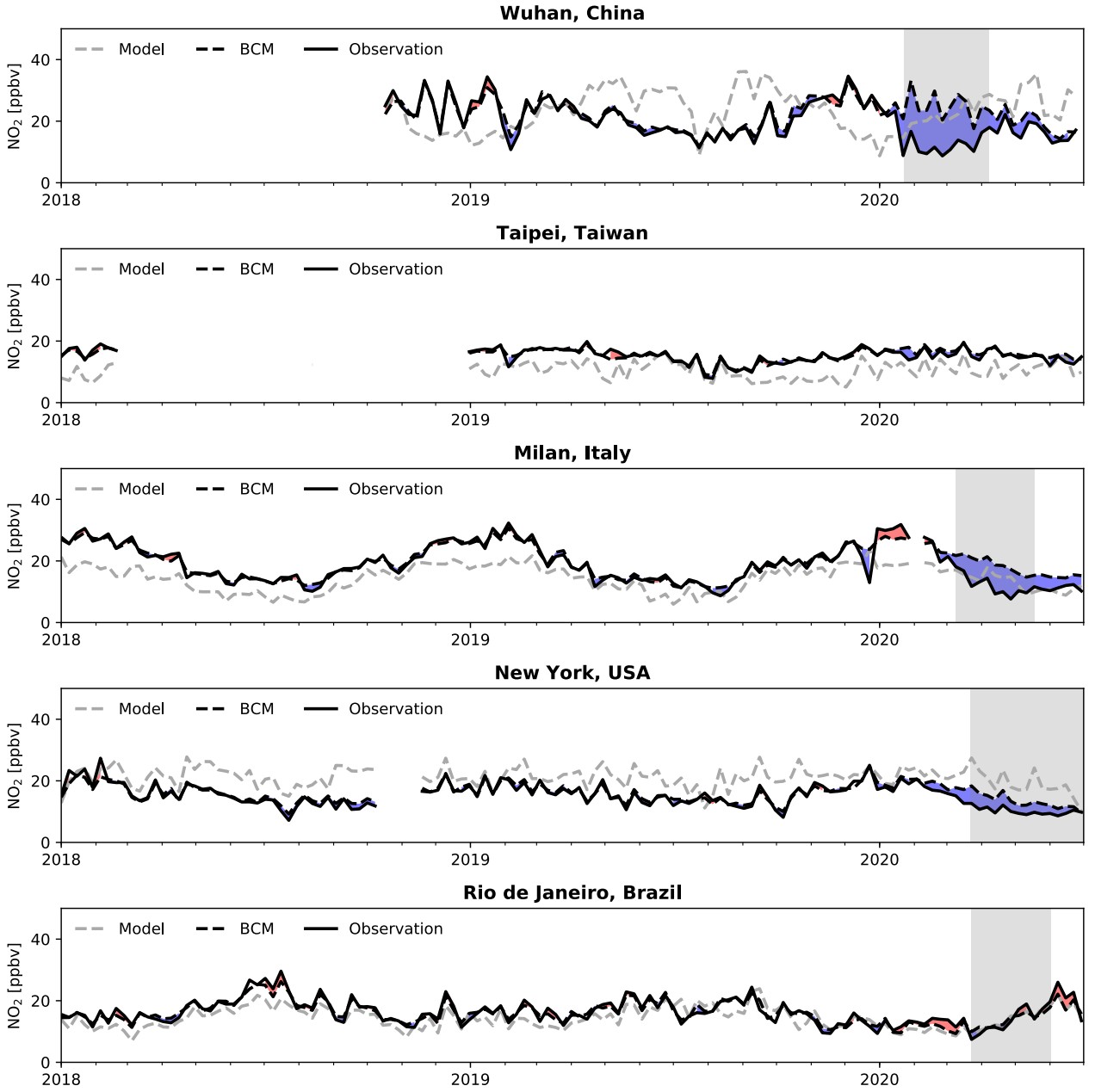

**Figure 4.** Comparison of NO$_2$ surface concentrations (ppbv = nmol mol-1) for Wuhan, Taipei, Milan, New York, and Rio de Janeiro for January 2018 through June 2020. Observed values are shown in solid black, the original GEOS-CF model simulation is shown in dashed grey, and the BCM predictions are in dashed black. The area between observations and BCM predictions is shaded blue (red) if observations are lower (higher) than BCM predictions. Grey areas represent the period of lockdown. Shown are the 7-day average mean values for the 9, 18, 19, 14 and 2 observational sites in Wuhan, Taipei, Milan, New York, and Rio de Janeiro, respectively. Observations for China are only available starting in mid-September 2018.


To evaluate the large-scale impact of COVID-19 restrictions on air quality, we aggregate the individual observation-model comparisons by country. We note that our estimates for some countries (e.g., Brazil, Colombia) are based on a single city and likely not representative of the whole country. On a country level, we find the sharpest and earliest drop in $NO_2$ over China, where observed concentrations fell, on average, 55 (51-59)% below their expected value in early February when restrictions were implemented

(Fig. 5). Concentrations remained at this level until late February, at which point they started to increase until restrictions were significantly relaxed in early April. Our analysis suggests that Chinese $NO_2$ concentrations have recovered to within 5 (1-9)% of the business as usual since then. For 2019 (dashed line in Fig. 5) the BCM shows a reduction in $NO_2$ concentrations around Chinese New Year (5th February 2019), and it is likely that some reduction around the equivalent 2020 period (25th January

2020) would have occurred anyway. However, the 2020 reductions are significantly larger and more prolonged than in 2019. Similar to China, India shows large reductions in $NO_2$ concentration (58 (49-67)%) coinciding with the implementation of restrictions in mid-March (Fig. 5); however, $NO_2$ concentrations have not yet recovered by the end of June, reflecting the prolonged duration of lockdown measures. Other areas of Asia, such as Hong Kong and Taipei, implemented smaller restrictions than

China or India and they show significantly smaller decreases (less than 20%).

For Europe and the United States, we find widespread $NO_2$ reductions averaging 22 (19-25)% in March and 33 (30-36)% in April (Fig. 5). In some countries, recovery is evident as lockdown restrictions are removed or lessened (e.g., Greece, Romania) but in 29 out of 36 countries, concentrations remain 20% or more below the business-as-usual scenario throughout May and June.


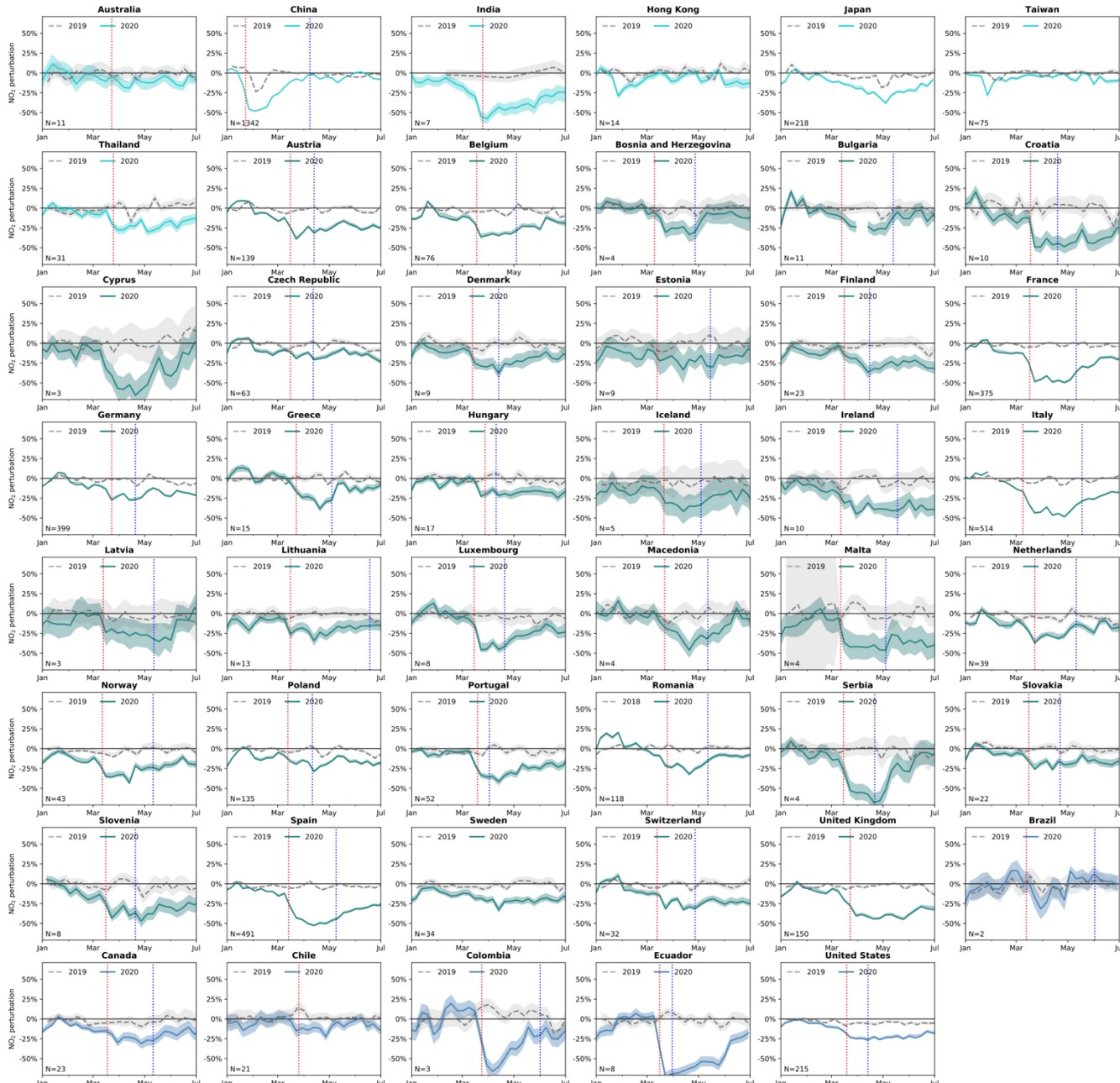

**Figure 5:** Seven-day average fractional difference between observed NO₂ and the BCM predictions for 46 countries between January 1 through June 30, aggregated from all sites across each country (number of sites in the bottom left of each panel). The thick line indicates the mean across all sites for the first half of 2020, with the shaded area representing the uncertainty estimate. Differing colours indicate differing regions (cyan: Asia & Australia; green: Europe; blue: Americas). The grey dashed line indicates the equivalent average for the same six month period in 2019 (note that 2019 data was included in the training). The red dashed vertical line indicates COVID-19 restriction dates, and the blue line indicates the beginning of easing measures.


## 3.2 Ozone

We follow the same methods for developing a business-as-usual counterfactual for $O_3$ as we did for $NO_2$
in section 3.1. Any change in local $O_3$ concentration arising from COVID-19 restrictions is set against a
large seasonal increase in (background) concentrations in the Northern Hemisphere springtime (Fig. 6).
Due to the longer atmospheric lifetime of $O_3$ compared to $NO_2$, the local $O_3$ signal is expected to be
comparatively small. This makes attributing changes in $O_3$ concentration more challenging than for
$NO_2$. Our analysis shows an $O_3$ increase of up to 50% for some periods in cities with large $NO_2$
reductions (e.g., Wuhan, Milan, Quito; Fig. 3 and Fig. A9-A11), but there is much less convincing
evidence for a systematic $O_3$ response across cities or on a regional level (Fig. 7). For example, our
analysis shows little $O_3$ difference in Beijing and Madrid during lockdown despite $NO_2$ declines
comparable to Wuhan or Milan (Fig. A9-A11). $O_3$ enhancements of up to 20% are found over Europe
(e.g., Belgium, Luxembourg, Serbia), with a peak in early April, approximately two weeks after
lockdown started (Fig. 7).

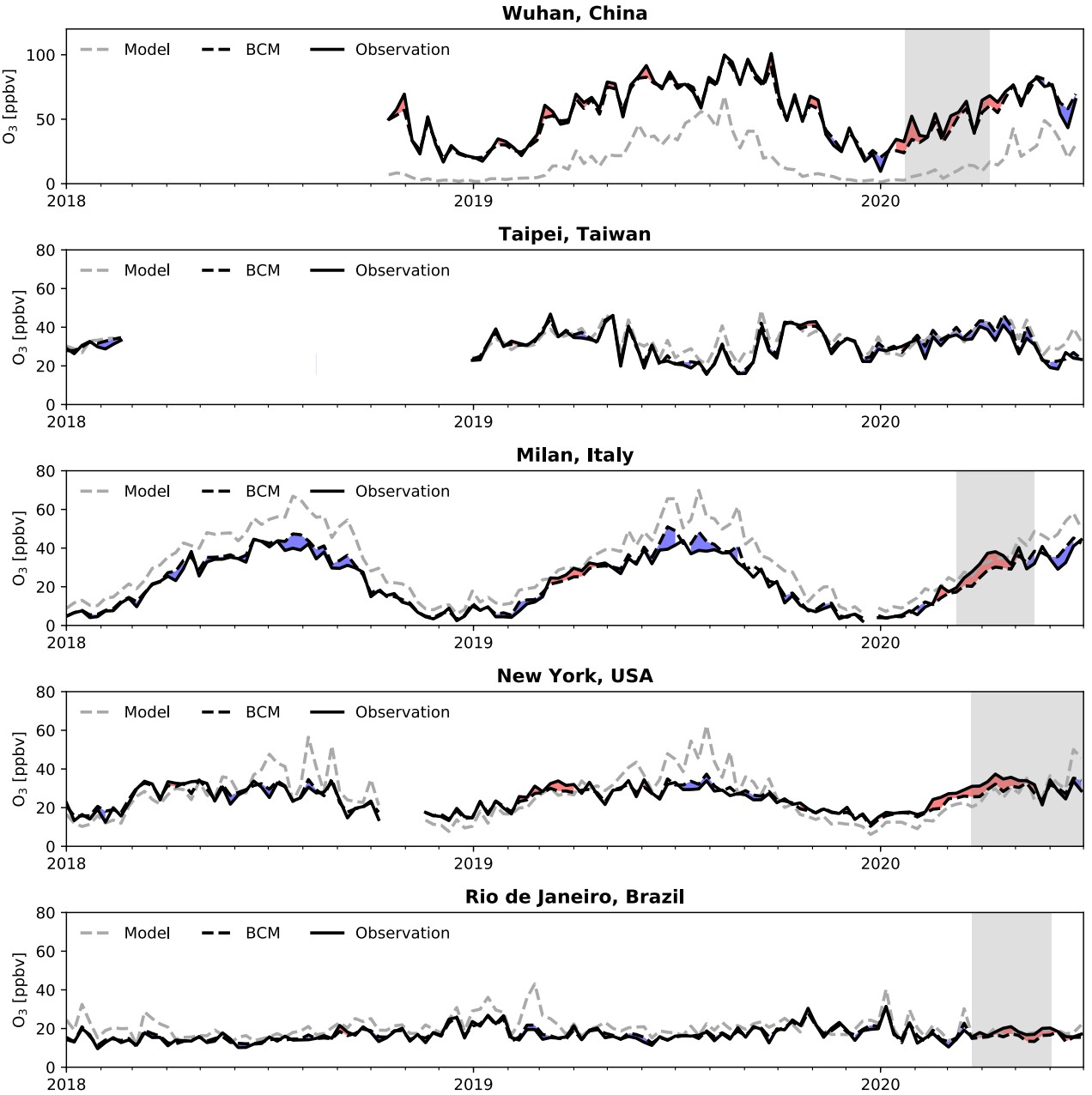

**Figure 6:** Comparison of O$_3$ surface concentrations for Wuhan, Taipei, Milan, New York, and Rio de Janeiro for January 2018 through June 2020. Observed values are shown in solid black, the original GEOS-CF model simulation is shown in dashed grey, and the BCM predictions are in dashed black. The area between observations and BCM predictions is shaded blue (red) if observations are lower (higher) than BCM predictions. The grey areas represent the period of lockdown. Shown are the 7-day average mean values for the 9, 18, 19, 14 and 4 observational sites in Wuhan, Taipei, Milan, New York, and Rio de Janeiro, respectively. Observations for China are only available starting in mid-September 2018.


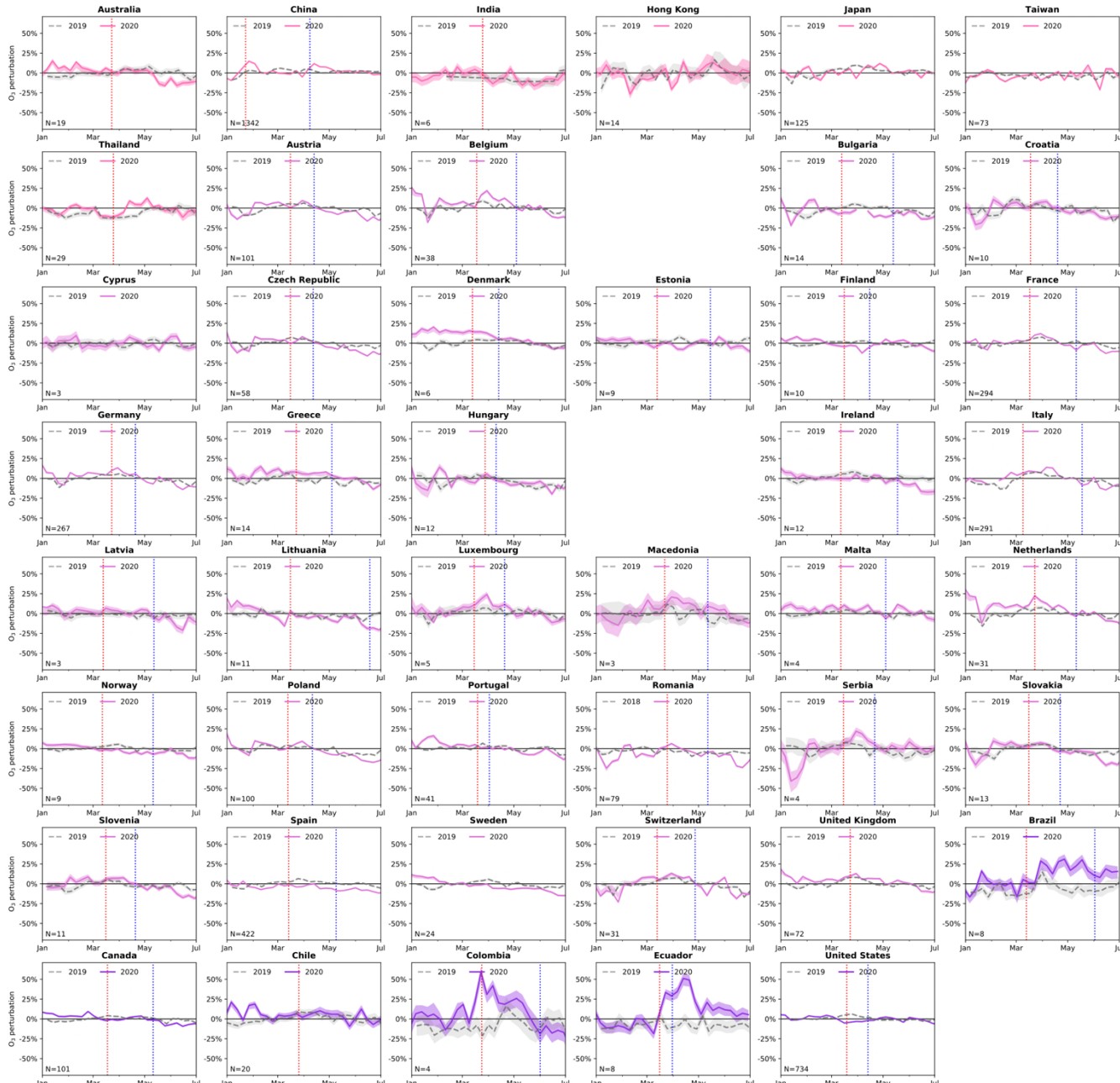

**Figure 7:** Similar to Figure 5 but for $O_3$ and without Bosnia and Herzegovina and Iceland. Differing colors indicate differing regions (pink: Asia & Australia; light purple: Europe; dark purple: Americas).

The analysis of $O_3$ is complicated by its nonlinear chemical response to $NO_x$ emissions. In the presence of sunlight, $O_3$ is produced chemically from the oxidation of volatile organic compounds in the presence of $NO_x$ (Seinfeld and Pandis, 2016). Therefore, a decline in $NO_x$ emissions could decrease $O_3$ production and thus suppress $O_3$ concentrations. On the other hand, the process of $NO_x$ titration, in

which freshly emitted NO rapidly reacts with $O_3$ to form $NO_2$, acts as a sink for $O_3$ (Seinfeld and Pandis, 2016). Odd oxygen ($O_x = NO_2 + O_3$) is conserved when $O_3$ reacts with NO and thus offers a tool for separating these competing processes. Figure 8 presents the global mean diurnal cycle for $O_3$ and $O_x$ for the 5-month period since February 1, 2020 for both the observations and the BCM model, based on the individual hourly predictions at each observation site aggregated by local hour. The analysis of $O_3$ and $O_x$ is based on the same set of observation sites where both $NO_2$ and $O_3$ observation are available (see Fig. 1). Compared to the BCM model, there has been an increase in the concentration of night time $O_3$ (midnight-5.00 local time, Fig. 8a) by 1 part per billion by volume (ppbv = nmol mol$^{-1}$) compared to the BCM, whereas $O_x$ shows a decrease of 1 ppbv (Fig. 8b). While these changes are small in magnitude, they represent a multi-month aggregate over 3,485 observation sites that are statistically significant at the 1% confidence interval. It should be noted that the biases of the machine learning models show little diurnal variability (Fig. A12-13), suggesting that this result is not caused by poor model performance during specific times of the day.

Our results indicate that during the night, reduced NO emissions led to a reduction in $O_3$ titration, allowing $O_3$ concentrations to increase. During the afternoon, we find that $O_3$ concentrations are lower by 1 ppbv (Fig. 8a), while observed $O_x$ concentrations are lower than the baseline model by almost 2 ppbv at 14:00 local time (Fig. 8b). We attribute the lower $O_x$ to reduced net $O_x$ production due to the lower $NO_x$ concentration, but as titration is also reduced, daytime $O_3$ concentrations are little changed. Overall changes to mean $O_3$ concentrations are small, but there is a flattening of the diurnal cycle.

As shown in the lower panels in Fig. 8, both factors - enhanced night time $O_3$ and reduced daytime $O_x$ - are more pronounced at locations where pre-existing $NO_2$ concentrations are high (> 15 ppbv). This suggests that the observed $O_3$ deviations from the BCM are indeed coupled to $NO_x$ reductions due to COVID-19 restrictions, given that those are most pronounced at polluted sites.

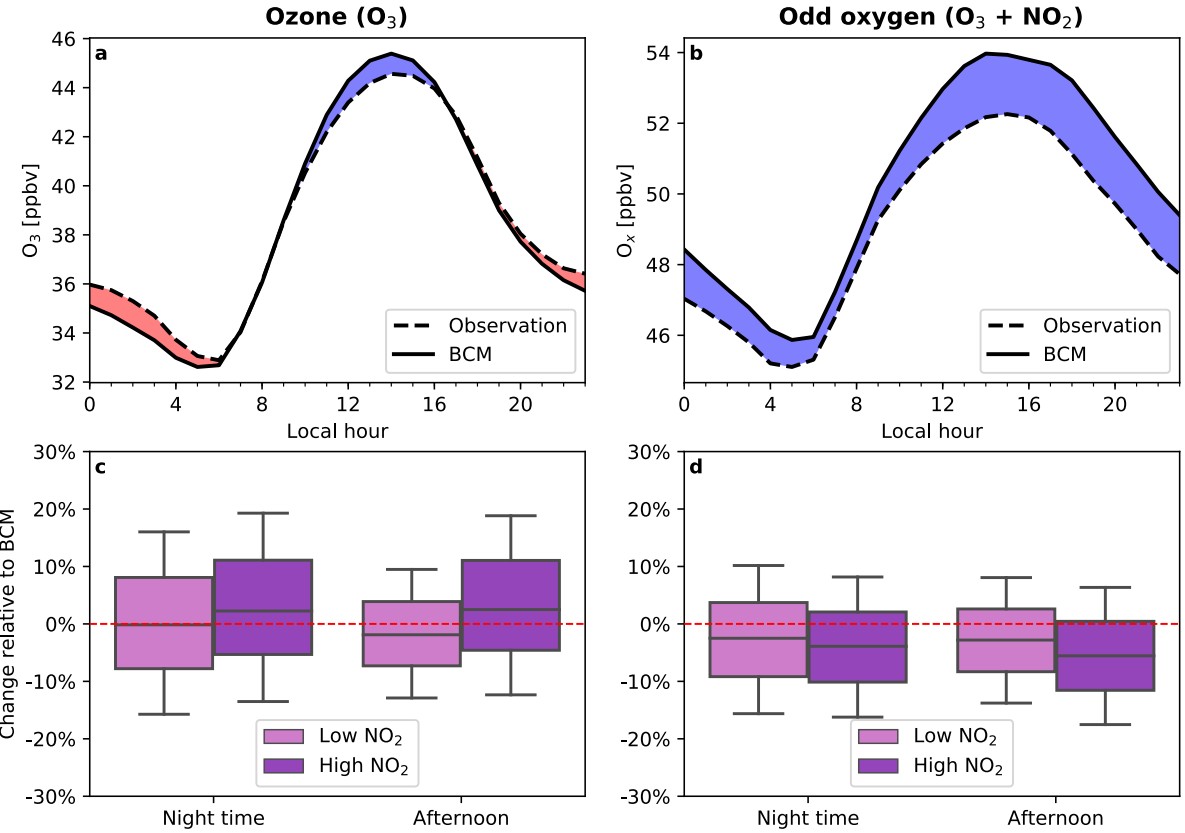

**Figure 8:** Observed and BCM modelled diurnal cycle of $O_3$ (a) and $O_x$ (b) averaged across all surface observation sites between February 1, 2020 through June 30, 2020 with estimated corresponding changes in surface $O_3$ (c) and $O_x$ (d) relative to the BCM. Barplots (c and d) show observed changes during night time (0-5 local time) and the afternoon (12-17 local time) for locations with low (< 15 ppbv) and high (> 15 ppbv) $NO_2$ concentrations (based on the 2019 average).

## 3.3 $NO_x$ emission reductions

The $NO_2$ analysis presented in Section 3.1 implies a stark reduction in $NO_x$ emissions. However, due to the impact of atmospheric chemistry, changes in $NO_2$ concentrations do not reflect the same relative change in $NO_x$ emissions. Because of this, the $NO_2/NO_x$ ratio and the $NO_x$ lifetime, both of which depend on seasonality and the local chemical environment, need to be taken into account when inferring $NO_x$ emissions from $NO_2$ concentrations (Lamsal et al., 2011; Shah et al., 2020). To estimate the relationship between changes in $NO_x$ emission and changes in $NO_2$ concentrations, we conducted a sensitivity simulation for the time period December 1, 2019 to June 8, 2020 using the GEOS-CF model with perturbed anthropogenic emissions. The perturbation simulation uses anthropogenic $NO_x$ emissions scaled based on adjustment factors derived from $NO_2$ tropospheric columns observed by the NASA OMI instrument (Boersma et al., 2011). Daily scale factors were computed by normalizing coarse-resolution (2x2.5 degrees), 14-day $NO_2$ tropospheric column moving averages by the corresponding moving average for year 2018 (the emissions base year in GEOS-CF; section 2.2). Forest fire signals were filtered out based on QFED emissions and no scaling was applied over water. This

results in anthropogenic emission adjustment factors of 0.3 to 1.4 (Fig. A14), comparable to the
magnitude obtained from the observation-BCM comparisons at cities globally (Fig. 5) and capturing the range of expected $NO_x$ emission changes. However, it should be noted that the scale factors do not necessarily coincide in space and time with the ones derived from observations and the BCM, and they do not include any adjustment for the $NO_2/NO_x$ ratio.

Figure 9a shows the response of $NO_2$ surface concentration to a change in $NO_x$ emissions, derived from the comparison of the sensitivity experiment against the GEOS-CF reference simulation. Our results indicate that $NO_2$ concentrations drop, on average, by 80% of the fractional decrease in anthropogenic $NO_x$ emission, with a further diminishing effect for emission reductions greater than 50%. This reflects both the buffering effect of atmospheric chemistry and the presence of natural background $NO_2$. The
here derived average sensitivity of 0.8 between a change in surface $NO_2$ to a change in $NO_x$ emissions is comparable to the value of 0.86 (1/1.16) obtained by Lamsal et al. (2011) for the relationship between $NO_x$ emissions and tropospheric column $NO_2$ observations.

To infer the reduction in anthropogenic $NO_x$ emissions due to COVID-19 containment measures during
the first six months of 2020, we use the best linear fit between the simulated $NO_x/NO_2$ sensitivity (dashed purple line in Fig. 9a). To do so, we calculate the monthly percentage emission change at each observation site based on the $NO_2$ anomalies derived in Section 3.1 and the corresponding best fit $NO_x/NO_2$ sensitivity (Fig. 9a). This is a simplification as the local $NO_x/NO_2$ sensitivity ratio is highly dependent on the local environment. To account for this uncertainty, we assign an absolute error of 15%
to our $NO_x/NO_2$ sensitivity, as derived from the spread in the $NO_x/NO_2$ ratio in the sensitivity simulation (Fig. 9a). We then aggregate these estimates to a country-level by weighting them based on average $NO_2$ concentrations per location, thus giving higher weight to locations with more nearby $NO_x$ emission sources. It should be noted that for some countries, our estimates are based upon a small number of observation sites that might not be representative for the country as a whole. This is
particularly true for India and Brazil, where less than 10 observation sites are available. While the smaller observation sample size is reflected in the wider uncertainty associated with these emission estimates compared to countries with a much denser monitoring network (e.g., China or Europe), the applied extrapolation method might incur errors that are not reflected in the stated uncertainty ranges. To obtain absolute estimates in emission changes, the monthly country-level percentage emission
changes are convoluted with bottom-up emissions estimates for 2015 from the Emission Database for Global Atmospheric Research (EDGAR v5.0_AP, Crippa et al., 2018, 2020). The choice of EDGAR v5.0 as the bottom-up reference inventory (over e.g., the HTAP emissions inventory used in GEOS-CF) was motivated by the fact that its baseline has been updated more recently and the country emission totals - which our analysis is based on - are readily available.
As summarized in Table 2, we calculate that the total reduction in anthropogenic $NO_x$ emissions due to COVID-19 containment measures during the first six months of 2020 amounted to 3.1 (2.6-3.6) TgN (Fig. 9b and Table 2). This is equivalent to 5.5 (4.7-6.4)% of global annual anthropogenic $NO_x$ emissions (Table 2). Our estimate encompasses 46 countries that together account for 67% of the total
emissions (excluding international shipping and aviation). We have no information for significant

countries such as Russia, Indonesia, or anywhere in Africa due to the lack of publicly available near real-time air quality information. China accounts for the largest fraction of the total deduced emission reductions (28%), followed by India (25%), the United States (18%), and Europe (12%).

While our method does not allow for sector-specific emission attribution, we assume our results to be most representative for changes in traffic emissions (rather than, say, aircraft emissions) given the location of the observation sites. On average, traffic emissions represent 27% of total anthropogenic $NO_x$ emissions (Crippa et al., 2018), and our derived total $NO_x$ emission reduction from Jan-Jun 2020 corresponds to 21 (17-24)% of global annual traffic emissions. The share of transportation on total $NO_x$ emissions is higher in the US and Europe (approx.. 40%) compared to India and China (20-25%). Taking this into account, the derived ratio of $NO_x$ emission reductions to annual traffic emissions is 21 (16-26)% in the US, 25 (20-30)% in Europe, 39 (34-44)% in China, and 62 (55-69)% in India.

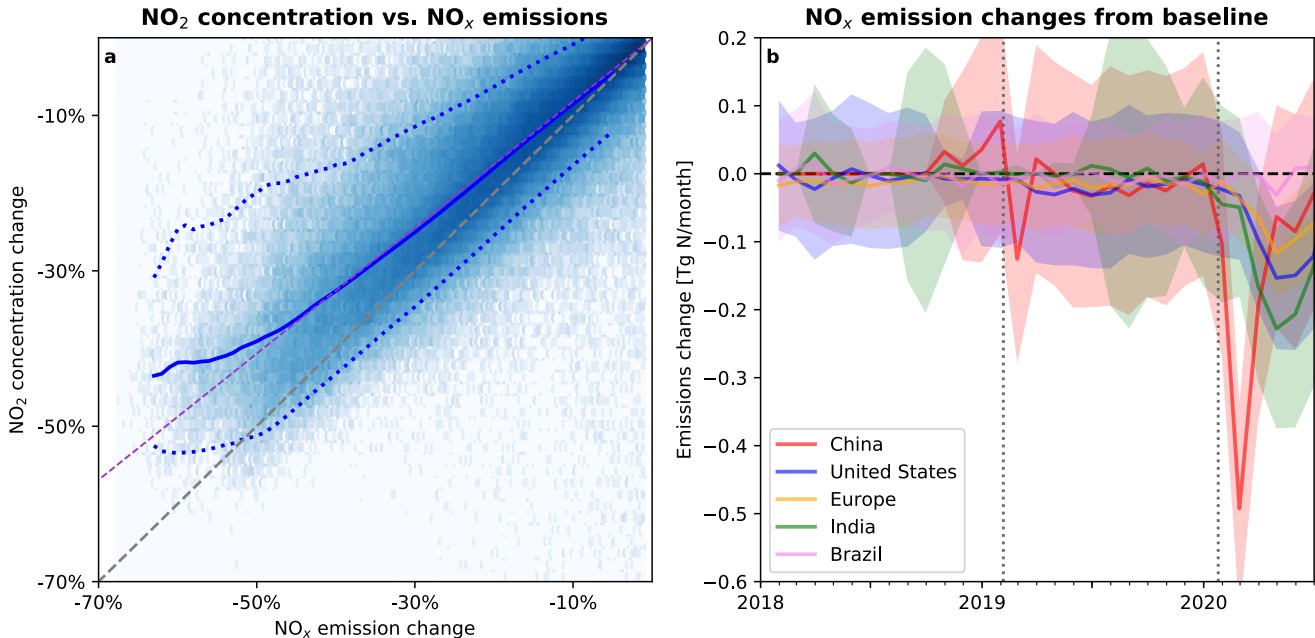

**Figure 9:** a) Response of $NO_2$ surface concentration (y-axis) to a change in $NO_x$ emissions (x-axis), as deduced from a model sensitivity simulation (see methods). The solid blue line shows the mean value across all individual grid cells (blue squares) and the dotted blue lines show the 5% and 95% quantiles. The dashed purple line shows the best linear fit. b) Estimated monthly change in $NO_x$ emissions from the baseline since 2018 for China (red), United States (blue), Europe (yellow), India (green), and Brazil (purple), as estimated from observed $NO_2$ concentration anomalies. Shaded areas indicate estimated emission uncertainties. Dotted grey lines indicate Chinese New Year 2019 and 2020.

**Table 2:** Anthropogenic $NO_x$ emission reductions in GgN month[-1] as derived from $NO_2$ concentration changes.

| | Baseline[1] | Feb-20 | Mar-20 | Apr-20 | May-20 | Jun-20 |
|---|---|---|---|---|---|---|
| Australia | 621 | -2.9 (-13.8-8.0) | -3.7 (-14.4-6.9) | -8.5 (-18.5-1.6) | -6.1 (-16.1-3.8) | -7.9 (-17.6-1.8) |
| Austria | 73 | -0.5 (-1.4-0.5) | -1.7 (-2.7--0.7) | -2.0 (-3.0--1.0) | -1.6 (-2.6--0.6) | -1.8 (-2.8--0.8) |
| Belgium | 98 | -1.0 (-2.4-0.3) | -2.0 (-3.4--0.7) | -3.2 (-4.5--1.9) | -2.4 (-3.8--1.1) | -1.8 (-3.2--0.5) |
| Bosnia and Herzegovina | 32 | 0.05 (-0.47-0.57) | -0.43 (-0.96-0.11) | -0.90 (-1.45--0.35) | -0.39 (-0.99-0.21) | -0.28 (-0.92-0.35) |
| Brazil | 1844 | -1.3 (-35.7-33.2) | -1.5 (-37.0-34.0) | -32.0 (-67.2-3.2) | 7.2 (-26.1-40.6) | 10.3 (-21.7-42.3) |
| Bulgaria | 46 | -0.12 (-0.83-0.58) | -0.60 (-1.32-0.12) | -1.20 (-1.93--0.46) | -0.67 (-1.44-0.11) | -0.41 (-1.20-0.37) |
| Canada | 755 | -6.3 (-17.1-4.6) | -12.2 (-23.3--1.1) | -19.8 (-31.4--8.1) | -18.5 (-30.5--6.5) | -11.4 (-23.8-1.0) |
| Chile | 202 | -0.5 (-3.9-2.9) | -0.7 (-4.0-2.6) | -2.8 (-6.0-0.4) | -1.6 (-4.7-1.4) | -1.0 (-4.1-2.1) |

| | | | | | | |
|---|---|---|---|---|---|---|
| China | 11876 | -517 (-669--366) | -191 (-342--39) | -63 (-215-89) | -82 (-235-70) | -30 (-182-123) |
| Colombia | 207 | 1.2 (-2.5-4.9) | -0.2 (-3.8-3.4) | -12.0 (-15.5--8.5) | -5.5 (-9.1--1.9) | -4.4 (-8.0--0.7) |
| Croatia | 24 | -0.25 (-0.64-0.14) | -0.55 (-0.95--0.15) | -1.03 (-1.44--0.63) | -0.96 (-1.37--0.55) | -0.90 (-1.31--0.48) |
| Czech Republic | 108 | -1.0 (-2.5-0.4) | -1.3 (-2.8-0.2) | -1.8 (-3.2--0.3) | -1.3 (-2.8-0.2) | -1.6 (-3.1--0.1) |
| Denmark | 48 | -0.5 (-1.3-0.3) | -0.8 (-1.6--0.1) | -1.4 (-2.1--0.6) | -1.0 (-1.8--0.2) | -0.8 (-1.5--0.0) |
| Ecuador | 133 | 0.5 (-1.6-2.6) | -3.8 (-5.9--1.8) | -8.9 (-11.0--6.9) | -7.5 (-9.6--5.4) | -4.0 (-6.1--2.0) |
| Estonia | 13 | -0.20 (-0.44-0.05) | -0.16 (-0.41-0.10) | -0.28 (-0.54--0.02) | -0.29 (-0.54--0.03) | -0.20 (-0.45-0.04) |
| Finland | 77 | -1.1 (-2.3-0.1) | -0.8 (-2.0-0.4) | -2.3 (-3.6--1.1) | -2.0 (-3.3--0.8) | -2.0 (-3.3--0.8) |
| France | 337 | -3.2 (-7.6-1.2) | -9.1 (-13.5--4.7) | -15.7 (-20.1--11.3) | -12.7 (-17.1--8.2) | -6.9 (-11.3--2.4) |
| Germany | 494 | -3.0 (-9.4-3.4) | -7.1 (-13.5--0.7) | -11.5 (-17.9--5.1) | -8.3 (-14.7--1.9) | -9.2 (-15.7--2.8) |
| Greece | 101 | 0.1 (-1.4-1.5) | -0.5 (-1.9-1.0) | -2.9 (-4.4--1.5) | -1.5 (-3.0--0.0) | -1.3 (-2.8-0.1) |
| Hong Kong | 90 | -1.5 (-2.8--0.2) | -0.2 (-1.6-1.1) | -0.4 (-1.7-1.0) | -0.3 (-1.6-1.0) | -1.2 (-2.6-0.2) |
| Hungary | 55 | -0.3 (-1.1-0.5) | -0.4 (-1.2-0.4) | -1.0 (-1.8--0.2) | -1.0 (-1.8--0.1) | -1.0 (-1.9--0.2) |
| Iceland | 2 | -0.04 (-0.08-0.01) | -0.04 (-0.09--0.01) | -0.09 (-0.14--0.04) | -0.07 (-0.12--0.01) | -0.04 (-0.10-0.01) |
| India | 4693 | -52 (-125-21) | -161 (-234--88) | -232 (-307--157) | -202 (-280--125) | -140 (-220--59) |
| Ireland | 35 | -0.3 (-1.0-0.3) | -0.8 (-1.4--0.2) | -1.3 (-1.9--0.7) | -1.4 (-2.0--0.8) | -1.2 (-1.8--0.6) |
| Italy | 357 | -1.9 (-6.5-2.7) | -9.7 (-14.4--5.1) | -15.6 (-20.2--10.9) | -12.4 (-17.1--7.8) | -7.7 (-12.3--3.0) |
| Japan | 996 | -4.1 (-17.2-9.0) | -12.6 (-25.7-0.6) | -23.4 (-36.7--10.2) | -28.7 (-41.9--15.4) | -18.0 (-31.3--4.7) |
| Latvia | 14 | -0.08 (-0.38-0.22) | -0.16 (-0.45-0.12) | -0.37 (-0.67--0.06) | -0.44 (-0.74--0.13) | -0.18 (-0.48-0.12) |
| Luxembourg | 12 | -0.17 (-0.48-0.15) | -0.27 (-0.58-0.05) | -0.50 (-0.82--0.18) | -0.40 (-0.72--0.07) | -0.32 (-0.64-0.01) |
| Lithuania | 20 | 0.01 (-0.18-0.19) | -0.23 (-0.42--0.05) | -0.47 (-0.65--0.29) | -0.31 (-0.49--0.13) | -0.23 (-0.42--0.05) |
| Macedonia | 10 | -0.00 (-0.17-0.16) | -0.08 (-0.25-0.09) | -0.33 (-0.50--0.16) | -0.28 (-0.46--0.10) | -0.06 (-0.24-0.12) |
| Malta | 3 | -0.00 (-0.06-0.06) | -0.07 (-0.13--0.01) | -0.13 (-0.20--0.07) | -0.11 (-0.18--0.05) | -0.11 (-0.18--0.05) |
| Netherlands | 121 | -1.4 (-3.1-0.4) | -2.6 (-4.3--0.8) | -3.4 (-5.1--1.7) | -2.1 (-3.9--0.4) | -2.0 (-3.8--0.2) |
| Norway | 63 | -0.8 (-1.7-0.0) | -1.5 (-2.4--0.6) | -1.9 (-2.8--1.0) | -1.5 (-2.5--0.6) | -1.1 (-2.0--0.2) |
| Poland | 284 | -3.3 (-7.1-0.5) | -3.2 (-7.0-0.5) | -5.9 (-9.7--2.1) | -4.0 (-7.8--0.2) | -5.1 (-9.0--1.3) |
| Portugal | 70 | -0.4 (-1.4-0.6) | -1.2 (-2.2--0.2) | -2.6 (-3.6--1.5) | -1.9 (-2.9--0.9) | -1.6 (-2.6--0.5) |
| Romania | 102 | 0.5 (-0.9-1.8) | -1.1 (-2.5-0.3) | -2.5 (-3.9--1.2) | -1.6 (-3.0--0.2) | -1.0 (-2.4-0.4) |
| Serbia | 63 | -0.48 (-1.54-0.59) | -1.77 (-2.87--0.68) | -3.83 (-4.94--2.72) | -2.24 (-3.44--1.04) | -0.82 (-2.07-0.43) |
| Slovakia | 33 | -0.20 (-0.67-0.28) | -0.43 (-0.90-0.05) | -0.61 (-1.09--0.13) | -0.54 (-1.04--0.05) | -0.57 (-1.07--0.07) |
| Spain | 333 | -2.9 (-7.2-1.5) | -8.6 (-13.0--4.2) | -17.0 (-21.3--12.6) | -13.9 (-18.3--9.5) | -10.0 (-14.5--5.6) |
| Sweden | 85 | -1.0 (-2.2-0.2) | -1.3 (-2.5--0.0) | -2.0 (-3.3--0.8) | -1.9 (-3.1--0.6) | -1.6 (-2.9--0.4) |
| Switzerland | 36 | -0.25 (-0.77-0.26) | -0.65 (-1.16--0.14) | -0.94 (-1.45--0.43) | -0.83 (-1.36--0.30) | -0.84 (-1.37--0.31) |
| Taiwan | 371 | -3.7 (-8.6-1.2) | -1.5 (-6.4-3.5) | -1.3 (-6.2-3.7) | -1.7 (-6.7-3.4) | -3.9 (-8.9-1.2) |
| Thailand | 458 | -2.6 (-9.2-4.0) | -4.8 (-11.7-2.0) | -10.7 (-17.6--3.8) | -11.6 (-18.6--4.7) | -8.5 (-15.6--1.4) |
| United Kingdom | 390 | -3.0 (-8.3-2.2) | -6.8 (-12.0--1.6) | -16.4 (-21.6--11.2) | -16.4 (-21.6--11.1) | -12.8 (-18.0--7.5) |
| United States | 6243 | -34 (-116-48) | -94 (-177--11) | -155 (-239--72) | -147 (-231--64) | -123 (-207--40) |
| Other countries[2] | 1307 | n/a | n/a | n/a | n/a | n/a |
| Shipping and Aviation | 671 | n/a | n/a | n/a | n/a | n/a |
| **Total** | **4681** | **-651 (-843--460)** | **-553 (-745--360)** | **-692 (-885--498)** | **-603 (-798--408)** | **-418 (-615--222)** |

[1] EDGAR v5.0_AP 2015 annual emissions expressed as GgN month$^{-1}$ (Crippa et al., 2020)

[2] Primarily Indonesia, Iran, Mexico, Pakistan, Russia, Saudi Arabia, South Africa, South Korea, Vietnam

## 3.4 Long-term impact of reduced NO$_x$ emissions on surface O$_3$

The response of O$_3$ to NO$_2$ declines in the wake of the COVID-19 outbreak is complicated by the competing influences of atmospheric chemistry. From February through June 2020, the diurnal observation-BCM comparisons suggest that the reduction in photochemical production was offset by a smaller loss from titration, as described in Section 3.2. This resulted in a flattening of the diurnal cycle and an insignificant net change in surface O$_3$ over a diurnal cycle. The competing impacts of reduced

NO$_x$ emissions on O$_3$ production and loss are dependent on the local chemical and meteorological environment. This is reflected in the variable geographical response of O$_3$ following the implementation of COVID-19 restrictions (Le et al., 2020; Dantas et al., 2020). Moreover, as atmospheric reactivity

increases through the Northern Hemisphere spring and summer, the relative importance of photochemical production is expected to increase in the Northern Hemisphere.

To assess the potential seasonal-scale impact of reduced anthropogenic emissions on $O_3$, we conducted two free-running forecast simulations between June 8 through August 31, 2020, initialized from the GEOS-CF simulation and the sensitivity simulation described in Section 3.3, respectively. Both simulations use the same biomass burning emissions based on a historical QFED climatology. For the forecast sensitivity experiment, we assume a sustained, time-invariant 20% reduction in global

anthropogenic emissions of $NO_x$, carbon monoxide (CO), and VOCs. We chose to alter not only the anthropogenic emissions of $NO_x$ but also of other pollutants whose anthropogenic emissions are highly correlated to $NO_x$, as a reduction in $NO_x$ emissions without corresponding declines in CO and VOC emissions seems unrealistic.

Figure 10 shows the differences between the reference forecast and the sensitivity simulation over the

United States, Europe and China. Our results indicate that sustained lower anthropogenic emissions lead to a general decrease in surface $O_3$ concentrations of 10-20% over Eastern China, Europe, and the Western and Northeast US during July and August relative to the business-as-usual reference forecast simulation. However, it is also notable that in some locations the model forecast $O_3$ concentrations increase by an equivalent amount (e.g., Scandinavia, South Central US and Mexico, Northern India),

reflecting the high nonlinearity of atmospheric chemistry. This highlights the complex interactions between emissions, chemistry, and meteorology and their impact on air pollution on different time scales.

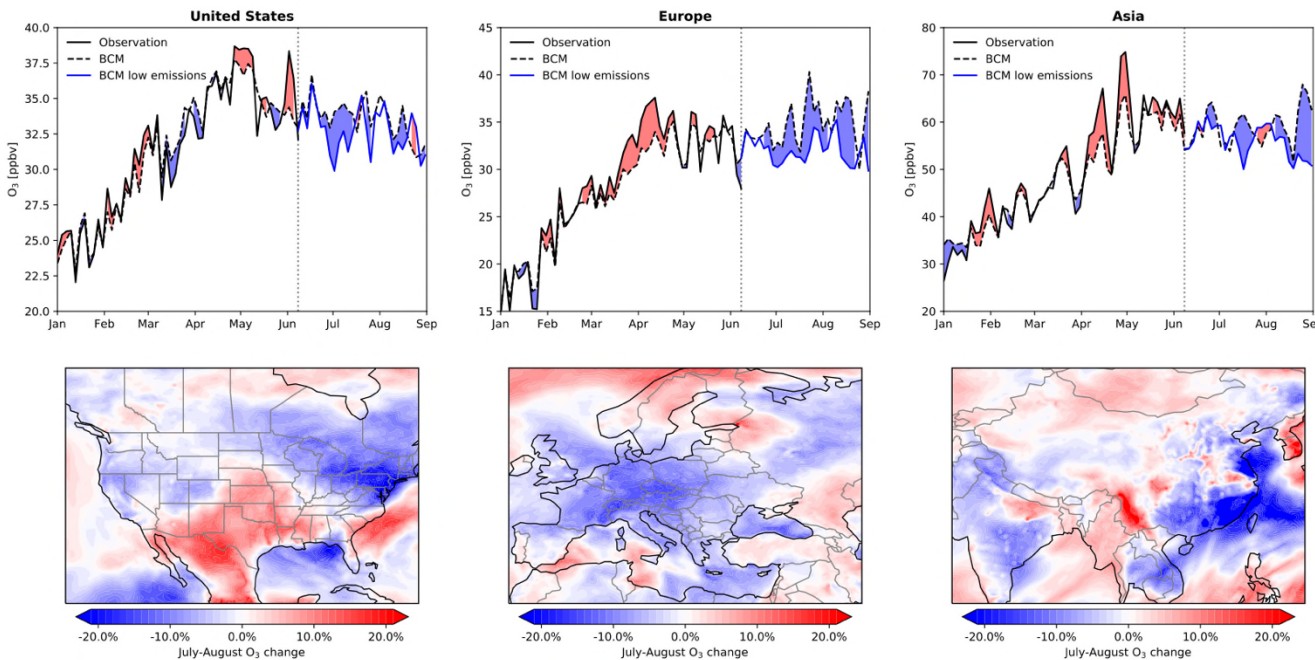

**Figure 10:** Change in mean surface $O_3$ over the United States, Europe, and Asia for a sensitivity simulation with altered anthropogenic
emissions. Top panels show daily average $O_3$ concentrations at all observation sites within the given region (solid black, Jan-Jun), the bias-corrected GEOS-CF model ("BCM", solid black, Jan 1st - Jun 8th) continued with a business-as-usual GEOS-CF forecast from Jun 9th - Aug 31st, and GEOS-CF forecast assuming sustained 20% anthropogenic emission reduction (blue). Bottom panels show mean changes in surface $O_3$ for July and August for the low emissions simulation relative to the business-as-usual forecast.

## 4 Conclusions

The combined interpretation of observations and model simulations using machine learning can be used to remove the compounding effect of meteorology and atmospheric chemistry, offering an effective tool to monitor and quantify changes in air pollution in near real-time. The global response to the COVID-19 pandemic presents a perfect testbed for this type of analysis, offering insights into the interconnectedness of human activity and air pollution. While national mitigation strategies have led to

substantial regional $NO_2$ concentration decreases over the past decades in many places (e.g., Hilboll et al., 2013; Russell et al., 2013; Castellanos and Boersma, 2012), the widespread and near-instantaneous reduction in $NO_2$ following the implementation of COVID-19 containment measures indicates that there is still large potential to lower human exposure to $NO_2$ through reduction of anthropogenic $NO_x$ emissions.


The here derived $NO_2$ reductions are in good agreement with other emerging estimates. For instance, we determine an 18% decline over China for the 20 days after Chinese New Year relative to the preceding 20 days, consistent with the 21% reduction reported in Liu et al. (2020a). Similarly, our estimated 22% reduction over China for January to March 2020 is in excellent agreement with the 21-23% reported by

Liu et al. (2020b). For Spain, we obtain an $NO_2$ reduction of 46% between March 14 to April 23, again in close agreement with the values reported in Petetin et al. (2020).

While large reductions in $NO_2$ concentrations are achievable and immediately follow curtailments in $NO_x$ emissions, the $O_3$ response is more complicated and can be in the opposite direction, at times by as

much as 50% (Jhun et al., 2015, Le et al., 2020). The $O_3$ response is dependent on season, time scale, and environment, with an overall tendency to lower surface $O_3$ under a scenario of sustained $NO_x$ emission reductions. This shows the complexities faced by policy makers in curbing $O_3$ pollution.

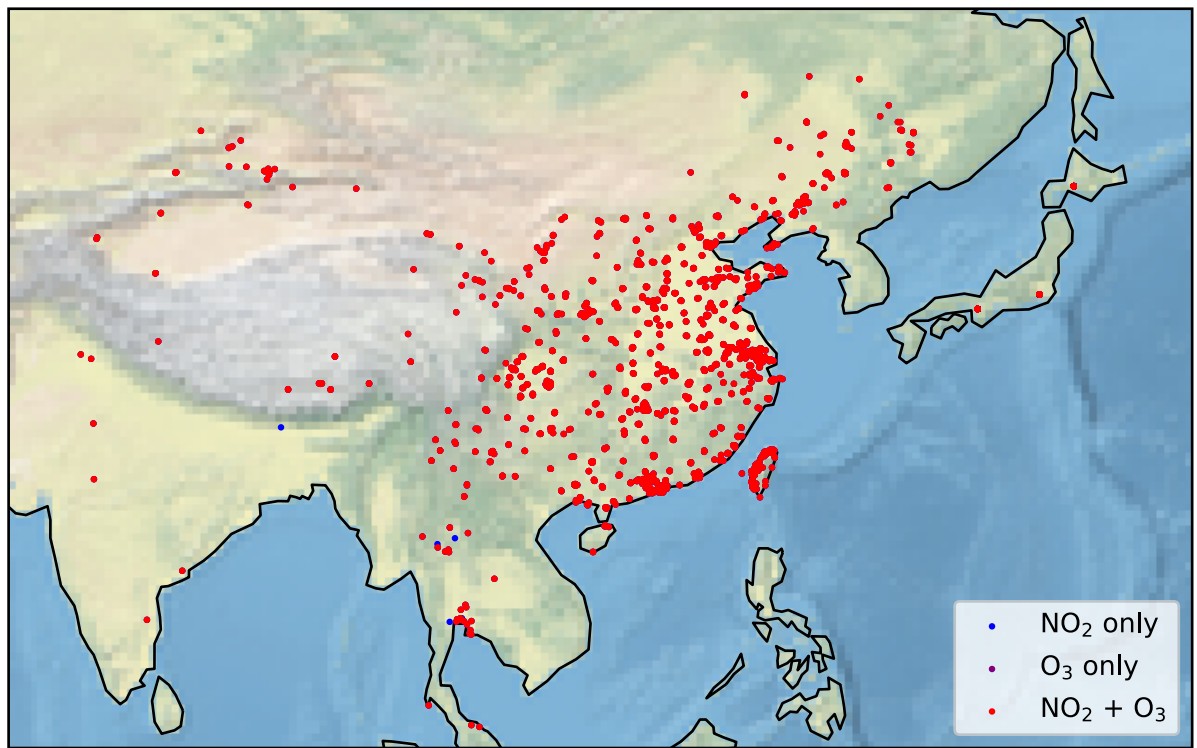

**Figure A1:** Close up of Chinese observation sites included in the analysis. Red points indicate sites with both NO2 and O3 observations, purple points show locations with O3 observations only and blue points show locations with NO2 observations only.

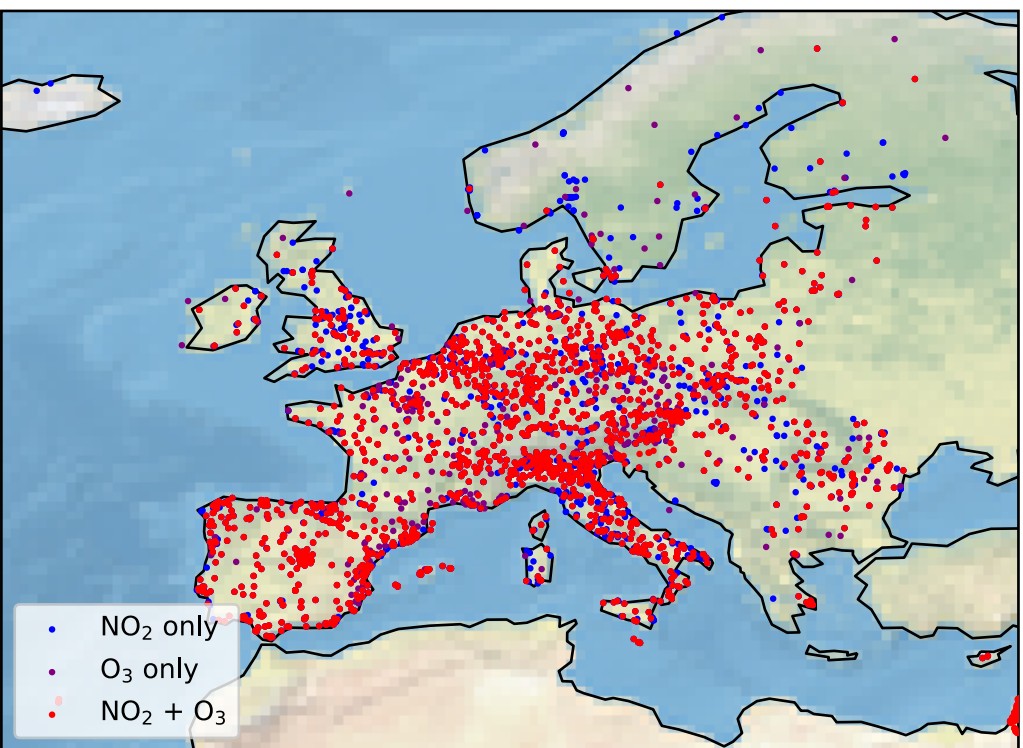


**Figure A2:** Close up of European observation sites included in the analysis. Red points indicate sites with both NO2 and O3 observations, purple points show locations with O3 observations only and blue points show locations with NO2 observations only.

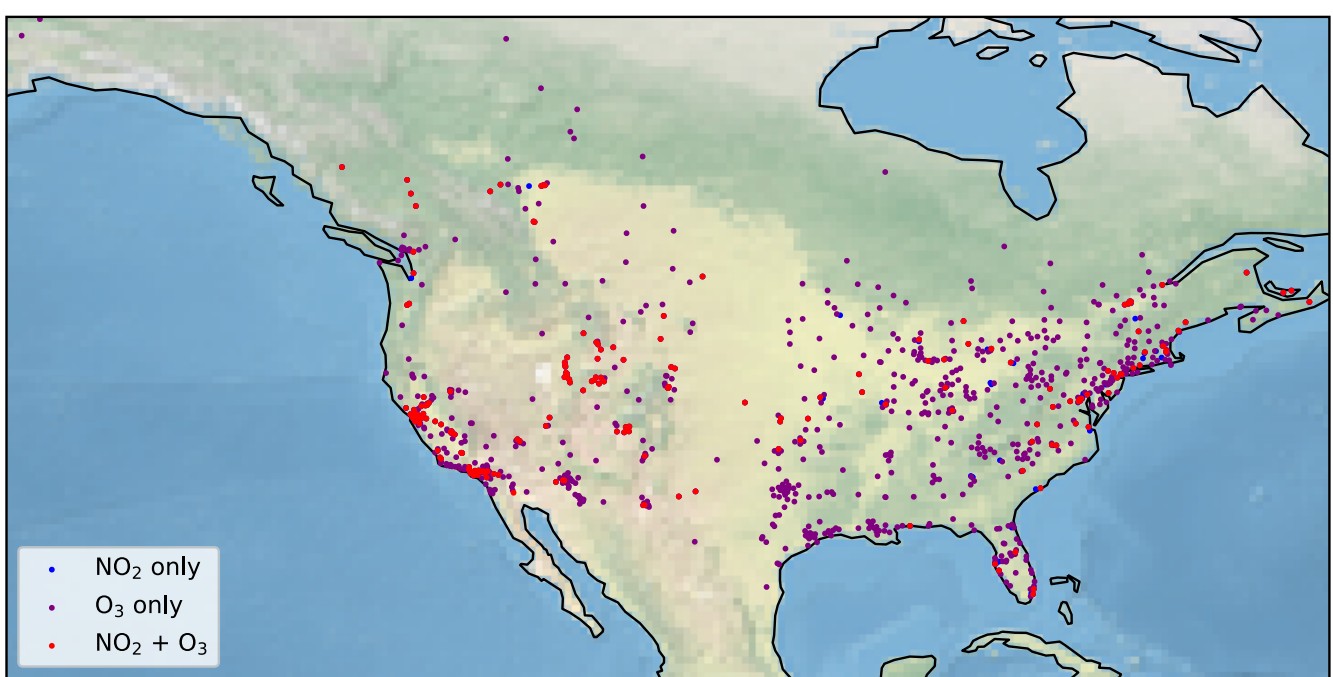

**Figure A3:** Close up of North American observation sites included in the analysis. Red points indicate sites with both NO2 and O3 observations, purple points show locations with O3 observations only and blue points show locations with NO2 observations only.

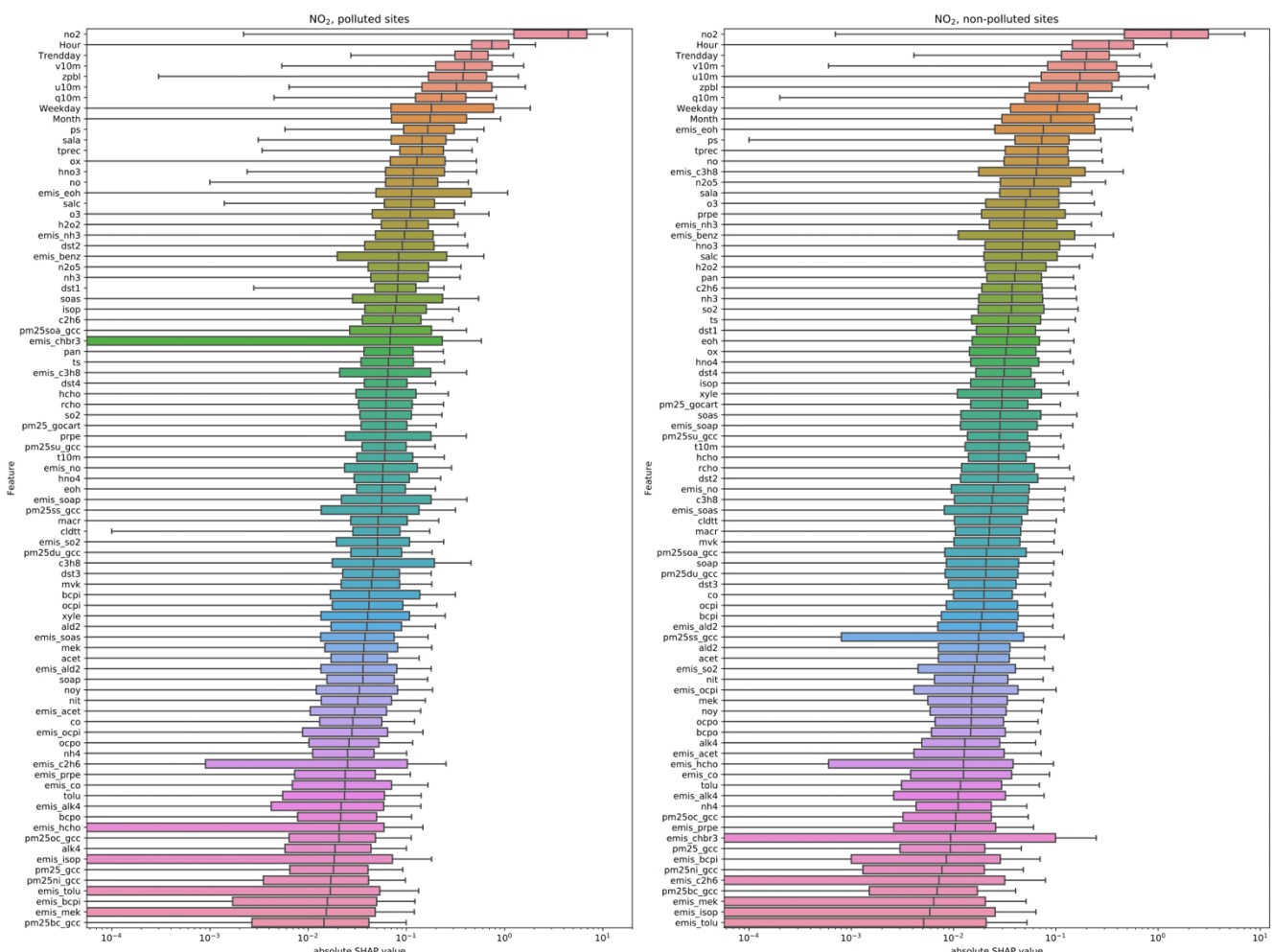

**Figure A4:** Importance of input variables (features) for the XGBoost models trained to predict NO₂ model bias. Shown are the distribution
of the absolute SHAP values for each feature, ranked by the average importance of each feature. Higher SHAP value indicates higher
feature importance. Left panel shows results for polluted sites (average annual NO₂ concentration > 15ppbv) and right panel shows results
for all other sites.

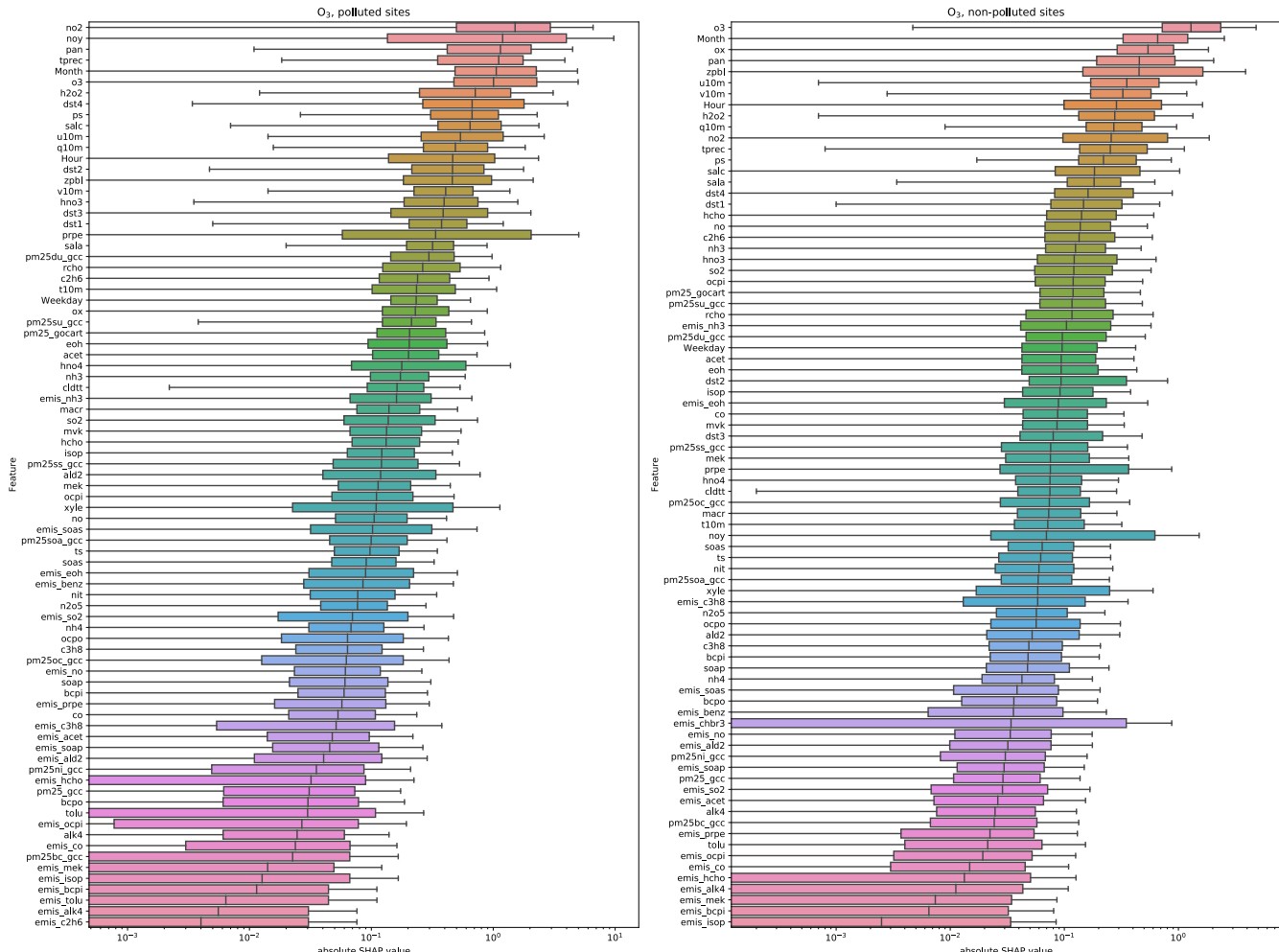


**Figure A5:** As Figure A4 but for O$_3$.

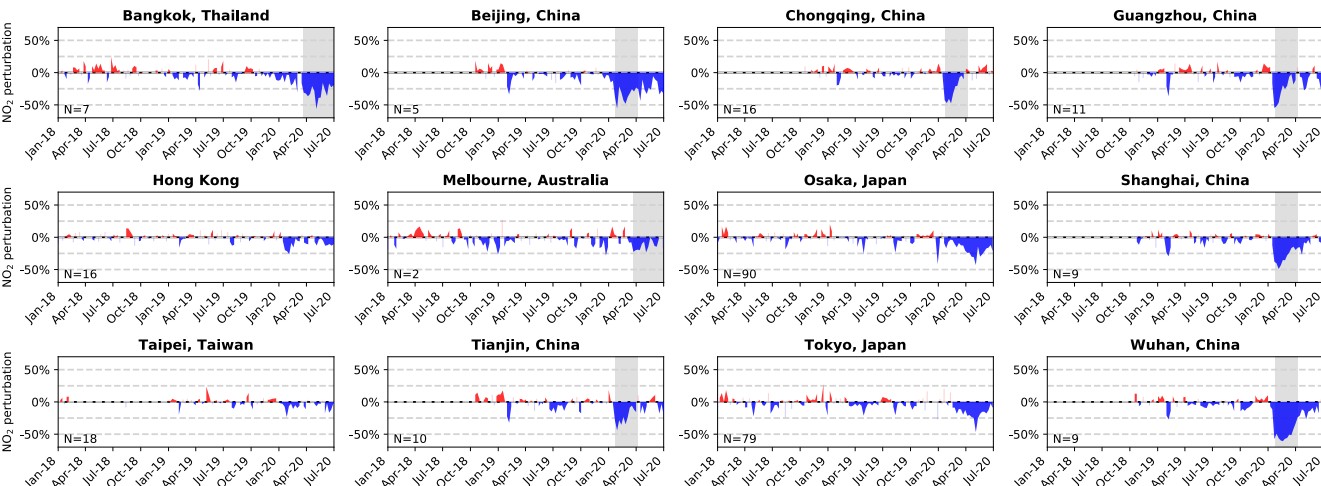

**Figure A6:** Normalized fractional NO$_2$ perturbations (observation - bias-corrected model, normalized by the bias-corrected model prediction) from Jan 1, 2018 through June 2020 for selected cities in Asia and Australia. Grey shaded areas indicate COVID-19 lockdown periods. Number of sites per city are shown in the bottom left of each panel.

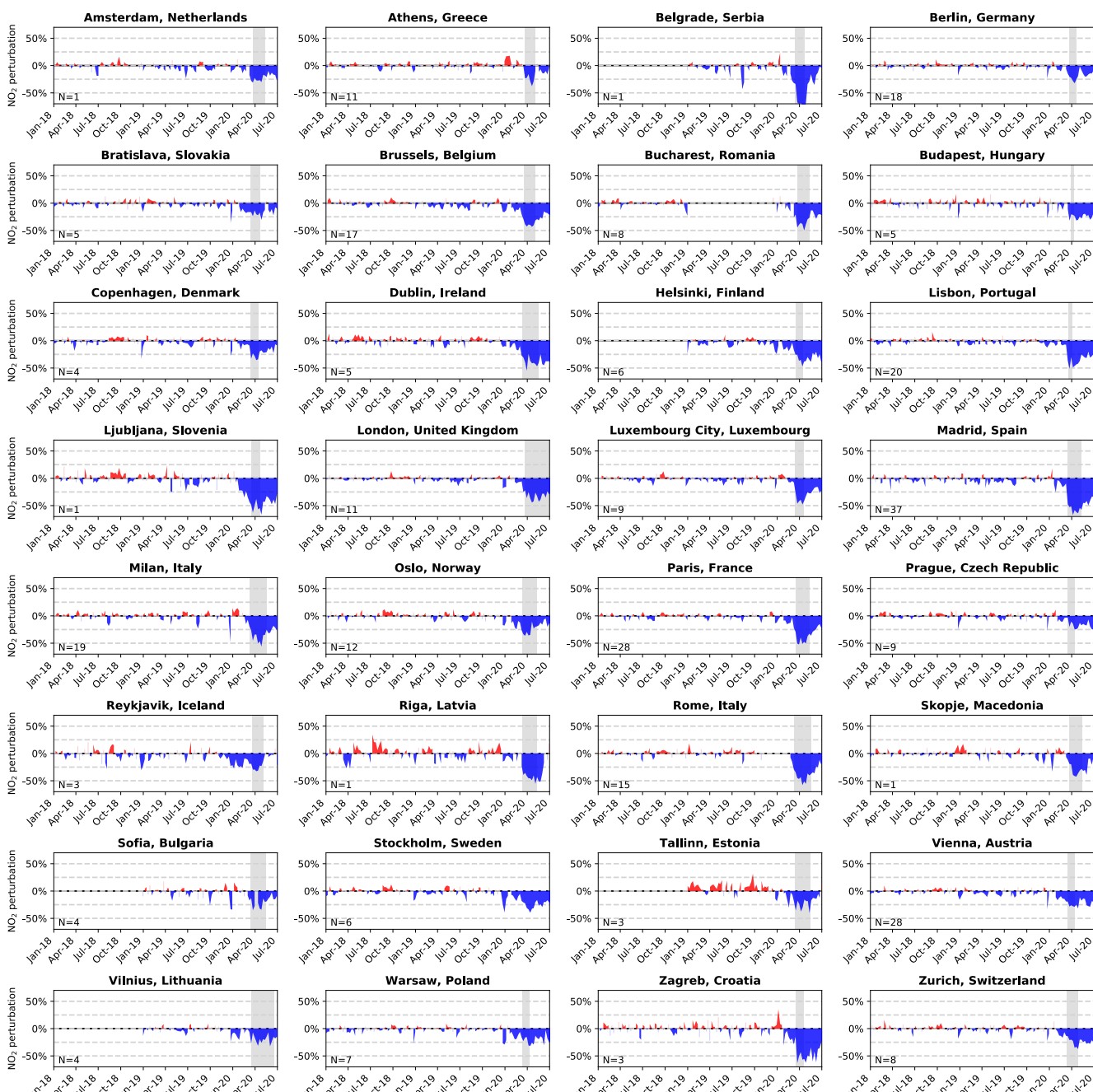

**Figure A7:** As figure A6 but for Europe.


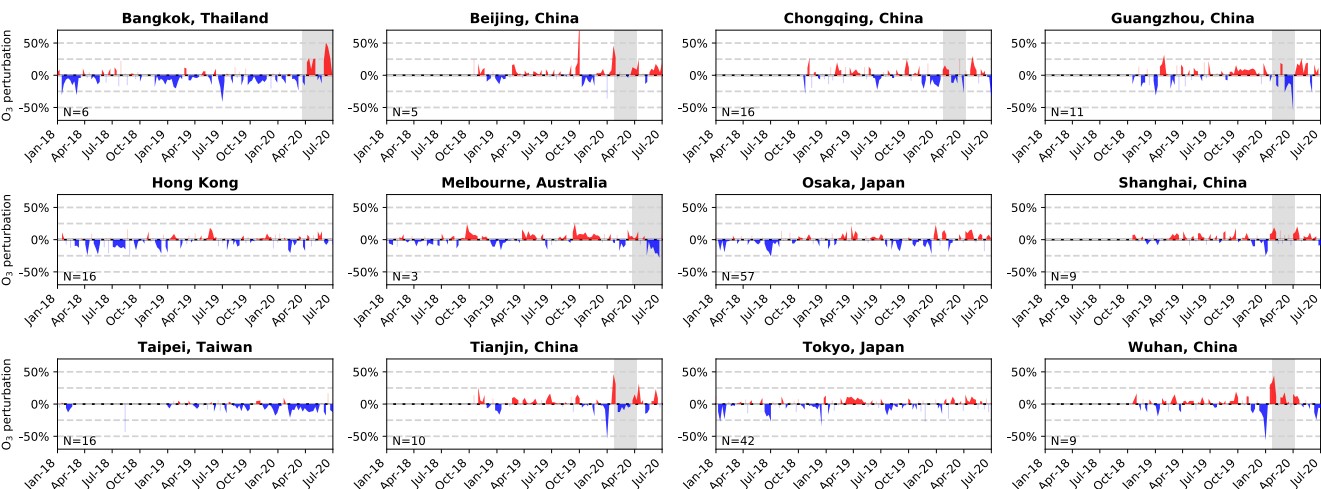

**Figure A8:** As Figure A6 but for North and South America.

**Figure A9:** Normalized fractional $O_3$ perturbations (observation - bias-corrected model, normalized by the bias-corrected model prediction) from Jan 1, 2018 through June 2020 for selected cities in Asia and Australia. Grey shaded areas indicate COVID-19 lockdown periods. Number of sites per city are shown in the bottom left of each panel.

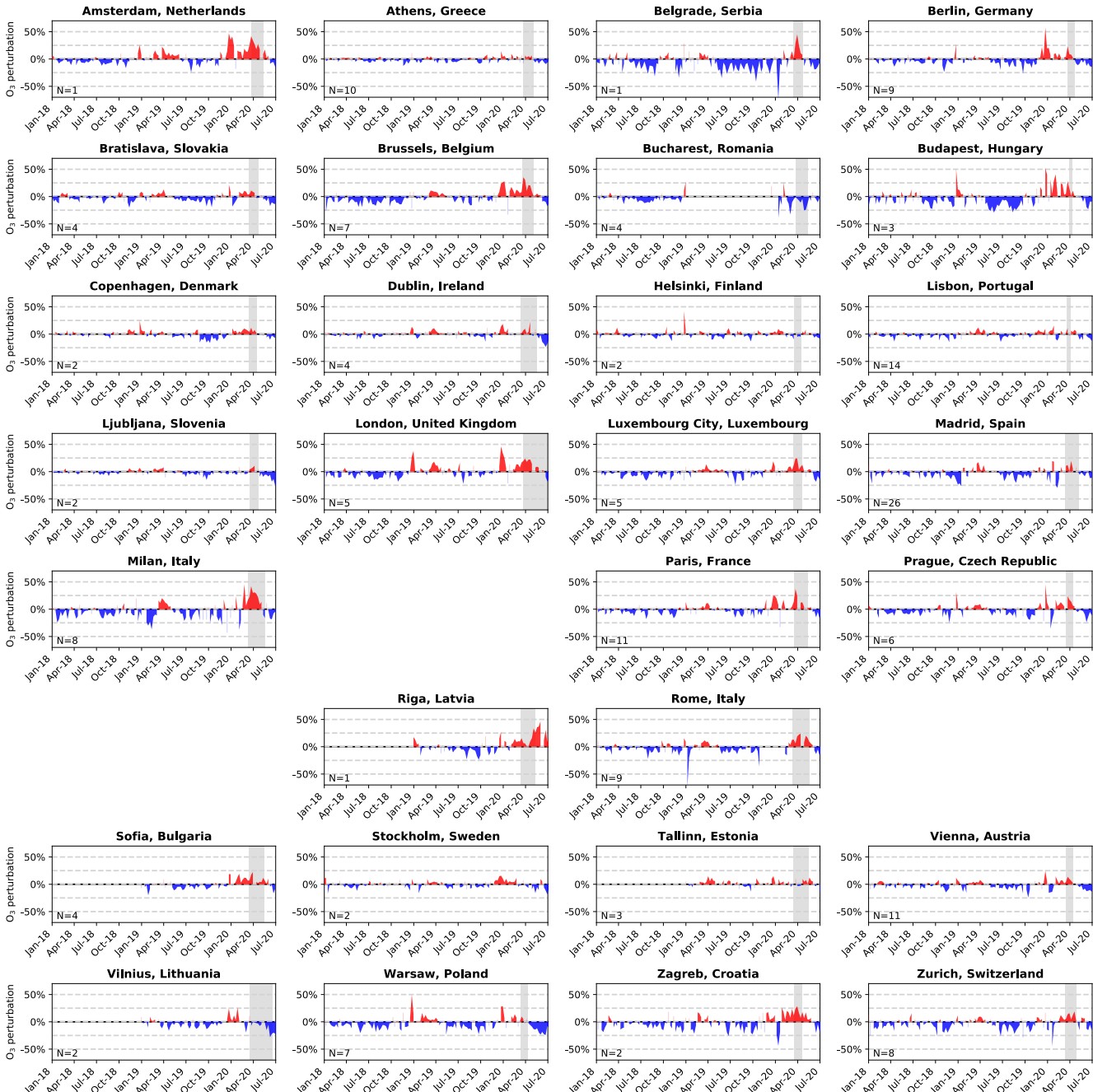

**Figure A10:** As Figure A9 but for Europe. No observations are available for Reykjavik, Oslo, and Skopje.

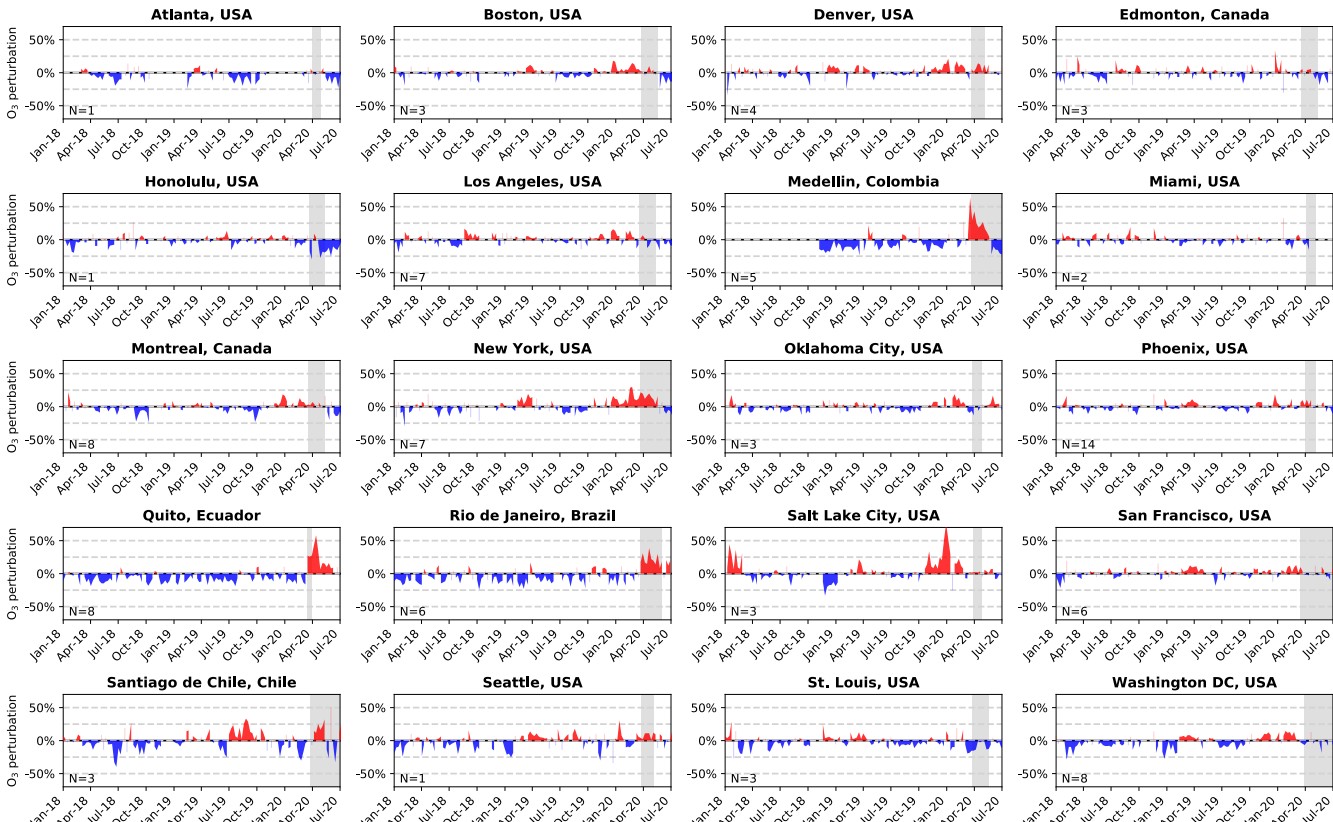

**Figure A11:** As Figure A9 but for North and South America.

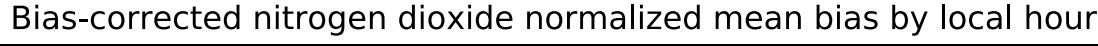

Figure A12: Distribution of normalized mean bias of the machine-learning corrected nitrogen dioxide concentrations at all available observation sites as a function of local hour. Shown are the results for the test dataset.


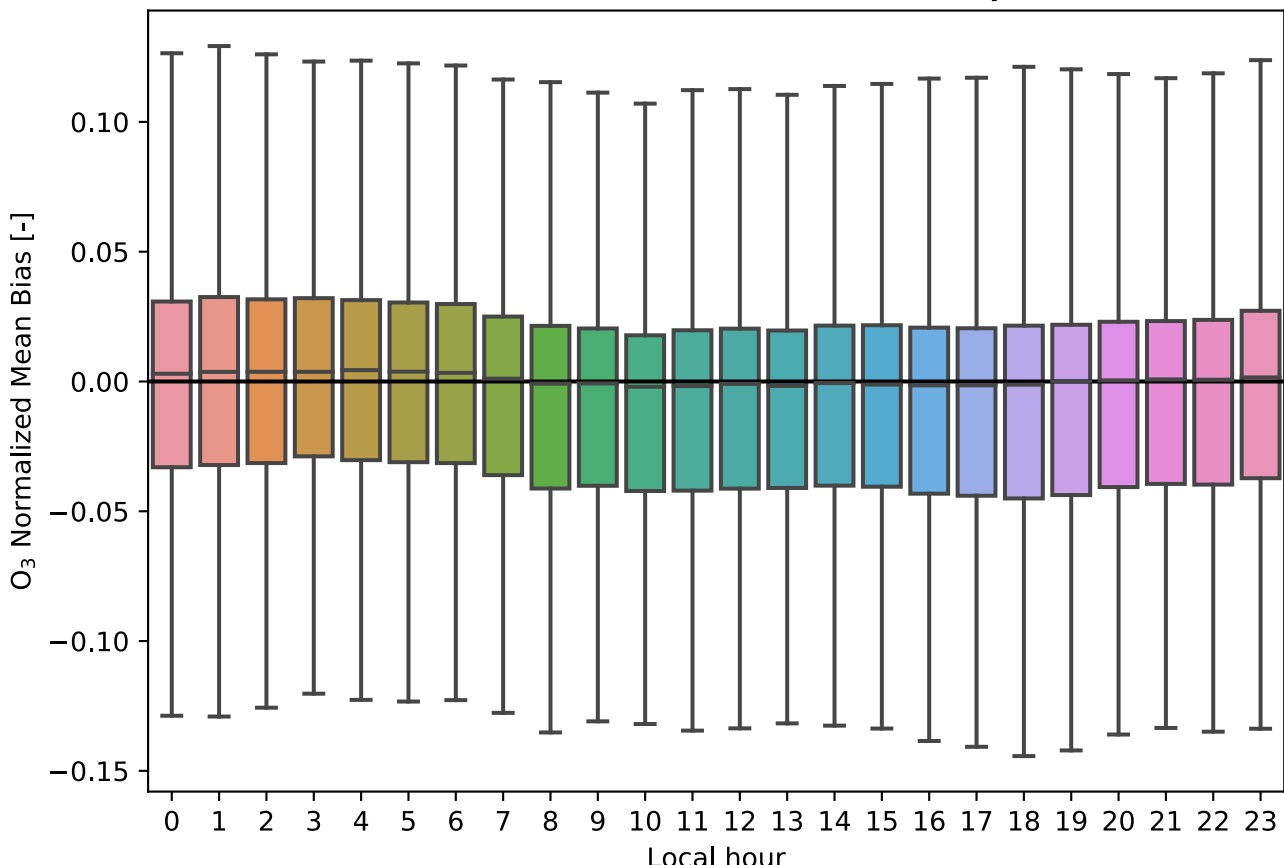

**Figure A13:** As Figure A12 but for ozone.

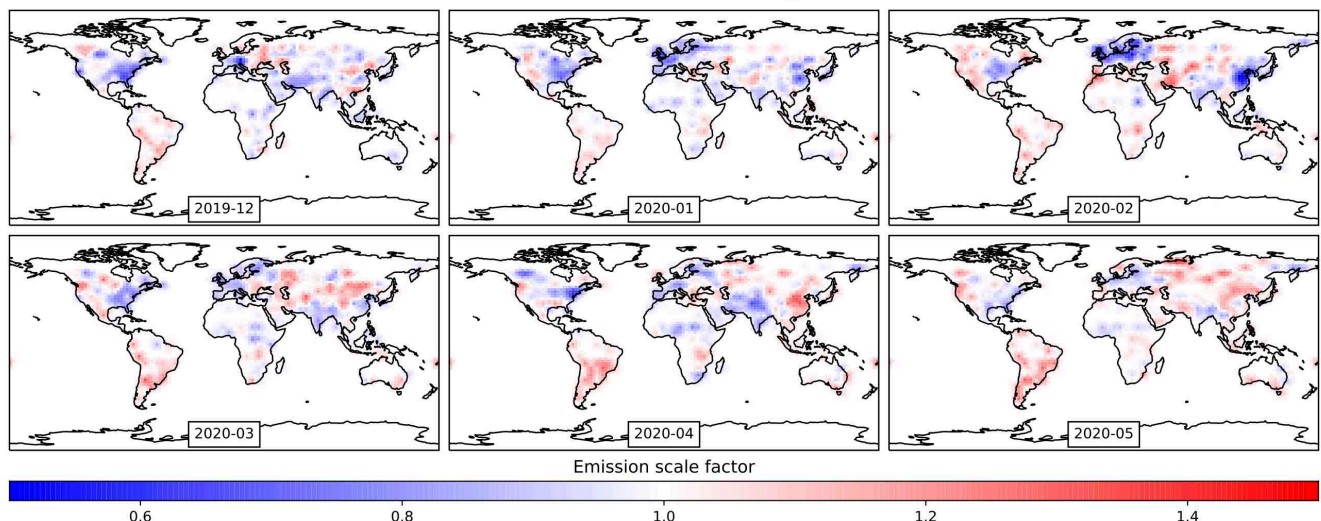

**Figure A14:** Emission scale factors used for model sensitivity simulation. Shown are the monthly average perturbations applied to the GEOS-CF anthropogenic base emissions.

**Table A1:** List of input features fed into the XGBoost machine learning model.

| Short name | Description | Unit |
|---|---|---|
| no2 | nitrogen dioxide | ppbv |
| no | nitrogen oxide | ppbv |
| noy | reactive nitrogen (no + no2 + nitrates) | ppbv |
| o3 | ozone | ppbv |
| co | carbon monoxide | ppbv |
| acet | acetone | ppbv |
| alk4 | alkanes | ppbv |
| ald2 | acetaldehyde | ppbv |
| hcho | formaldehyde | ppbv |
| c2h6 | ethane | ppbv |
| c3h8 | propane | ppbv |
| bcpi | hydrophilic black carbon | ppbv |
| bcpo | hydrophobic black carbon | ppbv |
| ocpi | hydrophilic organic carbon | ppbv |
| ocpo | hydrophobic organic carbon | ppbv |
| eoh | ethanol | ppbv |
| dst1 | dust with diameter of 0.7 microns | ppbv |
| dst2 | dust with diameter of 1.4 microns | ppbv |
| dst3 | dust with diameter of 2.4 microns | ppbv |
| dst4 | dust with diameter of 4.5 microns | ppbv |
| h2o2 | hydrogen peroxide | ppbv |
| hno3 | nitric acid | ppbv |
| hno4 | peroxynitric acid | ppbv |
| isop | isoprene | ppbv |
| macr | methacrolein | ppbv |
| mek | methyl ethyl ketone | ppbv |
| mvk | methyl vinyl ketone | ppbv |
| n2o5 | dinitrogen pentoxide | ppbv |
| nh3 | ammonia | ppbv |
| nh4 | ammonium | ppbv |
| nit | inorganic nitrates | ppbv |
| pan | peroxyacetyl nitrate | ppbv |
| prpe | alkenes | ppbv |
| rcho | aldehyde | ppbv |
| sala | fine sea salt aerosol | ppbv |
| salc | coarse sea salt aerosol | ppbv |
| so2 | sulfur dioxide | ppbv |
| soap | secondary organic aerosol precursor | ppbv |
| soas | simple secondary organic aerosol | ppbv |
| tolu | toluene | ppbv |
| xyle | xylene | ppbv |
| ox | odd oxygen (o3 + no2) | ppbv |
| pm25_gcc | total PM2.5 | µg m-3 |
| pm25ni_gcc | nitrate PM2.5 | µg m-3 |
| pm25su_gcc | sulfate PM2.5 | µg m-3 |
| pm25ss_gcc | sea salt PM2.5 | µg m-3 |
| pm25du_gcc | dust PM2.5 | µg m-3 |
| pm25bc_gcc | black carbon PM2.5 | µg m-3 |
| pm25oc_gcc | organic carbon PM2.5 | µg m-3 |
| pm25soa_gcc | secondary organic aerosol PM2.5 | µg m-3 |
| pm25_gocart | total PM2.5 as calculated by the GOCART model | µg m-3 |

**Table A1:** cont.

| Short name | Description | unit |
|---|---|---|
| Hour | hour of day | - |
| Weekday | day of the week | - |
| Month | month of the year | - |
| Trendday | days since Jan 1, 2018 | days |
| cldtt | total cloud fraction | unitless |
| ps | surface pressure | Pa |
| q10m | specific humidity at 10m | kg/kg |
| t10m | temperature at 10m | K |
| tprec | total precipitation | mm |
| ts | skin surface temperature | K |
| u10m | 10m East-West wind-speed | m/s |
| v10m | 10m North-South wind speed | m/s |
| zpbl | planetary boundary layer height | m |
| emis_no | nitroxen oxide emissions | µg m-2 s-1 |
| emis_co | carbon monoxide emissions | µg m-2 s-1 |
| emis_acet | acetone emissions | µg m-2 s-1 |
| emis_ald2 | acetaldehyde emissions | µg m-2 s-1 |
| emis_alk4 | alkanes emissions | µg m-2 s-1 |
| emis_benz | benzene emissions | µg m-2 s-1 |
| emis_c2h6 | ethane emissions | µg m-2 s-1 |
| emis_prpe | alkenes emissions | µg m-2 s-1 |
| emis_tolu | toluene emissions | µg m-2 s-1 |
| emis_xyle | xylene emissions | µg m-2 s-1 |
| emis_isop | isoprene emissions | µg m-2 s-1 |
| emis_bcpi | hydrophilic black carbon emissions | µg m-2 s-1 |
| emis_bcpo | hydrophobic black carbon emissions | µg m-2 s-1 |
| emis_ocpi | hydrophilic organic carbon emissions | µg m-2 s-1 |
| emis_ocpo | hydrophobic organic carbon emissions | µg m-2 s-1 |
| emis_sala | fine sea salt aerosol emissions | µg m-2 s-1 |
| emis_salc | coarse sea salt aerosol emissions | µg m-2 s-1 |
| emis_so2 | sulfur dioxide emissions | µg m-2 s-1 |
| emis_soap | secondary organic aerosol precursor emissions | µg m-2 s-1 |
| emis_soas | simple secondary organic aerosol emissions | µg m-2 s-1 |
| emis_chbr3 | bromoform emissions | µg m-2 s-1 |

**Table A2:** National lockdown dates used for visualizations.

| Country | Start Date | End Date |
|---|---|---|
| Australia | March 23 | - |
| Austria | March 16 | April 13 |
| Belgium | March 18 | April 4 |
| Bosnia and Herzegovina | March 10 | April 27 |
| Brazil | March 13 | June 2 |
| Bulgaria | March 13 | May 13 |
| Canada | March 18 | May 11 |
| Chile | March 26 | - |
| China | Jan 23 | April 8 |
| Colombia | March 24 | June 1 |
| Croatia | March 18 | April 19 |
| Czech Republic | March 16 | April 12 |
| Denmark | March 13 | April 13 |
| Ecuador | March 16 | March 31 |
| Estonia | March 13 | May 15 |
| Finland | March 16 | April 15 |
| France | March 17 | May 11 |
| Germany | March 23 | April 20 |
| Greece | March 23 | May 4 |
| Hungary | March 28 | April 10 |
| Iceland | March 21 | May 4 |
| India | March 25 | - |
| Ireland | March 12 | May 18 |
| Italy | March 9 | May 18 |
| Latvia | March 13 | May 12 |
| Lithuania | March 16 | June 18 |
| Luxembourg | March 15 | April 20 |
| Macedonia | March 22 | April 12 |
| Malta | March 12 | May 4 |
| The Netherlands | March 23 | May 11 |
| Norway | March 12 | May 11 |
| Poland | March 13 | April 11 |
| Portugal | March 19 | April 2 |
| Romania | March 25 | May 12 |
| Serbia | March 15 | April 21 |
| Slovakia | March 16 | April 22 |
| Slovenia | March 16 | April 20 |
| Spain | March 14 | May 9 |
| Switzerland | March 13 | April 27 |
| Thailand | March 25 | - |
| United Kingdom | March 23 | - |
| United States | March 19 | April 13 |

**Table A3:** Monthly changes in $NO_2$ concentrations relative to the bias-corrected model predictions for cities in Asia & Australia. Values in *italic* denote values that are statistically different from business-as-usual ($p<0.001$ based on Kolmogorov-Smirnov test).

| Location | # | Jan-20 | Feb-20 | Mar-20 | Apr-20 | May-20 | Jun-20 |
|---|---|---|---|---|---|---|---|
| Bangkok, Thailand | 7 | -7.8% (-10.9%--4.6%) | -11.5% (-15.3%--7.7%) | *-20.4% (-25.9%--14.9%)* | *-27.4% (-31.8%--23.0%)* | *-39.1% (-43.6%--34.5%)* | *-25.7% (-30.7%--20.7%)* |
| Beijing, China | 5 | *-21.1% (-23.3%--19.1%)* | *-38.8% (-41.2%--36.3%)* | *-31.7% (-34.6%--28.7%)* | *-25.0% (-28.7%--21.4%)* | *-23.8% (-27.5%--20.2%)* | *-20.4% (-24.1%--16.6%)* |
| Chongqing, China | 16 | -11.1% (-13.0%--9.2%) | *-38.4% (-40.5%--36.3%)* | -11.6% (-13.6%--9.5%) | -0.1% (-2.1%-1.9%) | 4.3% (2.0%-6.5%) | 3.5% (1.1%-6.0%) |
| Guangzhou, China | 11 | -16.6% (-19.5%--13.7%) | *-31.8% (-35.1%--28.4%)* | -13.6% (-16.7%--10.5%) | -5.6% (-8.7%--2.5%) | -11.5% (-15.6%--7.5%) | -5.8% (-11.1%--0.6%) |
| Hong Kong | 16 | -6.9% (-8.8%--4.9%) | -13.9% (-16.0%--11.8%) | -3.7% (-5.8%--1.6%) | -4.1% (-6.1%--2.1%) | -4.7% (-6.9%--2.4%) | *-12.8% (-15.7%--10.0%)* |
| Melbourne, Australia | 2 | 4.6% (-5.6%-14.8%) | 1.5% (-9.1%-12.1%) | -7.6% (-16.5%-1.4%) | *-13.9% (-21.9%--5.8%)* | -11.9% (-19.3%--4.9%) | -7.8% (-14.1%--1.6%) |
| Osaka, Japan | 90 | -14.4% (-15.4%--13.4%) | -6.4% (-7.3%--5.5%) | -13.4% (-14.3%--12.5%) | *-25.7% (-26.8%--24.7%)* | *-28.1% (-29.2%--27.0%)* | -17.4% (-18.6%--16.3%) |
| Shanghai, China | 9 | -14.4% (-16.5%--12.3%) | *-39.8% (-42.1%--37.5%)* | *-23.3% (-25.7%--20.8%)* | *-19.3% (-21.7%--17.0%)* | -5.5% (-8.4%--2.7%) | -1.9% (-5.0%-1.2%) |
| Taipei, Taiwan | 18 | *-7.8% (-9.4%--6.2%)* | -6.4% (-8.0%--4.8%) | -5.1% (-6.7%--3.6%) | -2.0% (-3.6%--0.3%) | -3.0% (-4.7%--1.2%) | *-11.1% (-13.0%--9.3%)* |
| Tianjin, China | 10 | -7.2% (-8.7%--5.8%) | *-32.1% (-34.0%--30.3%)* | -14.5% (-16.5%--12.4%) | -8.4% (-10.9%--5.9%) | -1.0% (-3.7%-1.7%) | -2.5% (-5.5%-0.6%) |
| Tokyo, Japan | 79 | 1.6% (0.5%-2.7%) | -2.5% (-3.6%--1.4%) | -13.9% (-15.1%--12.8%) | -21.3% (-22.5%--20.1%) | *-26.7% (-28.0%--25.5%)* | -12.0% (-13.3%--10.6%) |
| Wuhan, China | 9 | *-20.4% (-23.6%--17.3%)* | *-56.3% (-59.4%--53.2%)* | *-51.4% (-54.6%--48.2%)* | *-22.0% (-25.3%--18.7%)* | -15.0% (-18.7%--11.2%) | -11.1% (-15.9%--6.2%) |

**Table A4:** Monthly changes in NO$_2$ concentrations relative to the bias-corrected model predictions for cities in Europe. Values in *italic* denote values that are statistically different from business-as-usual (p<0.001 based on Kolmogorov-Smirnov test).

| Location | # | Jan-20 | Feb-20 | Mar-20 | Apr-20 | May-20 | Jun-20 |
|---|---|---|---|---|---|---|---|
| Amsterdam, Netherlands | 1 | -1.5% (-8.2%-5.3%) | -8.5% (-17.4%-0.4%) | *-16.2% (-25.7%--6.4%)* | *-27.2% (-36.8%--17.6%)* | *-18.4% (-29.9%--6.9%)* | -17.2% (-28.2%--6.3%) |
| Athens, Greece | 11 | *13.8% (11.4%-16.1%)* | 1.9% (-0.2%-4.1%) | -5.0% (-7.1%--3.0%) | *-26.4% (-28.7%--24.2%)* | -8.8% (-10.9%--6.7%) | *-12.7% (-15.0%--10.5%)* |
| Belgrade, Serbia | 1 | 2.8% (-5.1%-10.7%) | -10.6% (-19.8%--1.4%) | *-40.9% (-50.2%--31.6%)* | *-71.7% (-81.4%--62.0%)* | *-31.0% (-44.1%--17.8%)* | -17.0% (-33.4%--0.6%) |
| Berlin, Germany | 18 | 2.1% (0.4%-3.8%) | *-7.3% (-9.4%--5.2%)* | -9.7% (-11.6%--7.8%) | *-25.3% (-27.3%--23.3%)* | *-12.6% (-15.0%--10.2%)* | -15.4% (-17.9%--12.9%) |
| Bratislava, Slovakia | 5 | 0.3% (-2.8%-3.3%) | -15.7% (-19.8%--11.5%) | *-18.3% (-22.4%--14.3%)* | *-18.2% (-22.3%--14.2%)* | *-12.4% (-17.6%--7.1%)* | -14.9% (-20.5%--9.3%) |
| Brussels, Belgium | 17 | -5.2% (-7.1%--3.4%) | -11.8% (-14.2%--9.3%) | *-25.4% (-27.4%--23.4%)* | *-40.8% (-42.6%--39.0%)* | *-30.7% (-33.0%--28.5%)* | -19.7% (-22.3%--17.0%) |
| Bucharest, Romania | 8 | -1.7% (-4.5%-1.0%) | -9.9% (-12.9%--7.0%) | *-21.6% (-24.6%--18.6%)* | *-39.8% (-43.0%--36.5%)* | *-23.3% (-26.9%--19.8%)* | -23.0% (-26.5%--19.5%) |
| Budapest, Hungary | 5 | -2.2% (-5.3%-0.8%) | -9.2% (-12.8%--5.5%) | -12.0% (-15.5%--8.6%) | *-23.6% (-27.2%--20.0%)* | *-24.4% (-28.8%--20.0%)* | -25.7% (-30.2%--21.3%) |
| Copenhagen, Denmark | 4 | -1.7% (-6.4%-2.9%) | -9.6% (-14.9%--4.3%) | -19.0% (-23.4%--14.7%) | *-25.7% (-30.1%--21.4%)* | *-18.7% (-24.1%--13.3%)* | -13.8% (-18.4%--9.1%) |
| Dublin, Ireland | 5 | *-8.7% (-14.7%--2.7%)* | *-9.3% (-17.2%--1.5%)* | *-28.8% (-34.4%--23.1%)* | *-39.2% (-43.7%--34.8%)* | *-38.0% (-44.2%--31.8%)* | *-40.8% (-47.3%--34.2%)* |
| Helsinki, Finland | 6 | -9.1% (-13.2%--4.9%) | -15.2% (-19.2%--11.1%) | *-18.2% (-22.0%--14.5%)* | *-38.8% (-43.0%--34.5%)* | *-33.0% (-37.5%--28.6%)* | -25.9% (-29.7%--22.2%) |
| Lisbon, Portugal | 20 | -6.1% (-8.1%--4.1%) | -3.4% (-5.3%-1.5%) | *-26.9% (-29.4%--24.3%)* | *-41.3% (-44.1%--38.6%)* | *-32.5% (-35.4%--29.6%)* | -28.1% (-32.0%--24.1%) |
| Ljubljana, Slovenia | 1 | -3.8% (-11.6%-4.1%) | -26.0% (-33.7%--18.4%) | *-47.9% (-56.2%--39.5%)* | *-49.8% (-59.1%--40.2%)* | *-41.4% (-53.1%--29.7%)* | -42.6% (-54.2%--31.0%) |
| London, United Kingdom | 11 | *-3.6% (-6.0%--1.1%)* | -4.3% (-7.3%--1.2%) | *-17.0% (-19.7%--14.3%)* | *-35.9% (-38.1%--33.6%)* | *-35.1% (-37.8%--32.3%)* | -28.8% (-31.9%--25.8%) |
| Luxembourg City, Luxembourg | 9 | 3.7% (1.0%-6.4%) | -3.4% (-6.7%--0.1%) | *-24.4% (-27.4%--21.4%)* | *-40.2% (-42.9%--37.6%)* | *-27.0% (-30.1%--23.9%)* | -19.2% (-22.7%--15.7%) |
| Madrid, Spain | 37 | 0.9% (-0.5%-2.4%) | -9.4% (-10.8%--8.0%) | *-31.7% (-33.6%--29.8%)* | *-60.6% (-62.6%--58.6%)* | *-51.8% (-54.0%--49.6%)* | -40.4% (-42.8%--38.0%) |
| Milan, Italy | 19 | *10.0% (8.8%-11.2%)* | -2.0% (-3.6%--0.3%) | *-25.7% (-27.2%--24.2%)* | *-41.6% (-43.3%--39.9%)* | *-34.7% (-36.9%--32.6%)* | -24.4% (-26.6%--22.1%) |
| Oslo, Norway | 12 | -6.9% (-9.3%--4.5%) | -16.0% (-18.5%--13.5%) | *-25.4% (-28.1%--22.6%)* | *-26.3% (-29.4%--23.2%)* | *-18.8% (-22.4%--15.2%)* | -12.2% (-15.7%--8.7%) |
| Paris, France | 28 | -3.2% (-4.5%--1.8%) | -12.7% (-14.4%--11.0%) | *-28.0% (-29.5%--26.6%)* | *-46.6% (-48.0%--45.2%)* | *-35.7% (-37.3%--34.1%)* | -21.1% (-22.9%--19.3%) |
| Prague, Czech Republic | 9 | 3.9% (1.6%-6.2%) | *-9.1% (-12.1%--6.0%)* | -10.1% (-12.6%--7.5%) | *-16.6% (-19.0%--14.2%)* | *-16.7% (-19.5%--13.9%)* | -17.7% (-20.6%--14.8%) |
| Reykjavik, Iceland | 3 | -20.9% (-28.3%--13.5%) | *-17.4% (-23.8%--10.9%)* | *-20.0% (-27.4%--12.6%)* | *-28.1% (-36.3%--19.9%)* | -6.7% (-17.5%-3.0%) | -0.2% (-9.9%-9.4%) |
| Riga, Latvia | 1 | -5.7% (-17.6%-6.3%) | *-7.6% (-20.5%-5.3%)* | *-23.9% (-35.1%--12.7%)* | *-45.5% (-60.5%--30.6%)* | *-48.9% (-62.8%--35.0%)* | -12.6% (-25.0%--0.2%) |
| Rome, Italy | 15 | N/A | -0.9% (-3.3%-1.5%) | *-32.3% (-34.3%--30.2%)* | *-50.0% (-52.1%--47.8%)* | *-39.3% (-41.6%--36.9%)* | -24.2% (-26.8%--21.7%) |
| Skopje, Macedonia | 1 | -9.5% (-17.5%--2.5%) | 5.6% (-2.9%-14.3%) | -8.1% (-16.6%-0.3%) | *-35.5% (-43.9%--27.2%)* | *-32.2% (-41.7%--22.7%)* | -18.0% (-27.8%--8.2%) |
| Sofia, Bulgaria | 4 | 8.5% (3.8%- | -3.4% (-8.4%- | *-16.5% (-* | *-27.3% (-* | -20.3% (- | -10.7% (- |

| | | | | | | | |
|---|---|---|---|---|---|---|---|
| | | 13.2%) | 1.6%) | *21.8%--11.2%)* | *34.7%--19.9%)* | 29.4%--11.1%) | *18.4%--3.5%)* |
| Stockholm, Sweden | 6 | -11.1% (-15.4%--6.8%) | *-11.8% (-15.8%--7.8%)* | *-18.0% (-22.0%--14.1%)* | *-33.0% (-37.1%--28.9%)* | *-23.7% (-28.0%--19.4%)* | *-20.9% (-25.5%--16.4%)* |
| Tallinn, Estonia | 3 | -3.2% (-12.0%-5.6%) | *-12.3% (-21.3%--3.3%)* | *-14.8% (-24.1%--5.4%)* | *-25.2% (-34.3%--16.0%)* | *-22.3% (-31.4%--13.5%)* | *-9.7% (-17.3%--2.1%)* |
| Vienna, Austria | 28 | -2.3% (-3.6%--0.9%) | *-14.4% (-16.3%--12.6%)* | *-20.6% (-22.3%--18.9%)* | *-26.2% (-27.9%--24.5%)* | *-16.5% (-18.8%--14.2%)* | *-25.4% (-27.6%--23.1%)* |
| Vilnius, Lithuania | 4 | -11.8% (-16.7%--7.0%) | -3.5% (-8.7%-1.7%) | -10.1% (-15.0%--5.2%) | *-24.4% (-29.8%--18.9%)* | -16.2% (-22.2%--10.2%) | -20.3% (-26.9%--13.7%) |
| Warsaw, Poland | 7 | -10.4% (-13.1%--7.6%) | *-15.7% (-18.7%--12.6%)* | *-15.2% (-18.0%--12.4%)* | *-26.0% (-28.9%--23.1%)* | *-18.3% (-21.6%--15.1%)* | *-17.5% (-21.4%--13.6%)* |
| Zagreb, Croatia | 3 | 8.0% (2.6%-13.4%) | *-15.2% (-21.1%--9.4%)* | *-32.0% (-38.0%--25.9%)* | *-51.5% (-57.7%--45.2%)* | *-50.0% (-56.7%--43.3%)* | *-42.3% (-50.2%--34.4%)* |
| Zurich, Switzerland | 8 | 2.2% (0.0%-4.5%) | *-11.8% (-14.9%--8.6%)* | *-15.1% (-17.9%--12.3%)* | *-28.0% (-30.9%--25.2%)* | *-23.4% (-27.0%--19.8%)* | *-25.5% (-29.1%--22.0%)* |


**Table A5:** Monthly changes in $NO_2$ concentrations relative to the bias-corrected model predictions for cities in North and South America. Values in *italic* denote values that are statistically different from business-as-usual ($p<0.001$ based on Kolmogorov-Smirnov test).

| Location | # | Jan-20 | Feb-20 | Mar-20 | Apr-20 | May-20 | Jun-20 |
|---|---|---|---|---|---|---|---|
| Atlanta, USA | 3 | 9.7% (4.6%-14.9%) | 0.8% (-4.5%-6.2%) | -3.3% (-8.7%-2.2%) | *-12.4% (-17.9%--6.9%)* | -6.7% (-12.5%--1.2%) | -7.1% (-13.5%--1.1%) |
| Boston, USA | 4 | -8.4% (-12.5%--4.8%) | -5.1% (-9.2%-1.4%) | *-20.7% (-25.8%--15.7%)* | *-23.4% (-29.3%--17.4%)* | *-27.0% (-33.4%--20.6%)* | -14.7% (-20.6%-8.4%) |
| Denver, USA | 4 | -4.5% (-7.5%--1.9%) | -7.5% (-11.1%--4.3%) | -20.5% (-24.6%--16.5%) | -21.2% (-26.1%--16.3%) | -35.8% (-41.3%--30.4%) | -24.1% (-29.5%--18.7%) |
| Edmonton, Canada | 3 | -4.0% (-7.5%--0.7%) | -11.5% (-15.3%--7.6%) | -15.2% (-20.1%--10.3%) | -18.0% (-24.5%--11.4%) | -34.9% (-43.5%--26.3%) | -32.3% (-41.1%--23.5%) |
| Honolulu, USA | 1 | -3.1% (-16.1%-10.0%) | -0.9% (-15.6%-13.9%) | -11.7% (-26.2%-2.8%) | -28.3% (-42.2%--14.3%) | -30.6% (-47.0%--14.3%) | -26.7% (-42.7%--10.6%) |
| Los Angeles, USA | 8 | 4.7% (2.8%-6.9%) | -4.1% (-6.3%--1.9%) | -11.9% (-15.1%--8.7%) | -28.2% (-31.7%--24.8%) | -27.5% (-31.2%--24.2%) | -23.4% (-27.6%--19.3%) |
| Medellin, Colombia | 4 | -2.0% (-6.6%-2.6%) | *11.6% (7.1%-16.0%)* | -7.1% (-12.1%--2.6%) | *-51.9% (-56.3%--47.8%)* | -24.8% (-29.5%-20.1%) | -21.1% (-26.2%--16.1%) |
| Miami, USA | 3 | -4.1% (-14.5%-6.3%) | *-22.6% (-32.8%--12.4%)* | *-12.9% (-21.7%--4.2%)* | *-29.2% (-40.9%--17.4%)* | N/A | N/A |
| Montreal, Canada | 5 | -5.4% (-8.9%--2.1%) | -4.4% (-8.3%-0.8%) | *-22.9% (-27.4%--18.7%)* | *-36.4% (-42.5%--30.7%)* | *-25.8% (-32.5%--19.1%)* | *-14.7% (-21.3%--8.1%)* |
| New York, USA | 4 | -5.5% (-9.1%--2.4%) | -8.0% (-11.4%--4.8%) | *-20.8% (-24.6%--17.0%)* | *-29.7% (-34.2%--25.2%)* | *-25.1% (-30.4%--19.8%)* | *-17.1% (-23.5%--10.7%)* |
| Oklahoma City, USA | 2 | -0.9% (-6.9%-45.0%) | -3.5% (-10.4%-3.4%) | -7.1% (-13.6%--1.3%) | -2.8% (-9.5%-3.8%) | -13.9% (-20.3%--7.4%) | -8.0% (-13.9%--2.2%) |
| Phoenix, USA | 5 | -9.5% (-12.0%--7.2%) | -12.6% (-15.7%--9.5%) | *-19.8% (-23.6%--16.0%)* | *-22.2% (-26.3%--18.1%)* | -13.8% (-18.1%--9.5%) | -11.0% (-16.1%--6.0%) |
| Quito, Ecuador | 8 | -10.7% (-13.9%--7.6%) | 2.5% (-0.5%-5.4%) | *-34.7% (-37.5%--31.8%)* | *-64.3% (-67.2%--61.4%)* | *-55.8% (-58.8%--52.7%)* | *-24.9% (-27.9%--21.8%)* |
| Rio de Janeiro, Brazil | 2 | -4.9% (-11.5%-1.7%) | 0.8% (-5.7%-7.2%) | -5.2% (-11.9%-1.5%) | -15.9% (-22.6%--9.2%) | 6.5% (1.5%-12.2%) | 4.4% (-0.5%-9.4%) |
| Salt Lake City, USA | 3 | -1.4% (-5.0%-2.2%) | -5.5% (-9.8%--1.9%) | *-15.4% (-21.2%--9.8%)* | *-24.7% (-31.9%--17.4%)* | -28.3% (-36.5%--20.0%) | -20.6% (-29.6%--11.6%) |
| San Francisco, USA | 7 | 4.6% (2.1%-7.2%) | -1.8% (-4.4%-0.7%) | *-14.8% (-18.1%--11.5%)* | *-26.7% (-30.8%--22.6%)* | *-18.4% (-22.7%--14.0%)* | *-15.7% (-20.6%--10.8%)* |
| Santiago de Chile, Chile | 4 | -6.6% (-16.9%-3.6%) | -10.5% (-19.2%--1.8%) | -8.4% (-16.4%--1.1%) | -20.9% (-26.8%--14.9%) | -11.8% (-16.7%--7.0%) | -6.9% (-11.9%--1.8%) |
| Seattle, USA | 1 | -8.6% (-18.0%-0.9%) | -2.5% (-11.7%-6.7%) | -12.7% (-22.4%--2.9%) | -34.0% (-44.7%--23.2%) | -28.9% (-41.6%--16.7%) | -19.8% (-33.6%--6.1%) |
| St. Louis, USA | 3 | -4.4% (-9.4%-0.6%) | -11.9% (-17.3%--6.6%) | *-16.4% (-21.7%--11.1%)* | *-24.8% (-31.1%--18.6%)* | -23.9% (-30.4%--17.4%) | -23.8% (-30.0%--17.6%) |
| Washington DC, USA | 7 | -7.0% (-9.9%--4.0%) | *-10.0% (-13.1%--6.9%)* | *-18.4% (-21.9%--14.8%)* | *-22.3% (-26.4%--18.2%)* | *-25.2% (-30.0%--20.3%)* | *-14.6% (-19.4%--9.9%)* |


**Table A6:** Monthly changes in $O_3$ concentrations relative to the bias-corrected model predictions for cities in Asia & Australia. Values in *italic* denote values that are statistically different from business-as-usual (p<0.001 based on Kolmogorov-Smirnov test).

| Location | # | 20-Jan | 20-Feb | 20-Mar | 20-Apr | 20-May | 20-Jun |
|---|---|---|---|---|---|---|---|
| Bangkok, Thailand | 6 | -6.1% (-9.4% - -2.8%) | -2.8% (-6.1% - 0.5%) | -10.2% (-13.9% - -6.5%) | *11.2% (7.4% - 15.1%)* | 3.6% (-2.1% - 9.2%) | *34.7% (28.0% - 41.3%)* |
| Beijing, China | 5 | 10.9% (6.2% - 15.6%) | 1.4% (-2.2% - 4.9%) | 1.7% (-1.1% - 4.5%) | *11.7% (9.4% - 14.0%)* | -4.4% (-6.1% - -2.3%) | 9.5% (8.1% - 11.0%) |
| Chongqing, China | 16 | -3.6% (-7.0% - -0.3%) | -2.4% (-4.9% - 0.1%) | 4.3% (2.6% - 5.9%) | 3.1% (1.6% - 4.6%) | 6.9% (5.8% - 8.1%) | 2.1% (0.8% - 3.3%) |
| Guangzhou, China | 11 | -0.9% (-3.4% - 1.7%) | 5.0% (2.3% - 7.8%) | -15.4% (-18.0% - -12.9%) | 4.8% (2.6% - 7.0%) | 11.3% (9.4% - 13.2%) | -1.5% (-3.8% - 0.9%) |
| Hong Kong | 16 | -0.1% (-2.7% - 2.6%) | -8.3% (-10.9% - -5.6%) | -2.9% (-5.6% - -0.2%) | -2.3% (-4.8% - 0.1%) | 9.0% (5.0% - 12.9%) | -4.5% (-10.3% - 1.4%) |
| Melbourne, Australia | 3 | 1.1% (-3.7% - 6.0%) | -8.9% (-14.4% - -3.4%) | 3.7% (-1.6% - 8.9%) | 2.7% (-2.5% - 7.9%) | -6.1% (-11.6% - -0.7%) | -19.6% (-25.1% - -14.1%) |
| Osaka, Japan | 57 | *12.7% (11.3% - 14.2%)* | -3.6% (-4.8% - -2.4%) | 1.4% (0.4% - 2.3%) | *11.4% (10.7% - 12.2%)* | 3.3% (2.5% - 4.0%) | 1.2% (0.3% - 2.1%) |
| Shanghai, China | 9 | -2.9% (-5.2% - -0.5%) | 5.8% (3.7% - 7.9%) | 0.6% (-1.0% - 2.2%) | 11.4% (9.9% - 12.9%) | 2.1% (0.9% - 3.3%) | 1.5% (-0.0% - 3.0%) |
| Taipei, Taiwan | 16 | -0.9% (-2.4% - 0.6%) | -9.9% (-11.3% - -8.6%) | -8.0% (-9.2% - -6.8%) | -9.6% (-10.6% - -8.6%) | -6.6% (-8.0% - -5.1%) | -5.2% (-7.1% - -3.3%) |
| Tianjin, China | 10 | 5.9% (2.6% - 9.2%) | -8.1% (-10.5% - -5.8%) | -1.4% (-3.4% - 0.6%) | *14.1% (12.4% - 15.8%)* | *-8.4% (-9.7% - -7.2%)* | 7.3% (6.1% - 8.5%) |
| Tokyo, Japan | 42 | -4.5% (-6.3% - -2.7%) | 1.7% (0.2% - 3.3%) | 0.3% (-0.9% - 1.4%) | 6.4% (5.4% - 7.4%) | 3.8% (2.8% - 4.8%) | -1.4% (-2.6% - -0.3%) |
| Wuhan, China | 9 | 8.6% (4.7% - 12.4%) | 12.3% (9.7% - 14.9%) | 4.3% (2.3% - 6.3%) | 10.4% (8.7% - 12.0%) | -2.4% (-3.7% - 1.1%) | -8.2% (-9.7% - -6.7%) |

**Table A7:** Monthly changes in $O_3$ concentrations relative to the bias-corrected model predictions for cities in Europe. Values in *italic* denote values that are statistically different from business-as-usual (p<0.001 based on Kolmogorov-Smirnov test).

| Location | # | Jan-20 | Feb-20 | Mar-20 | Apr-20 | May-20 | Jun-20 |
|---|---|---|---|---|---|---|---|
| Amsterdam, Netherlands | 1 | *21.6% (11.6% - 31.6%)* | 11.4% (5.9% - 16.9%) | *22.5% (16.9% - 28.1%)* | *26.4% (21.1% - 31.6%)* | 7.1% (2.7% - 11.4%) | -3.7% (-8.1% - 0.7%) |
| Athens, Greece | 10 | 1.0% (-1.3% - 3.2%) | 4.3% (1.9% - 6.6%) | 0.0% (-1.9% - 2.0%) | 4.4% (2.6% - 6.1%) | -3.9% (-5.5% - -2.2%) | -5.7% (-7.3% - -4.0%) |
| Belgrade, Serbia | 1 | -25.2% (-39.7% - -10.8%) | 0.9% (-7.3% - 9.1%) | 15.2% (9.0% - 21.5%) | *15.0% (10.1% - 19.9%)* | -5.5% (-10.5% - -0.5%) | -14.3% (-19.8% - -8.8%) |
| Berlin, Germany | 9 | 8.2% (5.4% - 11.1%) | *7.3% (5.5% - 9.2%)* | 7.4% (5.7% - 9.1%) | 1.8% (0.6% - 3.1%) | -6.2% (-7.4% - -4.9%) | -9.0% (-10.2% - -7.7%) |
| Bratislava, Slovakia | 4 | -7.1% (-11.1% - -3.1%) | 8.1% (5.7% - 10.5%) | 6.1% (4.1% - 8.2%) | 1.3% (-0.3% - 2.9%) | -3.2% (-4.8% - -1.3%) | -11.2% (-12.9% - -9.3%) |
| Brussels, Belgium | 7 | 8.7% (5.6% - 11.9%) | *11.2% (9.2% - 13.3%)* | *16.7% (14.5% - 18.8%)* | *18.3% (16.5% - 20.1%)* | 2.9% (1.3% - 4.5%) | -5.9% (-7.4% - -4.3%) |
| Bucharest, Romania | 4 | -12.3% (-16.8% - -7.8%) | 0.4% (-3.1% - 3.8%) | -11.8% (-14.4% - -9.2%) | -9.1% (-11.2% - -6.9%) | -13.3% (-15.5% - -11.2%) | -3.6% (-5.8% - 0.8%) |
| Budapest, Hungary | 3 | 18.2% (8.6% - 27.2%) | *20.1% (15.8% - 24.4%)* | 16.1% (12.4% - 19.8%) | 4.4% (1.6% - 7.3%) | -2.7% (-5.4% - 0.0%) | -13.1% (-15.9% - -10.2%) |
| Copenhagen, Denmark | 2 | 4.9% (2.1% - 7.7%) | 5.1% (3.0% - 7.3%) | 7.5% (5.7% - 9.4%) | 1.8% (0.1% - 3.5%) | -2.4% (-4.0% - -0.5%) | -5.2% (-6.7% - -3.2%) |
| Dublin, Ireland | 4 | *5.6% (2.3% - 8.8%)* | -2.0% (-4.6% - 0.6%) | 2.7% (-0.1% - 5.5%) | *8.7% (5.6% - 11.5%)* | -1.9% (-4.4% - 0.7%) | *-13.8% (-16.7% - -10.9%)* |
| Helsinki, Finland | 2 | 5.5% (0.9% - 10.1%) | -0.3% (-4.4% - 3.9%) | -3.7% (-7.5% - 0.0%) | -2.2% (-5.5% - 1.1%) | 3.1% (0.7% - 5.5%) | -4.7% (-7.1% - 1.7%) |
| Lisbon, Portugal | 14 | 6.6% (4.8% - 8.3%) | -3.8% (-5.5% - -2.2%) | 2.6% (1.5% - 3.7%) | 4.8% (3.8% - 5.8%) | *-6.4% (-7.4% - -5.5%)* | -4.3% (-5.4% - 3.1%) |
| Ljubljana, Slovenia | 2 | -4.9% (-8.5% - -0.9%) | 0.2% (-2.5% - 2.9%) | 0.8% (-1.5% - 3.0%) | 3.5% (1.7% - 5.3%) | -6.4% (-8.3% - 4.0%) | *-12.4% (-14.4% - -10.4%)* |
| London, United Kingdom | 5 | *8.6% (4.4% - 12.8%)* | *5.6% (2.8% - 8.3%)* | *14.9% (12.5% - 17.2%)* | *20.3% (17.9% - 22.7%)* | *6.7% (4.6% - 8.9%)* | -2.5% (-4.7% - -0.3%) |
| Luxembourg City, Luxembourg | 5 | 0.8% (-2.5% - 4.2%) | 3.4% (1.1% - 5.7%) | 13.0% (10.9% - 15.2%) | 10.2% (8.3% - 12.1%) | -0.2% (-2.0% - 1.5%) | -9.5% (-11.3% - -7.8%) |
| Madrid, Spain | 26 | 0.7% (-1.3% - 2.7%) | -9.2% (-11.1% - -7.3%) | 4.7% (3.7% - 5.7%) | 6.4% (5.5% - 7.3%) | -2.2% (-3.0% - -1.5%) | -6.1% (-6.8% - -5.4%) |
| Milan, Italy | 8 | 5.3% (-8.6% - 19.2%) | 14.6% (7.9% - 21.3%) | *22.6% (19.1% - 26.1%)* | *25.6% (23.3% - 27.9%)* | 0.7% (-1.1% - 2.6%) | -10.3% (-12.0% - -8.6%) |
| Paris, France | 11 | 5.7% (2.9% - 8.5%) | *3.8% (2.1% - 5.6%)* | *14.5% (12.8% - 16.3%)* | 8.3% (6.9% - 9.6%) | 2.5% (1.2% - 3.7%) | -6.2% (-7.5% - 4.9%) |
| Prague, Czech Republic | 6 | -11.0% (-14.6% - -7.3%) | 10.8% (8.5% - 13.0%) | 9.3% (7.2% - 11.4%) | 6.9% (5.2% - 8.6%) | -2.6% (-4.2% - -1.0%) | -12.6% (-14.2% - -11.1%) |
| Riga, Latvia | 1 | -1.3% (-6.1% - 3.5%) | 8.1% (3.9% - 12.4%) | *11.7% (7.9% - 15.5%)* | 3.7% (0.8% - 6.6%) | *32.6% (28.2% - 37.1%)* | *20.6% (14.6% - 26.6%)* |
| Rome, Italy | 9 | N/A | -5.1% (-8.4% - -1.7%) | 8.2% (6.1% - 10.3%) | *15.4% (13.4% - 17.3%)* | 12.5% (10.8% - 14.3%) | -5.8% (-7.4% - 4.3%) |
| Sofia, Bulgaria | 4 | 3.7% (0.3% - 7.2%) | *11.1% (8.4% - 13.8%)* | 10.6% (8.5% - 12.7%) | 4.0% (1.8% - 6.1%) | 4.7% (2.8% - 6.7%) | -3.8% (-5.6% - 1.8%) |
| Stockholm, Sweden | 2 | 5.8% (2.5% - 9.0%) | 5.8% (2.8% - 8.8%) | 0.6% (-1.9% - 3.1%) | -3.6% (-5.7% - -1.5%) | -8.1% (-10.5% - -5.8%) | -2.2% (-4.4% - 0.1%) |
| Tallinn, Estonia | 3 | 3.1% (-0.5% - 6.7%) | 2.1% (-1.1% - 5.2%) | 1.1% (-1.9% - 4.1%) | 0.2% (-2.4% - 2.8%) | 7.0% (4.1% - 10.0%) | -0.5% (-3.6% - 2.7%) |
| Vienna, Austria | 11 | -8.3% (-10.8% - | *8.0% (6.4% -* | 5.6% (4.2% - | 3.9% (2.8% - | -4.5% (-5.6% - - | -11.1% (-12.2% |

| | | | | | | | |
|---|---|---|---|---|---|---|---|
| | | -5.8%) | *9.6%)* | 6.9%) | 5.0%) | 3.4%) | - -10.0%) |
| Vilnius, Lithuania | 2 | 6.8% (1.8% - 11.7%) | 1.2% (-2.4% - 4.8%) | -3.9% (-6.9% - -0.4%) | -2.7% (-5.0% - -0.3%) | -5.5% (-8.1% - -2.8%) | *-20.3% (-22.9% - -17.7%)* |
| Warsaw, Poland | 7 | 3.3% (-0.2% - 6.8%) | 8.9% (6.5% - 11.4%) | 1.7% (-0.3% - 3.7%) | -2.5% (-4.1% - -0.8%) | *-13.9% (-15.5% - -12.4%)* | *-21.3% (-22.8% - -19.8%)* |
| Zagreb, Croatia | 2 | -12.3% (-22.6% - -2.0%) | 17.3% (11.1% - 23.5%) | *17.8% (13.3% - 22.3%)* | *15.3% (11.4% - 19.1%)* | 7.6% (3.9% - 11.4%) | -7.6% (-11.3% - -3.8%) |
| Zurich, Switzerland | 8 | -12.6% (-16.7% - -8.6%) | 2.8% (0.5% - 5.1%) | 7.3% (5.1% - 9.6%) | *12.4% (10.6% - 14.2%)* | -1.2% (-2.8% - 0.5%) | -6.9% (-8.6% - -5.2%) |

**Table A8:** Monthly changes in $O_3$ concentrations relative to the bias-corrected model predictions for cities in North and South America. Values in *italic* denote values that are statistically different from business-as-usual (p<0.001 based on Kolmogorov-Smirnov test).

| Location | # | Jan-20 | Feb-20 | Mar-20 | Apr-20 | May-20 | Jun-20 |
|---|---|---|---|---|---|---|---|
| Atlanta, USA | 1 | N/A | 7.4% (-8.5% - 23.2%) | -3.4% (-8.1% - 1.3%) | -0.9% (-4.7% - 3.0%) | -3.9% (-8.1% - 0.3%) | -9.9% (-14.5% - -5.3%) |
| Boston, USA | 3 | 6.1% (2.9% - 9.4%) | *8.4% (5.4% - 11.3%)* | 6.2% (4.0% - 8.3%) | 1.9% (-0.2% - 4.1%) | -4.5% (-6.6% - -2.1%) | -7.9% (-10.1% - -5.6%) |
| Denver, USA | 4 | 7.3% (4.1% - 10.6%) | 13.5% (10.3% - 16.7%) | -1.0% (-3.4% - 1.5%) | 7.6% (5.3% - 9.9%) | 6.1% (4.1% - 8.1%) | -0.6% (-2.4% - 1.3%) |
| Edmonton, Canada | 3 | 0.5% (-4.0% - 5.0%) | 1.0% (-2.4% - 4.3%) | -2.9% (-5.2% - -0.2%) | 3.1% (1.2% - 5.1%) | -7.7% (-9.6% - -5.4%) | -5.3% (-7.8% - -2.6%) |
| Honolulu, USA | 1 | 1.0% (-2.3% - 4.4%) | -6.7% (-9.7% - -3.2%) | -9.2% (-12.4% - -5.6%) | -6.4% (-10.5% - -2.3%) | *-18.0% (-22.0% - -14.1%)* | *-15.9% (-20.6% - -11.2%)* |
| Los Angeles, USA | 7 | 1.0% (-1.9% - 3.9%) | 10.0% (7.5% - 12.4%) | -2.5% (-4.5% - 0.6%) | -4.9% (-6.6% - -3.2%) | -5.7% (-7.3% - -4.1%) | -3.6% (-5.2% - -2.0%) |
| Medellin, Colombia | 5 | 1.3% (-3.6% - 6.3%) | *-4.2% (-8.1% - -0.2%)* | *19.0% (15.0% - 23.0%)* | *26.0% (21.0% - 31.0%)* | 8.5% (3.1% - 13.9%) | -14.8% (-20.2% - -9.4%) |
| Miami, USA | 2 | 3.2% (-3.7% - 10.4%) | -5.8% (-10.9% - -0.8%) | 1.2% (-2.0% - 4.4%) | -8.2% (-12.7% - -4.1%) | N/A | N/A |
| Montreal, Canada | 8 | 5.0% (2.5% - 7.5%) | *6.0% (3.8% - 8.1%)* | 3.1% (1.4% - 4.7%) | 3.6% (2.0% - 5.2%) | -2.8% (-4.3% - -1.3%) | -9.2% (-10.8% - -7.6%) |
| New York, USA | 7 | 6.5% (2.9% - 10.1%) | *18.5% (14.9% - 22.1%)* | *15.1% (12.8% - 17.4%)* | 15.3% (13.4% - 17.3%) | 6.7% (4.7% - 8.7%) | -4.4% (-6.3% - -2.5%) |
| Oklahoma City, USA | 3 | 7.6% (3.1% - 12.0%) | *9.1% (6.1% - 12.2%)* | -4.0% (-6.5% - -1.4%) | -3.8% (-5.7% - -1.8%) | 1.8% (-0.6% - 3.8%) | 6.0% (4.0% - 8.1%) |
| Phoenix, USA | 14 | -2.5% (-4.8% - -0.3%) | 3.7% (2.0% - 5.3%) | 3.0% (1.6% - 4.4%) | 4.6% (3.5% - 5.8%) | 0.3% (-0.7% - 1.3%) | -2.0% (-3.0% - -1.0%) |
| Quito, Ecuador | 8 | -9.8% (-13.2% - -6.5%) | -7.1% (-10.1% - -4.0%) | 6.3% (3.4% - 9.1%) | *37.4% (33.7% - 41.1%)* | *11.7% (8.1% - 15.2%)* | *5.9% (2.4% - 9.4%)* |
| Rio de Janeiro, Brazil | 6 | -7.2% (-11.3% - -3.2%) | -5.7% (-9.8% - -1.6%) | *4.0% (-0.1% - 8.1%)* | 23.9% (19.6% - 28.2%) | 17.6% (13.5% - 21.8%) | *11.3% (7.0% - 15.7%)* |
| Salt Lake City, USA | 3 | *29.6% (22.6% - 36.6%)* | *12.3% (7.5% - 17.2%)* | -1.4% (-4.4% - 1.5%) | 1.1% (-1.6% - 3.7%) | 3.1% (0.8% - 5.5%) | -2.0% (-4.2% - 0.3%) |
| San Francisco, USA | 6 | -0.9% (-3.5% - 1.8%) | 8.5% (5.9% - 11.0%) | 7.1% (5.3% - 8.9%) | -2.2% (-3.9% - -0.5%) | 3.6% (1.8% - 5.5%) | -3.4% (-5.6% - -1.3%) |
| Santiago de Chile, Chile | 3 | -2.0% (-5.8% - 1.7%) | *-14.5% (-18.6% - -10.6%)* | -6.9% (-11.0% - -2.7%) | 12.3% (7.1% - 17.4%) | 6.7% (0.3% - 13.1%) | 3.1% (-7.2% - 13.4%) |
| Seattle, USA | 1 | *8.0% (2.2% - 13.8%)* | -8.3% (-14.2% - -2.5%) | 4.9% (-0.2% - 9.9%) | 6.7% (1.9% - 11.5%) | 5.2% (0.7% - 9.7%) | -3.4% (-9.0% - 2.1%) |
| St. Louis, USA | 3 | -5.3% (-12.7% - 2.1%) | 9.5% (5.4% - 13.6%) | *-13.9% (-16.9% - -10.9%)* | -7.3% (-9.9% - -4.7%) | -6.6% (-9.4% - -4.0%) | -3.5% (-5.8% - -0.8%) |
| Washington DC, USA | 8 | 8.5% (5.2% - 11.9%) | *11.1% (8.7% - 13.5%)* | -2.2% (-3.8% - -0.6%) | 0.2% (-1.3% - 1.7%) | -3.6% (-5.1% - -2.1%) | -1.3% (-2.8% - 0.2%) |

*Data availability.* The model output and air quality observations used in this study are all publicly available (see methods). The output from the GEOS-CF sensitivity simulation as well as the bias-corrected model predictions are available from CAK per request.

*Author contributions.* CAK and MJE designed the study and conducted the main analyses. CAH and SM contributed OpenAQ observations. TO provided observations and interpretations for Japan. FCM and BBF provided observations and interpretations for Rio de Janeiro, and MVDS provided observations and interpretations for Quito. RGR provided observations for Melbourne and helped analyze results for Australia. KEK and RAL conducted the GEOS-CF simulations. KEK and CAK conducted the GEOS-CF sensitivity experiments and forecasts. LHF conducted $NO_x$ to $NO_2$ sensitivity simulations. SP contributed to overall study design and context discussion. All authors contributed to the writing.

*Competing interests.* The authors declare that they have no conflict of interest.

*Acknowledgements.* Resources supporting the model simulations were provided by the NASA Center for Climate Simulation at the Goddard Space Flight Center (https://www.nccs.nasa.gov/services/discover). We thank Jenny Fisher (U. Wollongong, Australia) for helpful discussions. CAK, KEK and SP acknowledge support by the NASA Modeling, Analysis and Prediction (MAP) Program. MJE and LHF are thankful for support from the University of York's Viking, HPC facility.

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
