# Peer review of "Global Impact of COVID-19 Restrictions on the Surface Concentrations of Nitrogen Dioxide and Ozone"

_Atmospheric Chemistry and Physics, 2020_

## Referee Comment (RC1) · Anonymous Referee #1 · 14 Oct 2020

Keller et al. are investigating here the impact of COVID-19 restrictions on both NO2 and O3 surface concentrations. To estimate these changes taking into account the influence of the meteorology, they designed an interesting approach relying on a global simulation with the GEOS-CF model primarily bias-corrected using machine learning models. Compared to the recent studies covering this topic, the main strengths of this study are its spatial scale (since more than 5,000 stations in 46 countries are considered) and the fact that two important trace gases are included (NO2 and O3). The authors notably highlighted a strong variability of the NO2 changes in general agreement with the level of mobility restrictions put in the different countries, while a lower response of O3 is found. The paper is well written and relevant for our scientific

community. It should thus be accepted after addressing one single major comment regarding the methodology and other minor suggestions.

Major comment : - My major comment is about the machine learning methodology. Some information are missing or at least confusing, which explains why I classify it as a major comment but it might be only a minor one requiring only to provide more details in the text. My concerns are related to the way training and test datasets are obtained and how the machine learning models are tuned. Due to the substantial autocorrelation typically found in hourly air quality time series, using a random selection for splitting the datasets into training and validation data might lead to too optimistically good performances. For instance, the way I see it, if the model ingests a training data at a time t (and learn the corresponding model bias) and used for prediction at t+1, given that the model includes time features that allow to locate temporally this point, it will simply learn that around that time t, the model error is X, and then consider that the error at t+1 should also be close to X. In other words, the model might not learn properly the relationships between the model error and the features other than the temporal ones (most importantly, the meteorological parameters). In addition, it seems that no cross-validation (using for instance K-fold or time series cross-validation) is performed at any time (the word never appears in the manuscript), while this is important for tuning the models and ensuring robust estimates of their performance. Actually, it seems also that no tuning is performed during the preparation of the machine learning models. Also, there is often some confusion between the terms "training" (the phase in which you train your model), "validation" (the phase in which you tune your model and/or your select among different types of models) and "test" (the phase in which you evaluate the final performance of your final model already tuned). I guess what you mean here by "validation" is "test"? But since you are not mentioning how/if your models are tuned, it is quite confusing. Please clarify your methodology regarding these different points. On top of that, I agree with the comment of the editor that some discussion of the uncertainties of your approach should be included in the paper.

Minor comments : - L73 : Which data availability at the hourly scale are you requiring for considering that a given day is valid? Please add this information - Fig. 1 : Eventually, adding three panels zooming on North America, Europe and East Asia might be useful since the red points are completely hiding the blue and purple points in Europe and Asia (or if there are much less blue/purple points, you could plot them above the red points) - L93 : You used OMI observations for scaling the anthropogenic emissions from 2010 to 2018. Why not scaling emissions up to 2019 included? Does the same procedure applied for 2019 highlights noticeable changes of NOx emissions between 2018 and 2019? - L103 : Please indicate here that your XGBoost is making predictions at the hourly scale. Please also mention clearly that one XGBoost model is trained for each station, independently from the others. - L110 : Which "mean" are you referring to here? The overall mean over the period 2018-2019? Or the seasonal monthly mean? Did you test applying the machine learning without removing outliers? Which performance is obtained ? Strongly stagnant conditions might lead to a peak of NO2, and you want your model to learn this type of event, so at first sight, I don't understand why this step is needed (or even wanted). Please provide here a more complete justification of your methodological choices. - L110 and Figure 2 : Please comment a bit more your results. Notably, I am wondering why your results are different at stations around #3000. Is this a specific region? Do you have any idea of the reason for that? Also, I am curious, why not simply normalising the RMSE by the average concentration? (rather than the range between 5th an 95th percentiles) - L116 : You mention 49 species and 31 modelled emissions : please provide the complete list of the species taken into account here (eventually in Supplementary Material or Appendix) - Section 2.3 : Please indicate the features importance obtained with XGBoost, for both NO2 and O3. This is an information especially interesting in your study since you are using a lot of features, many of them probably not very useful for making the predictions (?) - L151 : Just to know, how these cities have been selected? Were they selected following an objective approach (for instance, all largest cities or cities with strongest data availability) or arbitrarily? - Fig. 5 : It would be useful to indicate the number of stations included

in each country (for instance in the title of each panel). - L182 : I would expect that the machine learning model (trained with data from late 2018 to end of 2019) would learn the reduction of NO2 associated to the Chine New Year, but the results for 2019 presented in Fig. 5 suggest that it is not the case. Any idea of the possible reason for that ? Is it simply because no training data are available in the first part of 2018 ? In any case, this could reduce the trust we have in the prediction done in 2020, at least for this specific country and this specific period of the year. Maybe including a new input variable representing if a day is holiday or not could help solving this issue. - The authors evaluated their machine learning models by checking the mean biases, errors and correlations over the entire period, which is fine for the analysis conducted in Sect. 3.1. The analysis of the diurnal variations of O3 and Ox is interesting but should come with an evaluation of the performance of the machine learning models at the diurnal scale : does the bias of the machine learning models show any diurnal variability ? I think it is important to show (eventually in the Supplement) and discuss a diurnal plot of the bias (similar to Fig. 8a) for both training and test datasets, just to ensure that the very small mean biases obtained over the entire period do not hide error compensation of stronger biases during specific times of the day. - L205 : I am not sure we can consider that the stronger seasonality of O3 compared to NO2 would bring more challenges since the seasonality is expected to be taken into account by the "month" feature. However, prediction business-as-usual O3 is possibly more challenging than for NO2 due to the more complex processes driving its concentration (e.g. secondary pollutant produced by more complex chemical reactions, involving more numerous precursors, potential strong influence of dry deposition, long-range transport) - L225 : The results shown for Ox are based on a subset of stations where both NO2 and O3 collocated measurements are available? If yes, is the same subset used for showing the results of O3 alone? Please clarify this point. In any case, it would be nice to have both O3 and Ox on a similar subset of stations to allow fair comparisons. - L262 : "natural background NO2" - L260 : Is this value of 80% obtained at global scale? How variable it is spatially (and more specifically, from one country to the other)? - L263 : Why using

EDGAR[2015] rather than HTAP[2018] ? - Fig. 9a : Results shown in Fig. A7 figure are not exactly what I would expect and thus deserve more discussion, highlighting more clearly the potential uncertainties. For instance, if I understand correctly, NO2 emissions (estimated using the OMI NO2 tropospheric column taken here as a proxy of the NOx emissions) would have decreased more strongly during February 2020 (before the lockdown) than in March-April (during the lockdown), which is likely not true. A potential issue I see here is that the authors are not taking into the influence of the meteorology on the NO2 tropospheric columns. Also, the final number of 5% of reduction of global NOx emissions should also be discussed. Is it consistent with what we could expect during that COVID-19 period, namely a strong reduction of traffic emissions? (what is the contribution of traffic to global NOx emissions?). Therefore, please discuss in more detail this section. - L305-317 : This paragraph corresponds to a new analysis and should thus be included in a dedicated section rather than in the conclusion. Also, a more detailed information should be provided regarding this work. The authors say "we assume a sustained reduction in global anthropogenic emissions of NOx, CO and VOCs". Which reductions were used for CO and VOC emissions ? Also, given that the estimated reduction of NOx emissions is highly variable in time (Fig. 9b), to what "sustained" corresponds here? - Figures in Appendix : Please increase the resolution of these plots.

---

## Referee Comment (RC2) · Anonymous Referee #2 · 20 Oct 2020

In studying the effect of COVID-19 restrictions on air pollution, the meteorological variability complicates a direct comparison with pre-lockdown periods. The authors are well aware of this, and tackle this problem by comparing ground observations against model simulations based on a business-as-usual emission inventory. Local modelling biases (due to representation error, wrong emissions, meteo, or chemistry) are corrected for by a machine learning approach, trained in a pre-lockdown period (2018-2019). The paper is well written, and presents a sound and well-developed approach, hence I recommend its publications after addressing the following minor issues.

I agree with the major comment of the previous reviewer to provide more details about

the machine learning methodology and how the potential pitfall of autoregression of time series is dealt with.

For NO2, The machine learning approach appears to be surprisingly powerful to adjust a rather coarse chemical transport model (25 x 25 km2 resolution) to the local situation, given the strong gradients found in cities. Figure 2: it would be interesting to make a distinction between (rural) background stations and street stations. Is the bias correction method sufficiently strong to solve the representation error of the latter category?

Figure 4 shows underprediction of the uncorrected model for Milan and Taipei, overprediction for NYC, and alternating under- and overprediction for Wuhan. In my opinion, your analysis in 3.1 lacks some words about what we can learn from the modelling biases. Are representation errors dominating, or are we looking at e.g. wrong emission estimates?

Figure 5, just out of curiosity: is there a reason why so many observation sites in Romania measure significantly higher NO2 than expected by the BCM?

Figure A1: Showing results for more Chinese mega-cities would be instructive, especially given the strong local observation network in China.

Figures A1-A3: Sometimes strong NO2 reductions are already visible months before the official lockdown starts (e.g. Ljubljana, Vienna, Dublin, Boston, and Denver). Any explanation?

Figures A1-A3: the blue and red lobes in the pre-COVID period can be used to estimate the error in your methodology and put the results (e.g. in Table A1-A3) in better perspective.

Figures A1-A3: I am missing an indication of n, the number of observation sites used for each city.

Section 3.2: Personally, I find the results for O3 less striking, although I directly ad-

mit that an O3 analysis is more subtle and less straightforward than NO2. Figure 8a shows the flattening of diurnal cycle, which is used to explain the marginal effect of the measures on average O3 concentrations in Figure 6 and 7. I think it would be more interesting to see these figures for daily peak values of O3, instead of daily mean values.

Section 3.3, lines 247-252: I had to read this several times to understand, and I am still not sure if I do by now. First it is stated that NO2 concentrations do not change 1:1 with changing NOx emissions, but in the following sentence it is suggested that NO2 columns from OMI are used to scale underlying NOx emissions. Also, I can not deduce how the sensitivity study is set up exactly. Please rewrite.

Section 3.3: Your emission reduction results (e.g. Figure 9b) are potentially prone to sampling biases. According to Figure 5, the results for India are based on only 7 stations (!). Furthermore, as the ground-based monitoring stations are typically located in cities, the results reflect emission reductions within cities (such as traffic), but not necessarily emission reductions of other sectors such as industry or power plants. This should be addressed in a short discussion.

Conclusions: lines 305-313 describe an additional experiment about the effect of NOx emission reduction on surface ozone, which, according to my taste, should be shifted backward (e.g. in an additional section 3.4) before the conclusions start.
* * *

---

## Short Comment (SC1) · 2 Nov 2020

*This review was prepared as part of graduate program course work at Wageningen University, and has been produced under supervision of Prof Wouter Peters. The review has been posted because of its good quality, and likely usefulness to the authors and editor. This review was not solicited by the journal.*

**Review of "*Global Impact of COVID-19 Restrictions on the Surface Concentrations of Nitrogen Dioxide and Ozone*" by Keller et al. (2020)**

The paper by Keller et al. (2020) aims to create a 'business as usual' model output to compare with the changes in nitrogen dioxide ($NO_2$) and ozone ($O_3$) concentrations observed during the 2020 lockdown situation. To create this 'business as usual' output (how emissions in the first half of 2020 would have been without lockdown) the authors adapt the NASA GEOS-CF model to include seasonal variability of air pollutants. The result is a biased-corrected model (BCM) that also includes meteorological and compositional information. Publicly available data on $NO_2$ and $O_3$ is used from 5,756 observation sites, from which most are in Europe, North America and China. A machine learning algorithm is used to predict the time-varying bias at each observation site. Reductions in $NO_2$ range from 60% in more severely affected cities as Wuhan, to little difference in less affected cities as Rio de Janeiro. They estimate a reduction of $NO_x$ ($NO+NO_2$) of 2.9 TgN during the first half of 2020, equivalent to 5.1% of the annual anthropogenic total. Following changes in $O_3$ concentrations is more difficult due to competing influences of non-linear atmospheric chemistry. The analysis does indicate a flattening of the $O_3$ diurnal cycle with $O_3$ increasing during the night and decreasing during the day. They do expect that the importance of photochemical production will increase in the Norther Hemisphere, resulting in an overall decrease in surface $O_3$, if $NO_x$ emissions continue to decrease as a result of COVID-19 restrictions.

The main reason for writing this paper is to report new knowledge on a very recent (and still ongoing) event that is of global importance. Keller et al. correctly identify the knowledge-gap for a need of quantification of the reduction in global emissions, fitting the scope of the journal. Their research contributes to the reporting and is based on suitable methods for quantification of reductions. Therefore, this overall well written research should be published. However, not in this current state. My major comment is on the absence of 3 important things that need to be added before publication: (1) statistics, (2) calculation steps and (3) definition of 'lockdown'. The 3 major comments are followed by a list of minor comments.

Major comments:

Major comment (1): Statistics need to be included. The aim of the paper is to quantify, uncertainties should be quantified as well. The reported numbers are easily disregarded without the proper statistics (e.g. p-values, t-tests or z-tests) and uncertainty ranges. This problem is present in figures 4 and 6 (but applies for all given emission changes in the manuscript). Here the difference between the BCM prediction and observations is shown, but without noting whether this difference is significant (or perhaps falls within the uncertainty range of the BCM prediction).

Example 1, line 145: "*For Wuhan, we find a reduction in $NO_2$ of **60%** relative to the expected BCM value for February and March 2020, and similar decreases are found over Milan (**60%**) and New York (**45%**) starting in mid-March and lasting through April (Fig. 4; Tables A1-A3).*"

How certain are these numbers? Is it between 62% and 58%? Or between 70% and 50%? I urge the authors to please quantify the uncertainty of these numbers by providing uncertainty ranges or mentioning of significance. This could be implemented similar to Le Quéré et al. (2020), here

reductions in emissions are provided by stating the range (representing ±1σ) instead of a single number.

Example 2, line 228: "*Compared to the BCM model, there has been an increase in the concentration of night time $O_3$ (midnight-5.00 local time, Fig. 8a) by **1 part per billion by volume** (ppbv = nmol mol-1) compared to the BCM, whereas $O_x$ shows a decrease of **1 ppbv** (Fig. 8b).*"

Is this reported 1 ppbv difference significant? I highly suggest you to report whether the modelled change is significantly different from the observations. The recent paper by Liu et al., (2020), also referenced by Keller et al., does report significance and thereby makes a more compelling case. Liu et al. derived uncertainty from 10000 Monte Carlo simulations from monthly statistics to estimate a 68% confidence interval. This procedure could be followed here as well. Another suggestion is to provide a paragraph on uncertainty estimation for the machine learning algorithm in the method section, similar to Petetin et al. (2020). Perhaps here the method of Hengl et al. (2017) could be useful. They describe a procedure for machine-learning uncertainty estimation with the use of the program R and the package 'xgboost'.

Major comment (2): It's unclear how numbers in the result section are constructed from the represented data, no calculation steps are mentioned in the method section. Most importantly, how is the reduction in global NOx emissions of 2.9 TgN calculated?

Line 253 states the following: "*This results in anthropogenic emission adjustment factors of **0.3 to 1.4** (Fig. A7).*"

Because of the lack of clarification on calculation steps or argumentation, it is unclear how the adjustment factors of 0.3 and 1.4 are determined. Is perhaps the approach of Mendoza & Russel (2001) used to derive adjustment factors for $NO_x$ emissions? Please refer to the used methodology or provide the calculation steps. The in the manuscript referred figure A7 does not provide the calculations either (even though this seems to be suggested). Figure A7 only shows the monthly average perturbations applied to the 2018 anthropogenic base emissions, ranging from 0.5 to 1.5. As a consequence, the resulting quantification of reduction in emissions loses credibility.

Lines 262-266: "*Based on bottom-up emissions estimates for 2015 from the Emission Database for Global Atmospheric Research (EDGAR v5.0_AP, Crippa et al., 2018, 2020) and using a constant concentration/emissions ratio of 0.8 based on the best fit line obtained from the model sensitivity simulation (dashed purple line in Fig. 9a), **we calculate** that the total reduction in anthropogenic $NO_x$ emissions due to COVID-19 containment measures during the first six months of 2020 amounted to **2.9 TgN** (**Fig. 9b** and **Table 2**).*"

It is clear a calculation is performed, but not how. How is the quite important 2.9 TgN reduction in anthropogenic $NO_x$ emission due to COVID-19 containment constructed? The 2.9 TgN is not in the referred Table 2 nor in Figure 9b. I urge the authors to provide the taken calculation steps resulting in the (quite important) 2.9 TgN reduction in anthropogenic $NO_x$ emission. This will improve the credibility of that given number.

Major comment (3): The manuscript mentions 'lockdown' situations but does not provide a definition of 'lockdown'. The restrictions vary per country (Ravindran & Shah, 2020) and the definition will have consequences on changes in $NO_2$ emissions. Some countries only enforced restrictions based on time, while keeping most forms of transport, schools and business open. Others have been reported to only had restrictions for part of the country. Please provide a definition of 'lockdown'.

Lines 156-162: *For Taipei and Rio de Janeiro, the observations and the BCM show little difference (Fig. 4), consistent with* **the less stringent** *quarantine measures in these places. Other cities with only short-term NO$_2$ reductions of less than 25% include Atlanta (USA), Budapest (Hungary), and Melbourne (Australia), again correlating with the* **comparatively relaxed** *containment measures in these places (Fig. A1-A3). In contrast, Tokyo (Japan) and Stockholm (Sweden), which also implemented* **a less aggressive COVID-19 response***, exhibit NO$_2$ reductions comparable to those of cities with official lockdowns (>20%), suggesting that economic and human activities were similarly subdued in those cities."*

This suggests that degrees of reduction in NO$_2$ emissions are linked to severity in measures taken by local governments (e.g. lines 156-162), however, the severity of measures per country are not characterised. I suggest providing an overview of 'lockdowns' via a table including severity of measures and start and end dates. As an example, take a look at Ravidran & Shah (2020), where countries were classified on severity by introducing colour codes.

Line 142: *The start and end dates for these are from https://en.wikipedia.org/wiki/COVID19_pandemic_lockdowns or based on local knowledge."*

Because of Wikipedia's quickly changing contents, stating the start and end dates in a table will be an improvement on the derived results and will be more concrete than the stated 'local knowledge'.

Lines 21-22: *Reductions in NO$_2$* **correlate** *with timing and intensity of COVID-19 restrictions, ranging from 60% in severely affected cities (e.g., Wuhan, Milan) to little change (e.g., Rio de Janeiro, Taipei)."*

Also, the manuscript mentions correlations in timing and intensity of COVID-19 restrictions and reductions in NO$_2$ (e.g. lines 21-22). A quantification of this correlation is however missing. Are these findings only based on eying the figures? Was a correlation test performed? I recommend adding quantification of the correlations.

Minor comments:

Table 1: The links for AEROS (Japan) and EPA Victoria (Australia, Melbourne) do not work.

119: Provide an argumentation on why all observations below or above 2 standard deviations from the mean are removed, contrary to Ma et al. (2020) where observations below or above 3 standard deviations were removed.

Figures 2 and 3: The presentation of the machine learning statistics could be simplified in form of a table. I fail to see how the representation of the machine learning statistics in a graph are useful to the reader (including the location#, since no information is supplied to deduct which location# is which location). I suggest replacing figures 2 and 3 by a table providing statistical performance, similar to Table 4 of Ivatt & Evans (2019).

Figures 4 and 6: Reductions in % are difficult to read in the figures, one must go back to the text for the actual numbers. Consider including the numbers in the figures, so they stand stronger by themselves. Both figures could be shortened on the x-as as well, starting at 2019. The (incomplete) data from 2018 does not contribute to the results. I would even consider replacing both figures 4 and 6 entirely by new figures that better meet the objective of quantifying the difference in reductions of NO$_2$ and O$_3$ concentrations (including notification of significance or uncertainty ranges, see major comment 1).

191: Consider replacing the vague terms 'some countries' and 'most countries'. These results are stronger when presented in numbers, for example: '42 out of 46 countries...'

208: Reconsider the phrasing of this result. Belgium, Italy, Luxembourg and Switzerland do not all four show pronounced peaks in early April, based on Figure 7.

221-225: Consider including chemical equations of the mentioned processes to improve readability of this paragraph.

252-258: Move this text to methods, it seems out of place here in the result section.

305-309: Move this text to methods as well, it seems out of place her in the conclusion section.

Figure 10: Consider moving this figure to the result section instead of below the conclusion.

**References:**

Hengl, T., Leenaars, J. G., Shepherd, K. D., Walsh, M. G., Heuvelink, G. B., Mamo, T., ... & Wheeler, I. (2017). Soil nutrient maps of Sub-Saharan Africa: assessment of soil nutrient content at 250 m spatial resolution using machine learning. Nutrient Cycling in Agroecosystems, 109(1), 77-102.

Ivatt, P. D. and M. J. Evans. Improving the prediction of an atmospheric chemistry transport model using gradient boosted regression trees, Atmos. Chem. Phys. Discuss., https://doi.org/10.5194/acp-2019-753, in review, 2019.

Le Quéré, C., Jackson, R.B., Jones, M.W. et al. Temporary reduction in daily global CO2 emissions during the COVID-19 forced confinement. Nat. Clim. Chang., https://doi.org/10.1038/s41558-020-0797-x, 2020

Liu, Z., Deng, Z., Ciais, P., Lei, R., Davis, S. J., Feng, S., ... & Zhu, B. (2020). COVID-19 causes record decline in global CO2 emissions.

Ma, J., Ding, Y., Cheng, J. C., Jiang, F., Tan, Y., Gan, V. J., & Wan, Z. (2020). Identification of high impact factors of air quality on a national scale using big data and machine learning techniques. Journal of Cleaner Production, 244, 118955

Mendoza-Dominguez, A., & Russell, A. G. (2001). Estimation of emission adjustments from the application of four-dimensional data assimilation to photochemical air quality modeling. Atmospheric Environment, 35(16), 2879-2894.

Petetin, H., Bowdalo, D., Soret, A., Guevara, M., Jorba, O., Serradell, K., and C. Pérez García-Pando. Meteorology-normalized impact of COVID-19 lockdown upon NO2 pollution in Spain, Atmos. Chem. Phys. Discuss., https://doi.org/10.5194/acp-2020-446, in review, 2020.

Ravindran, S., & Shah, M. (2020). Unintended consequences of lockdowns: Covid-19 and the shadow pandemic (No. w27562). National Bureau of Economic Research.

---

## Author Comment (AC1) · 23 Dec 2020

We thank the reviewer for his/her time and the thoughtful feedback. Below we list all referee remarks and suggestions (in *italics)* along with our responses.

**Reviewer comment:** *Keller et al. are investigating here the impact of COVID-19 restrictions on both $NO_2$ and $O_3$ surface concentrations. To estimate these changes taking into account the influence of the meteorology, they designed an interesting approach relying on a global simulation with the GEOS-CF model primarily bias-corrected using machine learning models. Compared to the recent studies covering this topic, the main strengths of this study are its spatial scale (since more than 5,000 stations in 46 countries are considered) and the fact that two important trace gases are included ($NO_2$ and $O_3$). The authors notably highlighted a strong variability of the $NO_2$ changes in general agreement with the level of mobility restrictions put in the different countries, while a lower response of $O_3$ is found. The paper is well written and relevant for our scientific community. It should thus be accepted after addressing one single major comment*
*regarding the methodology and other minor suggestions.*

*Major comment :*
*- My major comment is about the machine learning methodology. Some information are missing or at least confusing, which explains why I classify it as a major comment but it might be only a minor one requiring only to provide more details in the text. My concerns are related to the way training and test datasets are obtained and how the machine learning models are tuned. Due to the substantial autocorrelation typically found in hourly air quality time series, using a random selection for splitting the datasets into training and validation data might lead to too optimistically good performances. For instance, the way I see it, if the model ingests a training data at a time t (and learn the corresponding model bias) and used for prediction at t+1, given that the model includes time features that allow to locate temporally this point, it will simply learn that around that time t, the model error is X, and then consider that the error at t+1 should also be close to X. In other words, the model might not learn properly the relationships between the model error and the features other than the temporal ones (most importantly, the meteorological parameters). In addition, it seems that no cross-validation (using for instance K-fold or time series cross-validation) is performed at any time (the word never appears in the manuscript), while this is important for tuning the models and ensuring robust estimates of their performance. Actually, it seems also that no tuning is performed during the preparation of the machine learning models. Also, there is often some confusion between the terms "training" (the phase in which you train your model), "validation" (the phase in which you tune your model and/or your select among different types of models) and "test" (the phase in which you evaluate the final performance of your final model already tuned). I guess what you mean here by "validation" is "test"? But since you are not mentioning how/if your models are tuned, it is quite confusing. Please clarify your methodology regarding these different points. On top of that, I agree with the comment of the editor that some discussion of the uncertainties of your approach should be included in the paper.*

**Author's response:** We updated the machine learning methodology in the revised version of the manuscript and expanded its description. In the updated manuscript, all models have been trained using 8-fold cross validation, where the 8 batches represent quarterly chunks of model-observation pairs in order to minimize possible autocorrelation impacts. We also updated the notation of 'validation' and 'test' datasets.

These updates led to a slight deterioration of the model skill scores but have no discernible impact on the overall results and conclusions.

**R:** *Minor comments :*
*- L73 : Which data availability at the hourly scale are you requiring for considering that a given day is valid? Please add this information*
**A:** We only include days with at least 12 hours of valid data. We added this information to the manuscript.

**R:** *- Fig. 1 : Eventually, adding three panels zooming on North America, Europe and East Asia might be useful since the red points are completely hiding the blue and purple points in Europe and Asia (or if there are much less blue/purple points, you could plot them above the red points)*
**A:** We added figures with close-up maps of East Asia, Europe, and North America to the appendix.

**R:** *- L93 : You used OMI observations for scaling the anthropogenic emissions from 2010 to 2018. Why not scaling emissions up to 2019 included? Does the same procedure applied for 2019 highlights noticeable changes of NOx emissions between 2018 and 2019?*
**A:** We recognize that the initiali wording in this paragraph was misleading and we adjusted it in the updated version of the manuscript, with reference to the GEOS-CF description paper recently submitted for review (https://www.essoar.org/doi/10.1002/essoar.10505287.1).

**R:** *- L103 : Please indicate here that your XGBoost is making predictions at the hourly scale. Please also mention clearly that one XGBoost model is trained for each station, independently from the others.*
**A:** We added this information to the manuscript.

**R:** *- L110 : Which "mean" are you referring to here? The overall mean over the period 2018-2019? Or the seasonal monthly mean? Did you test applying the machine learning without removing outliers? Which performance is obtained ? Strongly stagnant conditions might lead to a peak of $NO_2$, and you want your model to learn this type of event, so at first sight, I don't understand why this step is needed (or even wanted). Please provide here a more complete justification of your methodological choices.*
**A:** The main motivation for this approach was to adjust for obviously erroneous observations, such as ozone or nitrogen dioxide concentrations of several thousand ppbv. Such values can occur in the OpenAQ database, whose values are reported in real-time and are not backfilled with quality-controlled data. To support this point, we performed two sensitivity simulations using more stringent thresholds of 3 or 4 standard deviations and did not find any change in our results.

*R: - L110 and Figure 2 : Please comment a bit more your results. Notably, I am wondering why your results are different at stations around #3000. Is this a specific region? Do you have any idea of the reason for that? Also, I am curious, why not simply normalising the RMSE by the average concentration? (rather than the range between 5th an 95th percentiles)*
**A:** We reordered the stations to reflect the four major regions considered in this study (China, Europe, USA, rest of the world). We chose the percentile window as the denominator for the NRMSE because it offers a better reflection of the concentration variability at the given site. The results using the RMSE normalized by the annual mean would look qualitatively very similar.

*R: - L116 : You mention 49 species and 31 modelled emissions : please provide the complete list of the species taken into account here (eventually in Supplementary Material or Appendix)*
**A:** The full list of input features is given in Table A2 in the Appendix.

*R: - Section 2.3 : Please indicate the features importance obtained with XGBoost, for both $NO_2$ and $O_3$. This is an information especially interesting in your study since you are using a lot of features, many of them probably not very useful for making the predictions (?)*
**A:** We added a new paragraph to the revised version of the manuscript, discussing the SHapely Additive exPlanations (SHAP) values for both the $NO_2$ and $O_3$ bias correctors in more detail. The SHAP values are similar to the 'classic' feature importance but better take into account the role of feature interactions. The distribution of all input feature importances is shown in the Figures A4 and A5 in the appendix.

*R: - L151 : Just to know, how these cities have been selected? Were they selected following an objective approach (for instance, all largest cities or cities with strongest data availability) or arbitrarily?*
**A:** We chose these 5 cities rather arbitrarily. Wuhan, Milan and New York represent early outbreak 'hotspots' that received a lot of media attention, and Taipei and Rio de Janeiro offer examples of different government responses to the pandemic (as also reflected in the data). We provide more detail on our motivation for showcasing these 5 cities in the revised version of the manuscript.

*R: - Fig. 5 : It would be useful to indicate the number of stations included in each country (for instance in the title of each panel).*
**A:** We provide the number of sites in the inset of each figure.

*R: - L182 : I would expect that the machine learning model (trained with data from late 2018 to end of 2019) would learn the reduction of $NO_2$ associated to the Chinese New Year, but the results for 2019 presented in Fig. 5 suggest that it is not the case. Any idea of the possible reason for that ? Is it simply because no training data are available in the first part of 2018 ? In any case, this could reduce the trust we have in the prediction done in 2020, at least for this specific country and this specific period of the year. Maybe including a new input variable representing if a day is holiday or not could help solving this issue.*

**A:** This is an excellent comment and the idea to add holidays as an additional input feature is intriguing (albeit somewhat cumbersome to implement on a global dataset!). We are actually quite happy to see that the model did not learn the $NO_2$ reduction associated with Chinese New Year, as such a behavior in our eyes would indicate a possible overfitting. Rather, we hope to capture the 'regular' model bias with the machine learning models and accept the fact that unusual events such as holidays cannot be captured. We feel this is the more conservative approach, especially since for China, we would have only one holiday to train the model on (year 2019).

*R: - The authors evaluated their machine learning models by checking the mean biases, errors and correlations over the entire period, which is fine for the analysis conducted in Sect. 3.1. The analysis of the diurnal variations of O3 and Ox is interesting but should come with an evaluation of the performance of the machine learning models at the diurnal scale : does the bias of the machine learning models show any diurnal variability ? I think it is important to show (eventually in the Supplement) and discuss a diurnal plot of the bias (similar to Fig. 8a) for both training and test datasets, just to ensure that the very small mean biases obtained over the entire period do not hide error compensation of stronger biases during specific times of the day.*
**A:** We added the hourly skill scores of the test data set in the appendix, and also note it in the discussion of the results. Note that the skill scores for the training and validation data show the same indifference to the time of the day.

*R: - L205 : I am not sure we can consider that the stronger seasonality of O3 compared to NO2 would bring more challenges since the seasonality is expected to be taken into account by the "month" feature. However, prediction business-as-usual O3 is possibly more challenging than for NO2 due to the more complex processes driving its concentration (e.g. secondary pollutant produced by more complex chemical reactions, involving more numerous precursors, potential strong influence of dry deposition, long-range transport)*
**A:** We agree with the reviewer and changed the wording in the updated version of the manuscript to reflect the fact that compared to $NO_2$, $O_3$ concentrations are much more influenced by large-scale processes and the local $O_3$ signal is thus expected to be much smaller.

*R: - L225 : The results shown for Ox are based on a subset of stations where both NO2 and O3 collocated measurements are available? If yes, is the same subset used for showing the results of O3 alone? Please clarify this point. In any case, it would be nice to have both O3 and Ox on a similar subset of stations to allow fair comparisons.*
**A:** The analysis is indeed based on the subset of stations where both $NO_2$ and $O_3$ observations are available. We clarified this in the manuscript.

*R: - L262 : "natural background NO2"*
**A:** We changed the wording as suggested.

*R: - L260 : Is this value of 80% obtained at global scale? How variable it is spatially (and more specifically, from one country to the other)?*

**A:** The 80% average sensitivity is the global mean value over the simulated sensitivity period (Dec-Jun). For the emission calculation, we updated the methodology and now use a variable $NO_x/NO_2$ ratio, depending on the inferred (percentage) $NO_2$ decrease. We acknowledge that this is still a simplification as the $NO_x/NO_2$ ratio is variable in both space and time. To account for this, we assign a rather large (absolute) uncertainty of 15% to the $NO_x/NO_2$ sensitivity ratio. We updated the manuscript, figures and tables accordingly.

*R: - L263 : Why using EDGAR[2015] rather than HTAP[2018] ?*
**A:** We chose EDGAR over HTAP because it's baseline inventory is more up-to-date (2015 vs. 2010). We added this information to the manuscript.

*R: - Fig. 9a : Results shown in Fig. A7 figure are not exactly what I would expect and thus deserve more discussion, highlighting more clearly the potential uncertainties. For instance, if I understand correctly, $NO_2$ emissions (estimated using the OMI $NO_2$ tropospheric column taken here as a proxy of the $NO_x$ emissions) would have decreased more strongly during February 2020 (before the lockdown) than in March-April (during the lockdown), which is likely not true. A potential issue I see here is that the authors are not taking into the influence of the meteorology on the $NO_2$ tropospheric columns.*
**A:** The main goal of the sensitivity simulation was to obtain $NO_x/NO_2$ sensitivity ratios for a wide variety of (realistic) emission changes. Rather than using a fixed NOx emission ratio (as e.g., done in Lamsal et al., 2011), we chose to use the OMI $NO_2$ tropospheric columns as a proxy for emission changes. This is an obvious oversimplification but serves the stated goal of the sensitivity simulation. We clarified this aspect in the updated version of the manuscript.

*R: Also, the final number of 5% of reduction of global NOx emissions should also be discussed. Is it consistent with what we could expect during that COVID-19 period, namely a strong reduction of traffic emissions? (what is the contribution of traffic to global NOx emissions?). Therefore, please discuss in more detail this section.*
**A:** Traffic emissions are approximately 27% of total anthropogenic $NO_x$ emissions. Using this information, we estimate that our derived NOx emission reductions correspond to 17-24% of global traffic emissions. We added a discussion on this to the manuscript.

*R: - L305-317 : This paragraph corresponds to a new analysis and should thus be included in a dedicated section rather than in the conclusion. Also, a more detailed information should be provided regarding this work. The authors say "we assume a sustained reduction in global anthropogenic emissions of NOx, CO and VOCs". Which reductions were used for CO and VOC emissions ? Also, given that the estimated reduction of NOx emissions is highly variable in time (Fig. 9b), to what "sustained" corresponds here?*
**A:** We moved this analysis to its own paragraph (Section 3.4.) and added more detail on the methodology of this sensitivity experiment. The emission reduction used for the forecast simulation was fixed at -20%, i.e., assuming no variability in time. This is an obvious simplification but serves the stated purpose of the sensitivity experiment.

*R: - Figures in Appendix : Please increase the resolution of these plots.*

**A:** We changed the layout to 4 panels per column to increase the resolution.

---

## Author Comment (AC2) · 23 Dec 2020

We thank the reviewer for his/her time and thoughtful feedback. Below we list all referee remarks and suggestions (in *italics)* along with our responses.

**Reviewer comment:** *In studying the effect of COVID-19 restrictions on air pollution, the meteorological variability complicates a direct comparison with pre-lockdown periods. The authors are well aware of this, and tackle this problem by comparing ground observations against model simulations based on a business-as-usual emission inventory. Local modelling biases (due to representation error, wrong emissions, meteo, or chemistry) are corrected for by a machine learning approach, trained in a pre-lockdown period (2018- 2019). The paper is well written, and presents a sound and well-developed approach, hence I recommend its publications after addressing the following minor issues.*
*I agree with the major comment of the previous reviewer to provide more details about the machine learning methodology and how the potential pitfall of autoregression of time series is dealt with.*
**Author's response:** We overhauled the machine learning methodology in the revised version of the manuscript to better address the potential issue of auto-correlation, and overall expanded significantly on the description of the methodology and associated uncertainty estimation.

**R:** *For $NO_2$, The machine learning approach appears to be surprisingly powerful to adjust a rather coarse chemical transport model (25 x 25 km2 resolution) to the local situation, given the strong gradients found in cities. Figure 2: it would be interesting to make a distinction between (rural) background stations and street stations. Is the bias correction method sufficiently strong to solve the representation error of the latter category?*
**A:** We found no difference in skill scores between background sites and polluted sites, and added this information to the manuscript.

**R:** *Figure 4 shows underprediction of the uncorrected model for Milan and Taipei, overprediction for NYC, and alternating under- and overprediction for Wuhan. In my opinion, your analysis in 3.1 lacks some words about what we can learn from the modelling biases. Are representation errors dominating, or are we looking at e.g. wrong emission estimates?*
**A:** We added a (short) discussion about the possible reasons for the model bias to the revised version of the manuscript.

**R:** *Figure 5, just out of curiosity: is there a reason why so many observation sites in Romania measure significantly higher NO, than expected by the BCM?*
**A:** The large uncertainty range in Romania was caused by two sites with much higher $NO_2$ concentrations than the BCM. Because we used the overall 5/95% quantiles as uncertainty estimate, this resulted in the shown large uncertainty range. For the updated version, we completely overhauled the uncertainty calculation, which now in our view results in more realistic uncertainty estimates. For instance, our uncertainties are now based on the model-observation mismatches obtained on the test data, and the stated uncertainty estimates are higher for countries with only a few observations compared to countries with a dense network.

*R: Figure A1: Showing results for more Chinese mega-cities would be instructive, especially given the strong local observation network in China.*
**A:** We added three more Chinese cities to the analysis (Chongqing, Guangzhou, and Tianjin).

*R: Figures A1-A3: Sometimes strong NO2 reductions are already visible months before the official lockdown starts (e.g. Ljubljana, Vienna, Dublin, Boston, and Denver). Any explanation?*
**A:** Many countries issued 'soft' stay-at-home orders before the 'hard' lockdowns started, and in many locations the $NO_2$ observations start to reflect this change in human behavior ahead of the lockdowns. We discuss this now in more detail in the newly added Section 2.4 (Lockdown dates).

*R: Figures A1-A3: the blue and red lobes in the pre-COVID period can be used to estimate the error in your methodology and put the results (e.g. in Table A1-A3) in better perspective.*
**A:** As already mentioned above, we reworked the uncertainty estimates based on the model-observation mismatches on the test data. This is similar to the here suggested approach but a bit more restrictive as it is based on the test data only.

*R: Figures A1-A3: I am missing an indication of n, the number of observation sites used for each city.*
**A:** We added this information to the figures.

*R: Section 3.2: Personally, I find the results for $O_3$ less striking, although I directly admit that an $O_3$ analysis is more subtle and less straightforward than $NO_2$. Figure 8a shows the flattening of diurnal cycle, which is used to explain the marginal effect of the measures on average $O_3$ concentrations in Figure 6 and 7. I think it would be more interesting to see these figures for daily peak values of $O_3$, instead of daily mean values.*
**A:** We considered this but were worried about 'sensationalizing' our findings by focusing on the ozone peak values. While focusing on the afternoon (or daytime) ozone values is common, the goal of this study was to analyse the overall impact of COVID-19 lockdowns on ozone and we thus find it more appropriate to show the daily mean changes. The changes in afternoon ozone (as well as nighttime ozone) is discussed in detail in Section 3.2 and highlighted in Figure 8.

*R: Section 3.3, lines 247-252: I had to read this several times to understand, and I am still not sure if I do by now. First it is stated that $NO_2$ concentrations do not change 1:1 with changing $NO_x$ emissions, but in the following sentence it is suggested that $NO_2$ columns from OMI are used to scale underlying $NO_x$ emissions. Also, I can not deduce how the sensitivity study is set up exactly. Please rewrite.*
**A:** We updated the description of the sensitivity experiment in the revised version of the manuscript.

*R: Section 3.3: Your emission reduction results (e.g. Figure 9b) are potentially prone to sampling biases. According to Figure 5, the results for India are based on only 7 stations (!). Furthermore, as the ground-based monitoring stations are typically located in cities, the results*

*reflect emission reductions within cities (such as traffic), but not necessarily emission reductions of other sectors such as industry or power plants. This should be addressed in a short discussion.*

**A:** Our emission estimates for countries such as India or Brazil are indeed susceptible to errors from a variety of sources, including sampling errors and the assumed $NO_x/NO_2$ ratio. We revisited the uncertainty calculation in the new version of the manuscript to better reflect these uncertainties, and also expanded the discussion in section 3.3 (in addition to adding a new section 2.3.4 dedicated to the calculation of uncertainty associated with the machine learning methodology).

*R: Conclusions: lines 305-313 describe an additional experiment about the effect of NOx emission reduction on surface ozone, which, according to my taste, should be shifted backward (e.g. in an additional section 3.4) before the conclusions start.*

**A:** We moved this analysis to a separate section 3.4 (Long-term impact of reduced $NO_x$ emissions on surface $O_3$)

---

## Author Comment (AC3) · 23 Dec 2020

**Referee #3**

We thank Kirsten de Nooijer for the time taken to review this paper and for the thoughtful feedback. Below we list all referee remarks and suggestions (in *italics*) along with our responses.

**Reviewer comment:** *Major comment 1): Statistics need to be included. The aim of the paper is to quantify, uncertainties should be quantified as well. The reported numbers are easily disregarded without the proper statistics (e.g., p-values, t-tests or z-tests) and uncertainty ranges. This problem is present in figures 4 and 6 (but applies for all given emission changes in the manuscript). Here the difference between the BCM prediction and observations is shown, but without noting whether this difference is significant (or perhaps falls within the uncertainty range of the BCM prediction).*

**Author's response:** We updated the uncertainty estimation based on model-observation comparisons on the test dataset, and propagate the estimated uncertainties per location site to a city and country level. The numbers in the updated version of the manuscript include the estimated uncertainties. In addition, we highlight statistically significant concentration changes in the concentration tables provided in the Appendix (Tables A3-A8), using a (stringent) p-value of 0.001.

**R:** *Example 1, line 145: "For Wuhan, we find a reduction in $NO_2$ of 60% relative to the expected BCM value for February and March 2020, and similar decreases are found over Milan (60%) and New York (45%) starting in mid-March and lasting through April (Fig. 4; Tables A1-A3)." How certain are these numbers? Is it between 62% and 58%? Or between 70% and 50%? I urge the authors to please quantify the uncertainty of these numbers by providing uncertainty ranges or mentioning of significance. This could be implemented similar to Le Quéré et al. (2020), here reductions in emissions are provided by stating the range (representing ±1σ) instead of a single number.*

**A:** The estimated uncertainty ranges are provided in the updated version of the manuscript. The stated uncertainties take into account the number of observation sites, so that estimates that are based on fewer sites result in higher uncertainties (all else equal).

**R:** *Example 2, line 228: "Compared to the BCM model, there has been an increase in the concentration of night time O3 (midnight-5.00 local time, Fig. 8a) by 1 part per billion by volume (ppbv = nmol mol-1) compared to the BCM, whereas Ox shows a decrease of 1 ppbv (Fig. 8b)." Is this reported 1 ppbv difference significant? I highly suggest you to report whether the modelled change is significantly different from the observations. The recent paper by Liu et al., (2020), also referenced by Keller et al., does report significance and thereby makes a more compelling case. Liu et al. derived uncertainty from 10000 Monte Carlo simulations from monthly statistics to estimate a 68% confidence interval. This procedure could be followed here as well. Another suggestion is to provide a paragraph on uncertainty estimation for the machine learning algorithm in the method section, similar to Petetin et al. (2020). Perhaps here the method of Hengl et al. (2017) could be useful. They describe a procedure for machine-learning uncertainty estimation with the use of the program R and the package 'xgboost'.*

**A:** We added a section on the uncertainty estimation to the methods (Section 2.3.4) and use these uncertainties to quantify the significance of our findings. Based on this, we conclude that the 1ppbv change is indeed statistically significant, and we now state so in the manuscript.

*R: - Major comment (2): It's unclear how numbers in the result section are constructed from the represented data, no calculation steps are mentioned in the method section. Most importantly, how is the reduction in global NOx emissions of 2.9 TgN calculated?*
**A:** We revisited the description of the emission calculation, offering much more detail on the methodology to hopefully make it easier to follow.

*R: Line 253 states the following: "This results in anthropogenic emission adjustment factors of 0.3 to 1.4 (Fig. A7)." Because of the lack of clarification on calculation steps or argumentation, it is unclear how the adjustment factors of 0.3 and 1.4 are determined. Is perhaps the approach of Mendoza & Russel (2001) used to derive adjustment factors for $NO_x$ emissions? Please refer to the used methodology or provide the calculation steps. The in the manuscript referred figure A7 does not provide the calculations either (even though this seems to be suggested). Figure A7 only shows the monthly average perturbations applied to the 2018 anthropogenic base emissions, ranging from 0.5 to 1.5. As a consequence, the resulting quantification of reduction in emissions loses credibility.*
**A:** As already stated above, we updated the description of the emission calculation and also adjusted the uncertainty estimation, which now includes uncertainties for both the estimated $NO_2$ reductions and the assumed $NO_2/NO_x$ ratio. The emission estimates reported in the revised version of the manuscript now include these uncertainty estimates.

*R: Lines 262-266: "Based on bottom-up emissions estimates for 2015 from the Emission Database for Global Atmospheric Research (EDGAR v5.0_AP, Crippa et al., 2018, 2020) and using a constant concentration/emissions ratio of 0.8 based on the best fit line obtained from the model sensitivity simulation (dashed purple line in Fig. 9a), we calculate that the total reduction in anthropogenic $NO_x$ emissions due to COVID-19 containment measures during the first six months of 2020 amounted to 2.9 TgN (Fig. 9b and Table 2)."*
*It is clear a calculation is performed, but not how. How is the quite important 2.9 TgN reduction in anthropogenic NOx emission due to COVID-19 containment constructed? The 2.9 TgN is not in the referred Table 2 nor in Figure 9b. I urge the authors to provide the taken calculation steps resulting in the (quite important) 2.9 TgN reduction in anthropogenic $NO_x$ emission. This will improve the credibility of that given number.*
**A:** The methodology to calculate the emissions is now described in much more detail, along with a discussion of the corresponding uncertainties.

*R: - Major comment (3): The manuscript mentions 'lockdown' situations but does not provide a definition of 'lockdown'. The restrictions vary per country (Ravindran & Shah, 2020) and the definition will have consequences on changes in $NO_2$ emissions. Some countries only enforced restrictions based on time, while keeping most forms of transport, schools and business open. Others have been reported to only had restrictions for part of the country. Please provide a definition of 'lockdown'.*

**A:** A clear definition of lockdowns is indeed complicated by the various responses, often even within regions of a country. We provide the list of used lockdown dates in the Appendix (Table A2). In general, we emphasize that the main purpose of the lockdown dates are to guide the reader in the interpretation of the figures, rather than using them at 'face value' for statistical analysis. The interpretation of lockdown dates is further complicated by the fact that many countries issued 'soft' lockdowns before the official lockdowns, which already altered human behavior and resulted in a decrease in $NO_2$ concentrations in advance of the official stay-at-home orders. We discuss this problem in the newly added Section 2.4 in the manuscript.

*R: Lines 156-162: For Taipei and Rio de Janeiro, the observations and the BCM show little difference (Fig. 4), consistent with the less stringent quarantine measures in these places. Other cities with only short term NO2 reductions of less than 25% include Atlanta (USA), Budapest (Hungary), and Melbourne (Australia), again correlating with the comparatively relaxed containment measures in these places (Fig. A1-A3). In contrast, Tokyo (Japan) and Stockholm (Sweden), which also implemented a less aggressive COVID-19 response, exhibit $NO_2$ reductions comparable to those of cities with official lockdowns (>20%), suggesting that economic and human activities were similarly subdued in those cities."*
*This suggests that degrees of reduction in $NO_2$ emissions are linked to severity in measures taken by local governments (e.g. lines 156-162), however, the severity of measures per country are not characterised. I suggest providing an overview of 'lockdowns' via a table including severity of measures and start and end dates. As an example, take a look at Ravidran & Shah (2020) where countries were classified on severity by introducing colour codes.*
**A:** We added the lockdown dates used in this study to the Appendix (Table A2) but refrain from adding a lockdown severity measure because we don't feel comfortable with such a number on a country scale. For instance, how should one evaluate the severity of the lockdown for the United States where some cities (e.g. New York) were under a complete lockdown while other places saw little (official) restrictions? Rather, we emphasize in the manuscript that the main reason for adding lockdown dates is to support the visualizations.

*R: Line 142: The start and end dates for these are from https://en.wikipedia.org/wiki/COVID19_pandemic_lockdowns or based on local knowledge."*
*Because of Wikipedia's quickly changing contents, stating the start and end dates in a table will be an improvement on the derived results and will be more concrete than the stated 'local knowledge.*
**A:** We added the list of lockdown dates to the Appendix and also provide the date at which the lockdown dates were accessed.

*R: Lines 21 22: Reductions in $NO_2$ correlate with timing and intensity of COVID-19 restrictions, ranging from 60% in severely affected cities (e. Wuhan, Milan) to little change (e. Rio de Janeiro, Taipei)."*
*Also, the manuscript mentions correlations in timing and intensity of COVID-19 restrictions and reductions in $NO_2$ (e.g. lines 21-22). A quantification of this correlation is however missing. Are these findings only based on eying the figures? Was a correlation test performed? I recommend adding quantification of the correlations.*

**A:** We didn't mean to use the word correlation in the literal sense here, and recognize that its use was misleading. We changed the wording accordingly as we don't think that a correlation analysis of the derived concentration changes to the lockdown dates is scientifically warranted.

-Minor comments:

**R:** *Table 1: The links for AEROS (Japan) and EPA Victoria (Melbourne, Australia) do not work.*
**A:** We couldn't find any issues with the links but updated them again in the updated version of the manuscript.

**R:** *119: Provide an argumentation on why all observations below or above 2 standard deviations from the mean are removed, contrary to Ma et al. (2020) where observations below or above 3 standard deviations were removed.*
**A:** We updated the discussion about the removal of outliers (and its motivation), and also conducted two sensitivity studies using a threshold of 3 and 4 standard deviations, respectively. These sensitivity runs did not show any change in our results.

**R:** *Figures 2 and 3: The presentation of the machine learning statistics could be simplified in form of a table. I fail to see how the representation of the machine learning statistics in a graph are useful to the reader (including the location#, since no information is supplied to deduct which location# is which location). I suggest replacing figures 2 and 3 by a table providing statistical performance, similar to Table 4 of Ivatt & Evans (2019).*
**A:** We updated the figure so that statistics are grouped by region (as suggested by another reviewer), and discuss the statistics in more detail in the newly added Section 2.3.3.

**R:** *Figures 4 and 6: Reductions in % are difficult to read in the figures, one must go back to the text for the actual numbers. Consider including the numbers in the figures, so they stand stronger by themselves. Both figures could be shortened on the x-as as well, starting at 2019. The (incomplete) data from 2018 does not contribute to the results. I would even consider replacing both figures 4 and 6 entirely by new figures that better meet the objective of quantifying the difference in reductions of $NO_2$ and $O_3$ concentrations (including notification of significance or uncertainty ranges, see major comment 1).*
**A:** The percentage reductions are provided in the figures in the appendix as well as the tables, and the uncertainties are stated in the tables. The main objective of Figures 4 and 6 is to introduce the overall concept of our methodology and to show comparisons of observations and model values before and after the bias-correction. The time range 2018-mid-2020 is shown to highlight the full extent of the analysis data and to highlight how the model-observation comparisons look like for the entire previous time period (where available). Most other figures in the manuscript focus on relative changes derived from the bias-corrected model (e.g., the figures in the Appendix or Figures 5 and 7), and we find it important to show the full time series of the baseline model (as well as the effect of the bias-correction) for both $O_3$ and $NO_2$ in at least one figure.

**R:** *191: Consider replacing the vague terms 'some countries' and 'most countries'. These results are stronger when presented in numbers, for example: '42 out of 46 countries...'*

**A:** We updated this to '29 out of 36 countries…'.

*R: 208: Reconsider the phrasing of this result. Belgium, Italy, Luxembourg and Switzerland do not all four show pronounced peaks in early April, based on Figure 7.*
**A:** We changed the wording in the updated version of the manuscript.

*R: 221-225: Consider including chemical equations of the mentioned processes to improve readability of this paragraph.*
**A:** Detailed explanation of ozone chemistry, including the chemical equations, are provided in the references. We added an additional reference to Seinfeld and Pandis (2016) and also provide another reference for the $NO_x/NO_2$ ratio (Shah et al., 2020) in the updated discussion of the emissions calculation.

*R: 252-258: Move this text to methods, it seems out of place here in the result section.*
**A:** We expanded the description of the sensitivity simulation, so that this paragraph now hopefully seems less out of context. We prefer keeping it in this paragraph so that the entire section stands on its own.

*R: 305-309: Move this text to methods as well, it seems out of place here in the conclusion section.*
**A:** This is now discussed in newly added section 3.4.

*R: Figure 10: Consider moving this figure to the result section instead of below the conclusion.*
**A:** This figure is now discussed in the newly added section 3.4.

---

## Author Response (AR2)

**Author's response to reviewer and editor comments**

We are thankful for the constructive additional comments. Below we list all referee remarks and suggestions (in *italics)* along with our responses.

**Editor Comments**

**Editor comment:** *in line 43, it would be more correct to speak about NOx emissions, not NO2 emissions.*
**Author's response:** Thanks for pointing this out, we changed the text accordingly in the revised version of the manuscript.

**Editor comment:** *in the caption of Figure 5, it may be worthwhile to remind the readers that the 2019 data were part of the training data set.*
**Author's response:** We added this information to the caption of Figure 5.

**Anonymous Referee #1**

**Reviewer comment:** *The revised version has been substantially improved but I must say I am still a bit confused about the methodology implemented for training and evaluating the machine learning models.*
*First, the authors are not performing any tuning of their models while this may substantially improve the performance of the predictions. Rather, they are using the default hyper-parameters. Why so? This does not follow the good practices of the field. Is this choice made for computational reasons?*
**Author's response:** Performing hyperparameter tuning across all sites would indeed not be possible due to computational constraints. However, we did perform hyperparameter sensitivity tests at a handful of sites (grid search) and found only marginal improvements in performance. Due to this, we decided to stick with the default XGBoost model parameters. For clarification, we added the following sentence to section 2.3.1 of the revised version of the manuscript:

"The design of the XGBoost framework is determined by a set of hyperparameters, such as the learning rate, maximum tree depth, or minimum loss reduction. While a full hyperparameter optimization across all sites - e.g., by using a grid search approach – would be computationally prohibitive, we conducted hyperparameter sensitivity tests at few selected sites and found that the XGBoost performance only improved marginally at these sites when using other hyperparameter than the model defaults (less than 5% improvement). In addition, we found that the sites respond differently to the same change in hyperparameter setup, suggesting that there is no uniform hyperparameter design that is optimal across all sites. Based on this, we chose to use the default XGBoost model parameters at all locations, with a learning rate of 0.3, minimum loss reduction of 0, maximum tree depth of 6, and L1 and L2 regularization terms of 0 and 1, respectively."

**Reviewer comment:** *Secondly, cross-validation can be used for two different purposes : (1) for tuning the ML model, and/or (2) for estimating the performance of the final model. Given that no*

*tuning is performed, I understand the authors are thus using cross-validation here only to estimate the performance of their models. Then, regarding their strategy, at each station, the authors are training 8 different models (model M[8] trained on X[1,2,3,4,5,6,7] and tested on X[8], model M[2] trained on X[1,3,4,5,6,7,9] and tested on X[2], etc.), which should give them 8 values of RMSE (computed on X[8], X[2], etc., respectively) or any other statistical metric they are interested in. A simple and relatively robust approach to estimate the (test) performance of their predictions would consist in computing the corresponding average RMSE (ideally providing also the standard deviation). Which average RMSEs are obtained following this simple approach? Eventually, another approach could be to first gather all the test subsets on which predictions are made (X[8], X[2], etc.) and compute the overall RMSE. Any of these approaches would provide an estimate of the performance of their predictions. Then, in a second step, in order to get the best possible final ML model, a last ML model (to be used to make predictions in 2020) could be trained using the entire 2018-2019 dataset in order to take benefit from the largest possible dataset during the training phase. The performance previously estimated could be used as a conservative estimate of the performance of this final model ("conservative" because this final model may perform slightly better than the 8 models previously evaluated given that it has been trained on a slightly larger dataset).*

*Rather, for a reason I don't really understand, the authors are finally considering a new model that is the average of the 8 models initially trained ("Once trained, the final model prediction at each location consists of the average prediction of the eight models."), which sounds strange to me. Then, in order to estimate the performance of this final model, the authors are "[omitting] the center week of each training segment from the 8-fold cross validation and use it for testing only". Why one week? All this part of the methodology seems a bit "baroque" to me, both for evaluating a ML model and for taking into account the auto-correlation. Regarding the auto-correlation, considering a 8-fold cross-validation is already an improvement compared to the random splitting proposed in the first version of the manuscript. I do not really understand why the authors then need to left apart only one week for testing.*

*These different aspects of the methodology should be clarified and eventually corrected. The choices made need to be comprehensively described and justified, ideally following the good practices in the field of machine learning.*

**Author's response:** Following the reviewer's suggestion, we recalculated the model skill scores using the left out segment from the 8-fold cross validation as test segment. The methodology description in section 2.3.1 and 2.3.3 has been updated accordingly. The updated skill score values are almost identical to the previous ones.

**Reviewer comment:** *About the estimation of the uncertainties (Section 2.3.4), the authors are computing the uncertainties as the standard deviation of the model-observation residuals. One potential issue I see here is that they are assuming implicitly that individual ML models do not have any bias, which is roughly true when averaging all models at all stations, but not at individual stations where NMB ranges between -20 and +10% roughly (Fig. 3). As an illustration, if we consider an hypothetical ML model that would represent perfectly the observations but with a 1 ppbv (systematic) bias. In this case, the residuals all worth 1 ppbv, and the corresponding standard deviation is thus zero. So this model would be considered as perfect while it is not.*

*Then, another aspect is how to translate uncertainties estimated for hourly predictions at a given individual station to uncertainties over a longer period (7 days for instance) and entire country.*

*While it is likely reasonable to consider that predictions on longer time scales are reduced due to error compensations, it might not be always and fully the case on the spatial dimension on which model-observation errors might be at least partly correlated to each other. Consider for instance a set of 2 stations located close to each other. The concentrations observed at these stations might be quite well correlated to each other given the short distance separating them, as well as the ML predictions given the fact that the input variables used are taken from a geophysical model at 25x25 km resolution. Therefore, the way I understand it, the model-observation residuals at these 2 stations might not fully compensate each other, while the authors implicitly assume so. As a consequence, the uncertainties affecting the combination of these two stations would be reduced by a factor of 1.4 (=2^0.5), which might be overly optimistic, as might also be the uncertainties close to zero mechanically obtained in countries with numerous stations, as shown in Fig. 5. I think this should be further discussed, and the assumptions used to estimate the uncertainties should be clarified.*

**Author's response:** In the revised version of the manuscript, we recalculated the uncertainties based on the model-observation comparisons from the 8 test segments obtained from the 8 fold cross validation. Also, to clarify the assumptions that go into our uncertainty estimation, we added the following paragraph to section 2.3.4 of the revised version of the manuscript:

"This assumes that the errors across individual sites are uncorrelated, which they often are given the very local nature of the bias correction models. In addition, our uncertainty calculation also implies that the aggregated mean error approaches zero. Given that the average mean biases of the machine learning models are clustered around zero (Fig. 2 and Fig. 3), this is a valid general assumption - especially when aggregating across multiple sites. However, it might lead to overly optimistic uncertainty estimates for sites with a relatively large mean bias of 10% or higher."